

# Variations in soil chemical and physical properties explain basin-wide variations in Amazon forest soil carbon densities

Carlos Alberto Quesada[1,*], Claudia Paz[1,2], Erick Oblitas Mendoza[1], Oliver Phillips[3], Gustavo Saiz[4,5] and Jon Lloyd[4,6,7]

[1]Instituto Nacional de Pesquisas da Amazônia, Manaus, Cx. Postal 2223 – CEP 69080-971, Brazil

[2]Universidade Estadual Paulista, Departamento de Ecologia, CEP 15506-900, Rio Claro, São Paulo.

[3]School of Geography, University of Leeds, LS2 9JT, UK

[4]Department of Life Sciences, Imperial College London, Silwood Park Campus, Buckhurst Road, Ascot, Berkshire SL5 7PY, UK

[5]Department of Environmental Chemistry, Faculty of Sciences, Universidad Católica de la Santísima Concepción, Concepción, Chile

[6]School of Tropical and Marine Sciences and Centre for Terrestrial Environmental and Sustainability Sciences, James Cook University, Cairns, 4870, Queensland, Australia

[7]Universidade de São Paulo, Faculdade de Filosofia Ciências e Letras de Ribeirão Preto, Av Bandeirantes, 3900 , CEP 14040-901, Bairro Monte Alegre , Ribeirão Preto, SP, Brazil

*Correspondence to: Beto Quesada (quesada.beto@gmail.com)



**Abstract.**
We investigate the edaphic, mineralogical and climatic controls of soil organic carbon (SOC) concentration
utilising data from 147 pristine forest soils sampled in eight different countries across the Amazon Basin.
Sampling across 14 different World Reference Base soil groups our data suggest that stabilisation
mechanism varies with pedogenetic level. Specifically, although SOC concentrations in Ferralsols and
Acrisols were best explained by simple variations in clay content – this presumably being due to their
relatively uniform kaolinitic mineralogy – this was not the case for less weathered soils such as Alisols,
Cambisols and Plinthosols for which interactions between Al species, soil pH and litter quality seem to be
much more important.  SOC fractionation studies further showed that, although for more strongly
weathered soils the majority of SOC is located within the aggregate fraction, for the less weathered soils
most of the SOC is located within the silt and clay fractions. It thus seems that for highly weathered soils
SOC storage is mostly influenced by surface area variations arising from clay content, with physical
protection inside aggregates rendering an additional level of protection against decomposition. On the other
hand, most of SOC in less weathered soils is associated with the precipitation of aluminium-carbon
complexes within the fine soil fraction and with this mechanism enhanced by the presence of high levels of
aromatic, carboxyl-rich organic matter compounds. Also examined as part of this study were a relatively
small number of arenic soils (*viz.* Arenosols and Podzols) for which there was a small but significant
influence of clay and silt content variations on SOM storage and with fractionation studies showing that
particulate organic matter may accounting for up to 0.60 of arenic soil SOC. In contrast to what were in all
cases strong influences of soil and/or litter quality properties, after accounting for these effects neither
wood productivity, above ground biomass nor precipitation/temperature variations were found to exert any
significant influence on SOC stocks at all.  These results have important implications for our understanding
of how Amazon forest soils are likely to respond to ongoing and future climate changes.



## 1 Introduction

Global estimates for carbon stocks in the top 1 m of soil converge around 1500 Pg (Hiederer and Köchy, 2011), which is nearly three times that of above ground biomass estimates, and about twice the C content of the atmosphere (Batjes, 1996, 2014; Eswaran et al., 1993; Post et al., 1982). Soil depths beyond 1 m generally also contain carbon and therefore increase such soil carbon stock estimates substantially. For example, Jackson et al., (2017) estimate a total carbon stock of 2770 Pg in soils up to 3.0 m deep globally; this being nearly twice the 1.0 m depth estimates. Likewise, current estimates for the Amazon Basin forest region are 36.1 and 66.9 Pg of carbon for the top 0.3 and 1 m respectively (Batjes and Dijkshoorn, 1999), and with deep soil layers in the Eastern Amazon soils (from 1 to 8 m deep) being known to hold as much carbon as is contained in the top soil (Trumbore and Barbosa De Camargo, 2009). This makes the Amazon Basin forest soil carbon stocks of similar magnitude or even higher than the aboveground biomass for the forests themselves; the latter generally taken to total about 90 Pg C (Malhi et al., 2006; Mitchard et al., 2014).

The soil organic carbon pool (SOC) is a function of the amount and quality of organic material entering the soil and its subsequent rate of mineralization, which can be controlled by the various stabilization processes that protect SOC from decomposition (Bruun et al., 2010). For example, organic carbon may be stabilized in mineral soils through interactions with oxides and clay minerals (Kahle et al., 2004; Kaiser and Guggenberger, 2003; Mikutta et al., 2007; Saidy et al., 2012; Saiz et al., 2012; Wiseman and Püttmann, 2006), with SOC physically entrapped in soil aggregates (Baldock and Skjemstad, 2000) and/or stabilized by intermolecular interactions between SOC and the surface of clays and Fe and Al hydroxides (Oades, 1989). Thus, chemical adsorption on mineral specific surface area (SSA) has an important role on C stabilization (Kahle et al., 2003; Saggar et al., 1996, 1999; Saidy et al., 2012).

Specific surface area is itself dependent on clay mineralogy, with 1:1 alumino-silicates such as kaolinite (hereafter simply referred to as 1:1 clays) having low SSA and low cation exchange capacity ($I_E$). This contrasts with 2:1 alumino-silicates such as smectites and illites (hereafter simply referred to as 2:1 clays) having a much larger $I_E$ and SSA (Basile-Doelsch et al., 2005; Lützow et al., 2006). Hydrous Fe and Al oxides also provide reactive surface areas for organic matter binding, and with the content of Fe and Al oxides in soils often having been reported as strongly correlated to C content (Eusterhues et al., 2005; Kleber et al., 2005; Saidy et al., 2012; Wiseman and Püttmann, 2006). Iron and Al hydrous oxides nevertheless show different surface properties to those of clays. Specifically, whilst surface charges of clays are predominantly negative in the tropics (Sanchez, 1976), hydrous oxides generally have positive charges, which can further substantially vary in extent in different oxide types and levels of crystallinity (Cornell and Schwertmann, 1996). Thus, the SSA of clay and oxide mixtures, their chemical nature, and the types of charge predominant in organic matter all may play an important role in the C stabilization process (Saidy et al., 2012).



For acidic soils, SOC stabilization by Fe and Al oxides is likely to be dominated by ligand
exchange (a pH dependent process) involving carboxyl groups of SOC and simple OH groups on the
surface of the oxides (Kaiser and Guggenberger, 2003; Lützow et al., 2006; Wagai and Mayer, 2007): a
similar sorption mechanism to that occurring on the edges of 1:1 clay minerals such as kaolinite (Oades,
1989). Iron and Al oxides can also increase the stabilization of SOC through interactions with clay minerals
via a promotion of the formation of aggregates which then serve help to preserve SOC (Kitagawa, 1983;
Wagai and Mayer, 2007), also forming bridges between negative charges in kaolinite and positive charges
in organic matter mainly conferred by cationic amino (R-NH$_2$) and sulfhydryl (R-SH) groups (Wiseman
and Püttmann, 2006). Other factors such as the pH of soil and the organic matter loading present in the
system also influence C stabilization by mineral surfaces (Saidy et al., 2012).
Hydrous oxides themselves also vary in their capacity to stabilize C, with amorphous Fe and Al
oxides having comparatively higher capacity to stabilize C than more crystalline oxides (Kleber et al.,
2005; Mikutta et al., 2005). For example, on a mass basis, the C sorption capacity of ferrihydrite is 2.5
times higher than that of goethite (Kaiser et al., 2007), while amorphous Al oxides have a greater sorption
capacity than ferrihydrite (Kaiser and Zech, 2000). Despite these complexities, because many heavily
weathered soils consist primarily of kaolinite (Sanchez, 1976) it is common to find strong relationships
between [SOC] and soil clay fraction when only soils dominated by 1:1 clays are considered  (Burke et al.,
1989; Dick et al., 2005; Feller and Beare, 1997; Telles et al., 2003).
A second process that may also protect organic matter against microbial decay and which should
be much more relevant to 2:1 clays soils is the co-precipitation of dissolved organic matter (DOM) with Fe
and Al (Baldock and Skjemstad, 2000; Boudot et al., 1989; Nierop et al., 2002; Scheel et al., 2007). DOM
can be precipitated in the presence of Al, Fe and their hydroxides, with an efficiency of up to 90% of all
DOM present in the solution of some acidic forest soils (Nierop et al., 2002). The extent to which DOM
precipitates is largely influenced by soil pH, with higher pH values leading to an increase in precipitation
(Nierop et al., 2002). This is because pH affects both the solubility of DOM (which decreases at low pH)
and the speciation of Al. At higher pH levels (>4.2) the formation of hydroxide species such as Al(OH)$^3$
and tridecameric Al (Al$_{13}$) controls the solubility of Al, but with Al$^{+3}$ predominating at lower pH.
Moreover, the chemical nature of the carbon inputs into a soil may also potentially influence the nature and
extent of any DOM precipitation reactions, with high molecular weight derived from lignin and tannins
(e.g. aromatic compounds) with a large number of functional groups likely to be preferentially precipitated
from DOM (Scheel et al., 2007, 2008).
The retention of such precipitated DOM in the soil can contribute substantially to total soil C
pools and is considered one of the most important processes of SOC stabilization (Kalbitz and Kaiser,
2008). Indeed, mineralization rates of such metal-DOM precipitates have been reported to be 28 times
lower than that of original DOM, and with the resistance of precipitates against microbial decay increasing
with aromatic C content and large C:N ratios: This then resulting in a relatively stable pool that





accumulates in the soil (Scheel et al. 2007). Exchangeable Al concentrations are often very high for
Amazon Basin forest soils (Quesada et al., 2011), and with Al/OM co-precipitations particularly important
in such developing soils (Kleber et al., 2015), stabilization of DOM by precipitation with Al is likely to be
of considerable importance (and considerably more important than Fe-associated co-precipitations),
especially in the western area of the Amazon Basin where actively evolving soils dominate (Quesada et al.
2010).

Given the range of potential mechanisms discussed above, no single edaphic factor should be
considered the likely overriding control of SOC concentrations for Amazon Basin forest soils. And indeed,
although there is a current perception that clay content alone exerts strong influence over SOC
concentration of Amazon forest soils  (Dick et al., 2005; Telles et al., 2003), all of this work has been done
with highly weathered soils and with SOC from soil characterized by 2:1 mineralogical assemblages not
showing any sort of simple clay content dependency  (Quesada and Lloyd, 2016). This suggests that for
such soils – as has already been shown to be the case for other regions of the world with similar
pedogenetic levels (Bruun et al., 2010; Percival et al., 2000) – that variations in clay quality, oxide content
and metal-DOM interactions are likely to be just as, if not more, important in  influencing the extent of
SOC stabilization.

With the forest soils of the Amazon Basin varying substantially in their chemical and physical
properties (Quesada et al., 2010, 2011), it is important to consider how the different soils of the Basin may
differ in the mechanisms by which they stabilize and store SOC. Specifically, we hypothesized that soil
groups with contrasting pedogenetic development should differ in their predominant mechanism of SOC
stabilization, and that soils which share more similar weathering levels and/or chemical and mineralogical
characteristics should also share similar mechanism of SOC stabilization. Specifically, we rationalized that
strongly weathered soils dominated by 1:1 clays should have their C pools influenced primarily by clay
content. On the other hand, given that Al is the main product of weathering in the less weathered soils of
western Amazonia (Quesada et al. 2011), and with clay contents already shown to not explain well their
SOC densities (Quesada and Lloyd, 2016), we hypothesized that Al / organic matter interactions were
likely to be the main stabilization mechanism for such soils.

Finally, soil organic matter (SOM) is a complex mixture of carbon compounds and different soil
minerals. SOM consists of various functional pools, which are stabilized by different mechanisms, each
associated to a given turnover rate. Aiming to simplify this complexity, several soil organic matter
partitioning methods have been developed to separate SOM in different operationally defined pools or
fractions with contrasting chemical and physical characteristics (Denef et al, 2010). Such fractionation
methods may provide additional support for understanding soil carbon stabilization mechanisms, as well as
provide useful constraints for models of soil carbon dynamics (Trumbore and Zheng, 1996; Zimmermann
et al., 2007).

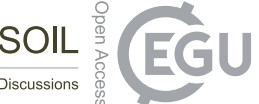

Therefore, we here explore the climatic, edaphic and mineralogical conditioning of soil carbon pools across
the diverse forest soils of the Amazon Basin focusing on three major questions:

1) What are the major edaphic and climatic factors explaining observed variations in soil organic
C across the Basin?;

2) Are the likely contrasting stabilization mechanism patterns hypothesized to operate also
associated with consistently different SOC physicochemical fraction distributions; and

3) How should the contrasting SOC retention mechanisms identified above influence our
understanding of the likely responses of the Amazon Basin forests to future changes in climate?


**2 Materials and Methods**
**2.1 Study sites and sampling**
Soils of 147 1-ha primary forest plots had been sampled across the Amazon Basin as part of this study
(Table 1). These include forests in Brazil, Venezuela, Guyana, French Guyana, Ecuador, Colombia, Peru
and Bolivia (Fig. 1).

Details of soil sampling protocol, laboratory analysis and soil classification can be found in
Quesada et al. (2010, 2011) and are thus only briefly described here. For each site five soil cores were
usually taken across the 1 ha plot to the depth of 2.0 m, with an additional 2.0 m soil pit also sampled in
each plot. Within each soil core, samples were collected over the following standardized depths: 0-0.05,
0.05-0.10, 0.10-0.20, 0.20-0.30, 0.30-0.50, 0.50-1.00, 1.00-1.50 and 1.50-2.00 m using an undisturbed soil
sampler (Eijkelkamp Agrisearch Equipment BV, Giesbeek, The Netherlands) and/or being collected from
the pit walls at the same depths. All samples were air dried as soon as possible with roots, detritus, small
rocks and particles over 2 mm then removed in the laboratory. Samples, sieved at 2 mm, were used in the
laboratory for analysis. Throughout this paper only results for surface soils (0 – 0.30 m) are reported.

**2.2 Soil Classification**
Soils were classified up to their Reference Soil Group (RSG) which represents the great order level in the
World Reference Base for Soil Resources (IUSS (International Union of Soil Science) Working Group
WRB, 2014). The classification performed was based on field and laboratory observations taken following
the standard approach from WRB Guidelines for Soil Descriptions (Jahn et al., 2006).

**2.3 Laboratory analysis**



Soil samples were analysed at different institutions depending on sampling location: Max-Planck Institute
fuer Biogeochemie (MPI), Jena, Germany; Instituto Venezuelano de Investigaciones Cientificas (IVIC),
Caracas, Venezuela; or Instituto Nacional de Pesquisas da Amazonia (INPA), Manaus, Brazil. All
laboratories were linked through inter-calibration exercises and strictly adhered to the same methodologies
and sample standards. For the Venezuelan soils, only cation exchange capacity was measured at IVIC, with
all remaining analysis being determined at MPI and INPA. Soil total reserve bases were analyzed in INPA
and Leeds laboratories (University of Leeds, School of Geography). For samples collected after 2008 (i.e.
not included in Quesada et al. 2010) all analyses were performed in INPA.

**2.3.1 Chemical analysis**
Soil pH was determined in $H_2O$ as 1:2.5. Exchangeable cations were determined at soil pH using the silver
thiourea method ( Ag-TU, Pleysier and Juo, 1980), with the analysis of filtered extracts then done by AAS
at INPA and IVIC or by ICP-OES in MPI. Each sample run was checked and standardized with extracts
from the Montana SRM 2710 soil standard reference (National Institute of Standards of Technology,
Gaithersburg, MD, USA). Effective cation exchange capacity ($I_E$) was calculated as the sum of $[Ca]_E +$
$[Mg]_E + [K]_E + [Na]_E + [Al]_E$, where $[X]_E$ represents the exchangeable concentration of each element in
$mmol_c$ $kg^{-1}$ soil. Total phosphorus was determined by acid digestion at 360 ºC using concentrated sulphuric
acid followed by $H_2O_2$ as described in Tiessen and Moir, (1993). In the same acid digestion extract, total
concentration for Ca, Mg, K and Na was determined and the weathering index Total Reserve Bases, $\Sigma_{RB}$,
calculated. This index is based on total cation concentration in the soil and is considered to give a chemical
estimation of weatherable minerals (Delvaux et al., 1989; Quesada et al., 2010), with $\Sigma_{RB}$ equal to $[Ca]_T +$
$[Mg]_T + [K]_T + [Na]_T$, where $[X]_T$ represents the total concentration of each element in $mmol_c$ $kg^{-1}$ soil.

**2.4 Determination of soil organic C and its fractions**
Concentrations of soil total organic carbon (SOC) and N were determined in an automated elemental
analyzer (Nelson and Sommers, 1996; Pella, 1990). All samples were free of carbonates as confirmed by
their acidic nature (Table 1). The partitioning of SOC in its different fractions was also performed for a
subset of sites (n = 30) as following Zimmermann et al., (2007). This fractionation scheme yields five
different fractions *viz.* labile C associated to the clay and silt (C+S), resistant C associated to clay and silt
($R_{C+S}$), C associated to sand and stable aggregates (S+A), particulate organic matter (POM) and the
dissolved organic C (DOC) component. Samples were dispersed using a calibrated ultrasonic probe-type
operating with an output-energy of 22 J $ml^{-1}$. They were subsequently wet sieved to separate <63 μm
particles (C+S) from >63 μm soil particles (POM + S+A). The entire <63 μm solution was then centrifuged
for 4 min at 1,200 rpm. The C+S obtained after centrifugation was oven dried at 40 °C for 48 hours and
subsequently weighed. The $R_{SOC}$ was obtained by incubating 1 g of C+S with 150 ml of sodium

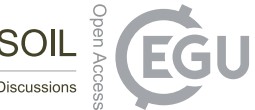

hypochlorite 6% (adjusted to pH 8). After this reaction, the remaining material was washed with distilled
water and oven dried at 40 °C for 48 hours. The labile C+S fraction was determined as the difference of
total C associated to clay and silt and the $R_{C+S}$. The DOC sample was obtained by vacuum filtering an 50
ml aliquot of the total water volume used in the wet sieving (after centrifugation) through a membrane filter
of 0.45μm and had C determined by TOC analyser. S+A and POM were separated following the
procedures described in Wurster et al. (2010) and Saiz et al. (2015). In short, 25 ml of sodium polytungstate
solution (1.8 g/cm$^3$, Sometu- EuropeTM, Berlin, Germany) was added to the >63 μm dried samples placed
in 50 ml centrifuge tubes. Samples were then centrifuged for 15 min at 1,800 rpm and left to rest overnight.
After this time, samples were left in the freezer for approximately 3 hours, after which POM and S+A was
separated by washing the frozen supernatant with distilled water. Both fractions were washed with distilled
water to remove any residue of polytungstate solution then dried at 40 °C for 48 h. All fractions were
analyzed in the same way as SOC. Leaf litter lignin estimates were available for 72 of the 147 sites, having
been obtained using the acid detergent fiber method (Van Soest, 1963) as part of the studies of Quesada
(2008) and Paz (2011).

**2.5 Selective mineral dissolution**

Soil samples were extracted for Fe and Al using established standard techniques as described in detail in
Van Reeuwijk, (2002). In short, replicate samples were shaken for 16h using Dithionite-Citrate and Na-
Pyrophosphate solution. The extraction with ammonium oxalate – oxalic acid solution at pH 3 was
performed in the dark, shaking for 4 hours. All extracts were determined for Fe and Al concentrations in
AAS. These methods provide useful quantitative estimates of soil oxide composition (Parfitt and Childs,
1988). The dithionite-citrate solution dissolves all iron oxides, such as goethite, gibbsite, ferrihydrite,
halloysite, allophane, but with hematite and goethite only partially dissolved. Although this mineral
dissolution method has a broad capacity to estimate Fe and Al in such minerals, it does not differentiate its
various crystalline forms or between short-range (amorphous) minerals and crystalline structures. The
ammonium oxalate – oxalic acid solution on the other hand, specifically dissolves short-range order
minerals such as allophane, imogolite, ferrihydrite, Al-humus complexes, lepidocrocite, Al-vermiculite and
Al hydroxy interlayer minerals. Therefore, the difference between the two methods is often used to estimate
the amount of crystalline minerals in the soil *viz.* ($Fe_d$-$Fe_o$), while negative values indicate the
predominance of short-range minerals. Further interpretation of selective dissolution data according to
Parfitt and Childs (1988) is shown in Table 2.

**2.6 Soil physical properties**

Soil particle size distribution was determined using the pipette method (Gee and Bauder, 1986). Soil bulk
densities were determined using samples collected inside the soil pits at the same depths of other samples



using standard container-rings of known volume (Eijkelkamp Agrisearch Equipment BV, Giesbeek, The
Netherlands). These were subsequently oven dried at 105 ºC until constant weight.

**2.7 Mineralogy**

Soil mineralogical characterization was attained through X-ray diffractometry (XRD) using a PW1050 unit
(Philips Analytical, Netherlands) attached to an X-ray generator DG2 (Hiltonbrooks Ltd, Crewe, UK).
XRD analyses require sample particle size to be very fine in order to obtain adequate statistical
representation of the components and their various diffracting crystal planes, as well as to avoid diffraction-
related artifacts (Bish and Reynolds, 1989). Therefore, samples were ground with a mortar and pestle using
acetone to avoid sample degradation from heat. Powdered samples were then mounted in holders by a back
filled method with the aid of a micro-rugose surface to minimize preferred orientation of the phases
present. Samples were continuously scanned from 3° to 70° (2θ) Ni-filtered CuKα radiation (λ=1.54185Å)
working at 40 kV and 40 mA. The scanning parameters were 0.020° step size and 1.0 sec. step
time. Interpretation and semi-quantitative analysis of the scans were achieved using the Rietveld refinement
method built-in within the Siroquant software (SIROQUANT; Sietronics Pty Ltd, Canberra, Australia). All
samples were analyzed at the Facility for Earth and Environmental Analysis at the University of St.
Andrews, Scotland, UK.

**2.8 Climatic and terrain elevation data**

Mean annual temperature $(T_A)$ and precipitation $(P_A)$ data come from BioClim (www.worldclim.org) and
site elevation $(E_V)$ estimates obtained from the SRTM database.

**2.9 Statistical analysis**

All analyses were carried out using the R statistical platform (R Development Core Team, 2016). In the
exploratory data phase, the non-parametric Kendall τ was used to quantify the strength of bivariate
associations with the aid of the correlation function available within the agricolae package (De Mendiburu,
2017). Multivariate Ordinary Least Squares Regression (OLS) were then performed relating SOC to other
soil properties with candidate variables chosen with reference to the Kendall rank correlations matrices,
after which there was an exhaustive exploration of regression models taking into account the *a priori*
hypothesis outlined in the Introduction. As a check to ensure that we had not overlooked any of the
measured variables as important potential determinants of [C] regression models, we also then checked for
the minimum Akaike Information Criterion regression models using the dredge function available within
MuMIn (Bartoń, 2013). Principal coordinates of soil mineralogical compositions were undertaken using the
princomp function after first transforming the data using the acomp function available within the





compositions package (van den Boogaart and Tolosana-Delgado, 2008). Kruskall-Wallis multiple
comparison tests (Siegel and Castellan Jr., 1998) were undertaken using the kruskalmc command available
within the pgirmess package (Giraudoux, 2013).

### 3 Results

#### 3.1 Clustering of soils types

Figure 1 shows the distribution of the sampled sites across the Amazon Basin, with the soils sampled
divided *a priori* into three "clusters" based on a previous analysis of a subset of sites presented here (Fyllas
et al., 2009; Quesada et al., 2010). This has been done according to the World Resource Base Reference
Soil Group (RSG) classification (WRB, 2014) *viz.* with one group being the typically more strongly
weathered Acrisol and Ferralsol soil types dominated by low activity clays (LAC); the second being other
less weathered soils types (here encompassing the Alisol, Cambisol Fluvisol, Gleysol, Leptosol, Lixisol,
Luvisol, Plinthosol, Regosol and Umbrisol soil groups), typically dominated by high activity clays (HAC)
and with a third group *viz.* exceptionally sandy soils (Arenosols and Podzols), the so called "Arenic" soil
types also being differentiated. From Fig. 1 the majority of the LAC soils sampled come from the eastern
area of the basin and with the majority of the HAC soils found closer to the Andes Cordillera. Arenic soils
are less abundant than either LAC or HAC soils, and were sampled in both the eastern and western portions
of the basin.

The contrasting chemistry of the three soil groups is shown in Fig. 2, where soil effective cation
exchange capacity, $I_E$, is plotted as a function of soil clay fraction, $\Phi_{clay}$ (0 to 0.3 m depth) with different
symbols for each RSG and with the contrasting $I_E$; $\Phi_{clay}$ domains indicated by different background colours.
This shows a minimal overlap between the Arenic and LAC/HAC soil types and with some of the former
having relatively high $I_E$ despite their very low clay content. There is some overlap between the LAC and
HAC soil clusters at intermediate $I_E$ and/or $\Phi_{clay}$, though with it also being clear that none of the sampled
LAC soils were characterised by a high $I_E$ and that none of the HAC soils had a very high or very low clay
content.

#### 3.2 Mineralogical analysis

Distinctions between the LAC and HAC clusters are further illustrated in Fig. 3, where for a subset of the
main dataset, mineralogical analysis of the bulk soil had been undertaken using X-ray Diffraction
Spectroscopy (XRD) and for which the results of a Principal Components Analysis (PCA) ordination are
shown in Fig. 3a. Here it can be seen that the first PCA axis (PCA1) serves to primarily differentiate the
soils according to their clay activity with the 1:1 clay minerals gibbsite, goethite and kaolinite all with large
negative weightings on the PCA1 axis and with the 2:1 potassium feldspar, plagioclase, smectite-illite and





chlorite minerals all with positive weightings. Accordingly (although mineralogy is not used in the RSG
(reference soil groups) classification system), almost all sites within our RSG based LAC cluster are
located with negative scores along the PCA1 axis and with almost all HAC soils with positive values. All
four Arenic soils subject to XRD had high PCA scores.
The contrast between the three soil groups is further shown in Fig. 3b where, shown as a
compositional plot, the contrasting relationships between the 1:1 and 2:1 minerals are considered along
with variations in quartz content. This diagram emphasises the almost total lack of 2:1 minerals found with
the LAC soil cluster, with these soils essentially being of a mixture of 1:1 minerals (primarily kaolinite: see
Table 1) and quartz in varying proportions. On the other hand, the HAC soils are all characterised by a high
quartz content and with less than 20% 1:1 minerals present: although of note, two Cambisols, one Regosol
and one Gleysol had 2:1 minerals constituting less than 1% in their fine earth fraction. Not unexpectedly,
having a quartz content of > 97%, all four Arenic soils are found clustered in the bottom right-hand corner
of the compositional triangle.

### 3.3 Univariate and bivariate comparisons

Using data averaged over the upper 0.3 m of the sampled soil profiles, Figure 4 shows as boxplots the
contrasts between our three *a priori* soil groups in terms of their carbon density [C]; total reserve bases $\Sigma_{RB}$,
effective cation exchange capacity $I_E$, fractional sand, silt and clay contents ($\Phi_{sand}$, $\Phi_{silt}$ and $\Phi_{clay}$) and
concentrations of dithionite and oxalate extractable aluminium and iron *viz.* $[Al]_d$, $[Al]_o$, $[Fe]_d$ and $[Fe]_o$
(Original data available in Table 1 and Appendix Table A1). This shows that, although there was no
significant difference between the three clusters in [C] (Fig. 4a; Kruskal-Wallis test; $p > 0.05$), there were
significant differences in the underlying chemistry at $p < 0.05$ not only between the Arenic soil cluster and
both the LAC and HAC clusters for $\Sigma_{RB}$ (Fig. 4b) $I_E$, (Fig. 4c), $[Al]_d$ (Fig. 4d), $[Al]_o$ (Fig. 4e), $[Fe]_d$ (Fig. 4f)
and $[Fe]_o$ (Fig 4g) but also with HAC soils having higher $\Sigma_{RB}$, $I_E$ , $[Fe]_d$ and $[Fe]_o$ than the soils in the LAC
cluster ($p < 0.05$). For pH, the situation was more complicated, but with the HAC soils having higher
values than the LAC soils ($p < 0.05$) but, with no difference between the Arenic soils and either the LAC or
HAC soils. Despite there being many differences in location at $p < 0.05$ or better as detected through the
non-parametric Kruskal-Wallis test, for all seven soil chemical properties presented in Fig. 4, overlap
between the LAC and HAC soils was in most cases considerable.
In terms of soil texture, as would reasonably be expected, $\Phi_{sand}$ was significantly higher at $p <$
0.05 for the Arenic versus LAC and/or HAC clusters (Fig. 4i) which was also reflected in significantly
lower $\Phi_{clay}$ for the Arenic soils ($p > 0.5$ Fig. 4j). On the other hand, there was no difference between $\Phi_{silt}$
for the Arenic *vs.* LAC soils, both of which, in turn, had a significantly lower $\Phi_{silt}$ than the soils of the HAC
cluster ($p < 0.05$; Fig. 4k). As is also evident from Fig. 2, there was much more variation in $\Phi_{clay}$ for the
LAC soils as opposed to the HAC soils.





Using Kendall's $\tau$ as a non-parametric measure of association, correlations between a wide range
of soil and climate properties potentially involved in differences in soil carbon storage are shown in Table
3, which takes the form of four one-sided correlation matrices *viz.* one half-triangle for each of the Arenic,
LAC and HAC clusters as well as for the (combined) dataset as a whole. Here, with $n > 30$ for the LAC and
HAC clusters we have indicated in bold all cases where $\tau > 0.30$ for these two groupings (as well as the
combined dataset) with this associating roughly with the probability of Type-II error being less than 0.05.
For the Arenic soil cluster with $n = 13$ the equivalent value is $\tau > 0.52$ and where one or more of the four
groupings has $p > 0.05$, this has been indicated for all four matrices using different colours to help cross-
referencing across the four diagonal matrices
Table 3 shows that, whilst there are many correlations which are significant at $p = 0.05$ to be
found in the dataset, only in a few cases are there significant correlations found for the same bivariate
combinations in two or more of the three soil clusters and/or when the three clusters are considered
together. For example, although there is clear association between soil texture and soil carbon density for
the LAC soils ($\tau = -0.56$ and $\tau = 0.54$ for $\Phi_{sand}$ and $\Phi_{clay}$ respectively), this is not the case for the HAC soils
($\tau = 0.06$ and $\tau = 0.19$) and with the association also being much less clear for the Arenic grouping ($\tau = -$
0.17 and $\tau = -0.24$). Consequently, when all three soil clusters are considered together we find $\tau$ of only -
0.21 and 0.31 for $\Phi_{sand}$ and $\Phi_{clay}$. That is to say, when all soils are considered together there is much weaker
association between soil carbon density and soil texture than when LAC soils are considered on their own.
This is also the case for the relationship between [C] and soil bulk density, $D_b$, for which we find $\tau = -0.47$
for LAC soils but markedly lower values for the HAC and Arenic soils ($\tau = -0.29$ and $\tau = -0.17$
respectively) as well as for the combined dataset ($\tau = -0.33$).
In a similar vein, although a high $I_E$ is clearly associated with a high [C] for LAC soils ($\tau = 0.37$)
and perhaps the Arenic soils as well ($\tau = 0.43$), for the HAC soils we find a $\tau$ of only -0.08 for the [C]; $I_E$
association, and for the dataset as a whole $\tau$ equals only 0.13.
On the other hand (simple physically based bivariate associations such as $T_a$ *vs.* $E_v$ aside) there are
cases where the strength of the bivariate associations seems to be consistent across all three soil groups. For
example, taking the relationship between total phosphorus, $[P]_t$, and mean annual air temperature, $T_a$, shows
$\tau = -0.29$, $\tau = -0.32$ and $\tau = -0.22$ for the LAC, HAC and Arenic soils respectively and with the combined
dataset yielding $\tau = -0.35$; a value higher than any of the individual clusters when considered on its own. A
second example of this is the relationship between dithionite extractable aluminium $[Al]_d$ and $\Phi_{clay}$ for
which we find $\tau = 0.31$ for LAC soils, $\tau = 0.20$ for HAC soils and $\tau = 0.36$ for Arenic soils and with $\tau =$
0.35 for the dataset as a whole. Although, not surprisingly there are many correlations between the
variation oxalate/dithionite extraction metrics for Fe and Al, it was only $[Al]_d$ that, on its own, showed any
marked association with [C] and here only for the LAC soils ($\tau = 0.37$) although we also note that $\tau = 0.29$
for the HAC soils and $\tau = 0.28$ for the dataset as whole.



Also of note are the many cases where there are reasonably high $\tau$ found for both the LAC and
HAC soils, but not for the Arenic ones: for example in the associations between Total Reserve Bases, $\Sigma_B$,
and organic matter CN ratio for which we observe $\tau = -0.44$ for LAC soils and $\tau = -0.56$ for HAC soils, but
with a value of only $\tau = -0.03$ for the soils in the Arenic cluster.

**3.4 Carbon/soil texture associations**
With a high $\tau$ observed for several [C] *vs.* soil texture associations (Section 3.3), the relationship between
soil carbon content and $\Phi_{clay}$ is shown in Fig, 5 with a separate panel used for each of the three soil clusters;
and with each panel having different ranges for both the *x*- and *y*-ordinates. For the LAC soils (Fig 5a)
strong linear relationship exists ($r^2 = 0.57$) and with there being little apparent difference between the
Ferralsol and Acrisol RSGs. But when LAC OLS regression line is repeated again within the Arenic soil
group [C]; $\Phi_{clay}$ association graph of Fig 5b (for which we also note the *x*- axis extends only one tenth that
of Fig 5a and with a y-axis 4-fold larger) it is clear that, not only does soil clay content exert little or any
control over [C] for these sandy soils, but also that many of the Podzols have [C] well in excess of even the
highest clay content LAC soils. With the LAC OLS regression line again repeated for the HAC soils in Fig.
5c it is similarly clear that many of the HAC soils have [C] appreciably higher than is expected on the basis
of the highly significant LAC [C]; $\Phi_{clay}$ relationship: but with no detectable [C]; $\Phi_{clay}$ association when
considered on their own ($r^2 = 0.01$).
The underlying OLS regressions of Figure 5 are outlined in more detail in Table 4 which, as well
as providing a [C]; $\Phi_{clay}$ OLS regression summary for the combined dataset as whole, also examines the
effects of including $\Phi_{silt}$ in the [C]; $\Phi_{clay}$ regression models: this being either as an additional term or as part
of a single ($\Phi_{silt} + \Phi_{clay}$) predictor – the latter, of course, also being equal to $-\Phi_{sand}$. Comparing the equations
for LAC, this analysis shows that the addition of the $\Phi_{silt}$ term to the [C]; $\Phi_{clay}$ regression increases the $r^2$
from 0.57 (Table 4a) to 0.61 (Model b) with a change in Akaike's Information Criterion ($\Delta$AIC) of -3.9 and
with the coefficients for both terms having very similar slopes, *viz* 16.6 ±2.1 g C kg$^{-1}$ clay and 14.4 ±6.2 g
C kg$^{-1}$ silt. For these LAC soils, taking silt and clay together as the one soil texture metric (Table 4c)
resulted in a similar $r^2$ and an intermediate slope of 16.2 ± 1.8 g C kg$^{-1}$(clay + silt).
Despite the strong relationships found for the LAC soils for both $\Phi_{clay}$ and $\Phi_{silt}$ , no such
association was evident for the HAC soils and, of the three models tested, none had a $r^2$ greater than 0.05
(Table 4d-f). For the Arenic soils, the addition of $\Phi_{silt}$ term to a simple [C] *vs.* $\Phi_{clay}$ model led to a $\Delta$AIC of
only -1.7 (compare equations of Table 4g and h), but where a summation term ($\Phi_{clay} + \Phi_{silt}$) was tested as a
single predictor variable this resulted in a marked improvement over and above the [C]; $\Phi_{clay}$ relationship
with a $\Delta$AIC of -3.6 and $r^2$ of 0.31 (Table 4i). Of note, Table 4i shows that the fitted slope for the Arenic
soils was 155 ± 63g C kg$^{-1}$(clay + silt), a value nearly 10 times that found for the LAC soils (Table 4*c*).
When all three soils groupings were considered together there was no significant relationship between [C]



and $\Phi_{clay}$: this being the case either with $\Phi_{clay}$ considered on its own, or when considered in conjunction
with $\Phi_{silt}$, and with all three models tested having $r^2 \le 0.01$ and $p > 0.13$ (Table 4j -l).

**3.5 Soil carbon/mineralogical associations**
As already noted in Section 3.1, of the many strong associations between the aluminium and iron oxide
measured and soil carbon concentration, one of the strongest and the most consistent across the three soil
groups was the [C]; $Al_d$ relationship, and this relationship is shown for all three soil groupings in Fig 6
(log-log scale) with the appropriate regression coefficients shown in Table 5 (models $m$ to $o$). This shows
reasonably strong relationships to be found between [C] and $Al_d$ for both the LAC (Fig. 6; $r^2 = 0.27$ $p <$
0.0001) and HAC soils (Fig. 6c: $r^2 = 0.23$ $p < 0.0001$), but not for the Arenic grouping (Fig. 6b; $r^2 = 0.09$ $p$
$> 0.17$). Here direct comparison with the soil texture models of Table 4 according to the AIC values is
confounded by slightly different datasets for the HAC soils (due to $Al_d$ only having been determined for 77
of the 83 HAC soils) and with the relationships here being log-log as opposed to linear. But nevertheless,
the very different $r^2$ between the two model types: with $r^2 = 0.27$ much lower for the [C]; $Al_d$ relationship
than for any of the [C] *vs.* soil texture models for the LAC soils (for which $r^2 > 0.57$) and with this being
the other way around for the HAC soils ($r^2 = 0.23$ for the [C];$Al_d$ relationship but with none of the soil
texture models having $r^2 > 0.05$) suggests that for the HAC soils that $Al_d$ is a much better predictor of [C]
than soil texture. Withal, simple soil texture metrics were the better predictors for the LAC soils.

With any role of $[Al]_d$ in the modulation of [C] also likely to be dependent on soil pH (see

Introduction) we then probed potential interactions of $[Al]_d$ and pH, at the same time evaluating the
potential role of other measured mineralogical factors by testing a range of multivariate models and
selecting on the basis of AIC: the net result of which is shown in Table 6 (model $q$). This model, which also
involves both pH and $[Fe]_o$ has a $\Delta$AIC of -17.7 as compared to the univariate $[Al]_d$ model of Table 5n
suggesting a drastic improvement through the addition of the two additional terms. But nevertheless, using
data for 41 of the 77 HAC sites for which we had leaf litter lignin content ($\Lambda$) measurements available there
was a clear relationship between the model residuals of Eqn 6q (Fig. 7a) and with this relationship also
being evident (though to a lesser extent) when a simpler model involving just $[Al]_d$ and pH was applied ($r^2$
$= 0.25$, AIC $= 85.1$; Fig. 7b). In both cases residuals increase with increasing $\Lambda$ meaning that at high $\Lambda$ the
models tend to underestimate [C] and *vice versa* at low $\Lambda$.

With this lignin effect being consistent with any pH dependent $[Al]_d$ precipitation reaction

mechanism as originally postulated, we thus probed a possible role of $\Lambda$ as a factor interacting with both
pH and $Al_d$ using the more limited dataset of 41 HAC sites for which the requisite data was available.
Model comparisons are shown in Table 7. Starting first with a simple model of [C] as a function of $[Al]_d$,
$[Fe]_o$ and pH (Table 7t which is the same model as Table 6q but in this case with the reduced 'leaf lignin
only' dataset) shows that indeed, the addition of a $\Lambda$ term results in a marked improvement in the model fit





(Table 7u; $r^2 = 0.46$, $\varDelta$AIC = -3.50) and that, for this reduced dataset at least, the [Fe]$_o$ term then becomes
redundant (Table 7v; $r^2 = 0.47$, $\varDelta$AIC = -2.0).

The goodness of fit of Equation 7v is shown in Figure 8 where the fitted soil carbon densities, $[\hat{C}]$

are plotted as a function of the actual values (log-log scale). This shows Equation 7*v* to provide a
reasonable and unbiased fit across a wide range of [C] for HAC soils, though with two locations (*viz.* POR-
02, a Plinthosol in the west of the basin and RIO-12, a Lixisol on the basin's northern periphery) being
substantially overestimated by the model.

Probing the effect of litter quality on soil C storage further, we examined the relationship of $\Lambda$

with both leaf litter and soil C/N ratios (denoted $\Phi_{CN}^{L}$ and $\Phi_{CN}^{S}$ respectively); this exercise being
undertaken with a view to see if we could find statistically significant relationships between $\Lambda$ and one or
both of $\Phi_{CN}^{L}$ and $\Phi_{CN}^{S}$ to allow incorporation of litter quality surrogate measures into an analysis using the
full HAC soil dataset. As is shown in Figure 9, there were indeed significant log-log relationships between
$\Lambda$ and both $\Phi_{CN}^{L}$ and $\Phi_{CN}^{S}$ for both HAC soils (but not for LAC soils and not between $\Phi_{CN}^{L}$ and $\Phi_{CN}^{S}$ for
HAC soils) and with the HAC $\Lambda$; $\Phi_{CN}^{S}$ giving a better fit ($r^2 = 0.32$, $p < 0.0001$, Figure 9b).

Taking then $\Phi_{CN}^{S}$ as our best available surrogate for litter quality, we then tested the effect of

adding this variable to the original HAC model as given in Table 6q, finding that, not only did this term
provide for a substantial reduction in AIC when added to a model already including pH, [Al]$_d$ and [Fe]$_o$, but
that also, upon the inclusion of the $\Phi_{CN}^{S}$ term that the negative [Fe]$_o$ term became, as for the lignin models
of Table 7, redundant (Table 6s).

The goodness of fit of the equation of Table 6s is shown in Figure 10 where the fitted soil carbon

densities $[\hat{C}]$ are plotted as a function of the actual values (log-log scale). This shows Equation 6s to
provide a reasonable and unbiased fit across a wide range of [C] for HAC soils, though with the same two
locations as were overestimated by the lignin model (Figure 9) similarly overestimated.

**3.6 Alternative models**
Although we have used AIC to assist with model selection in Sections 3.3, 3.4 and 3.5, our choice of
models to be tested has for all three soil types been guided by the background knowledge and hypothesis as
outlined in Section 1. It is therefore worth pointing out that if one takes a simple information criterion-
guided model selection approach then it is possible to find models with a lower AIC than those presented in
Tables 4 and 6. For example, for LAC soils there is a model involving all of $\Phi_{sand}$, $\Phi_{clay}$, [Al]$_d$, [Al]$_o$ [Fe]$_d$,



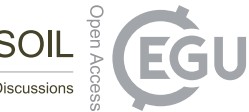

$[Fe]_{do}$ and $\Phi_{CN}^{S}$ which provides a significantly better fit than Equation b of Table 4 ($\Delta$AIC of -19.9). But
for this model many of the terms had $VIF > 10$ and after removal of these terms then the simpler [C] =
$\Phi_{sand}$, + $\Phi_{clay}$ equation is only 0.2 AIC units higher.

Likewise, if one applies a 'blind' information criterion selection criterion to the HAC soils then it

is possible to find a log-log model significantly better to that of Table 6c which retains the $[Al]_d$, term but
with log $\Sigma_{RB}$ substituting pH and, moreover, with an additional $\Phi_{clay}$ term included ($r^2 = 0.65$; $p < 0.0001$;
$\Delta$AIC= -20.5). Further, modifying this 'blindly selected' equation, by reinserting our previously
rationalised pH term in preference to log $\Sigma_{RB}$ term (thus effectively adding a $\Phi_{clay}$ term to the Equation of
Table 6v) results in a markedly inferior fit ($\Delta$AIC = +10.3). Nevertheless, the resulting equation, *viz* [C] =
pH + log $[Al]_d$ + log( $\Phi_{CN}^{S}$ ) + $\Phi_{clay}$, ($r^2 = 0.63$) is still a marked improvement on the equation of Table 7v
($\Delta$AIC= -10.2).

For the smaller Arenic soils dataset ($n = 10$) the lowest AIC linear model is as in Table 4h (i.e.

with, combined together, clay and silt only, $r^2 = 0.31$, $p = 0.035$). Although we do note that there does exist
a virtually uninterpretable log-log model found through the AIC minimisation procedure which involves all
of pH (negative coefficient), $\Phi_{sand}$ $[Al]_d$, $[Fe]_d$ and $\Phi_{CN}^{S}$ (positive coefficients) with an impressive sounding
$r^2 = 0.85$ (but due to the low degrees of freedom for which $p$ is only $< 0.039$).

### 509    3.7 Checking for model biases

In order to check if there were any systematic biases in the final models used (*viz.* the models as presented
in Table 4b for LAC soils, Table 4i for Arenic soils and Table *6s* for HAC soils) standardised model
residuals were examined in relationship to the soil variables $\Phi_{sand}$, $\Phi_{clay}$, $\Phi_{silt}$, $[Al]_d$, $[Al]_o$ $[Fe]_d$, pH and CN
ratio as well as the mean annual temperature $T_A$ and mean annual precipitation $P_A$ climate variables and
two vegetation-associated characteristics available for over 100 of the study sites *viz.* the above ground
wood productivity and above ground biomass: this data being essentially as in Quesada et al. (2012) but in
an updated and expanded form (O. L. Phillips and M. J. Sullivan, personal communication). These
relationships shown in the Appendix Figure A1 which shows that there was little if any evidence of
systematic model bias with the strongest association found for the standardized residuals being with $P_A$ ($\tau =$
0.09 $p = 0.18$).

### 521    3.8 SOC fractions and mineralogy

Further adding to our analysis, Table 8 shows results for soil carbon fractions for a subset of our study sites
($n = 30$). The [C] range in this reduced dataset is similar to the main dataset, with LAC soils ranging from





8.8 to 25.3 mg g$^{-1}$, with Arenic group ranging from 4.2 to 108.6 mg g$^{-1}$, and with the HAC soils ranging
from 5.5 to 24.8 mg g$^{-1}$. It also shows very similar relationships between the relevant edaphic parameters
and [C] as found for the larger dataset and described in section 3.2. Comparing the Kendall $\tau$ from Table 8
with results from Table 3, we find very similar correlations for both LAC and for all groups combined, but
with [C] in the reduced dataset having stronger correlations with clay content and $Al_d$ in LAC soils ($\tau$ =
0.64; $p<0.01$ and $\tau$ = 0.61; $p<0.01$, respectively). The main difference between datasets occurs in HAC
soils, where the reduced dataset used for fractionations shows stronger correlations between [C] and both
clay content and $I_E$ ($\tau$ = 0.49; $p< 0.02$ and $\tau$ = 0.72; $p < 0.001$, respectively) than is the case in the larger
dataset (Table 3).
Soil C fractionations revealed fundamental differences between the three soil groups as shown in
detail in Fig. 11. LAC soils (Fig. 11a) had on average 0.49 of its C in clay rich aggregates (Sand and
Aggregates fraction, S+A), with this increasing with [C] up to 0.74. This increase in S+A fraction in high
[C] soils seems to occur at the expense of the labile clay and silt fraction (C+S) which represents 0.20 of
soil carbon on average, but only 0.09 in the higher [C] soils. The proportion of C in POM and DOC
fractions varied little across the range of soil [C], while the resistant carbon associated to clay and silt
($R_{C+S}$) averaged of $0.2 \pm 0.07$ and showed no clear pattern,
On the other hand, the Arenic group have most of their carbon associated to POM and S+A
fractions (average proportion of 0.47 and 0.25, respectively) (Fig. 11b, Table 8), with the proportion of
POM reaching 0.70 in soils with higher overall [C]. Seasonally wet sands (denoted with $^F$ following the soil
type in Table 1) had the highest POM fractions, averaging 0.6 of total [C], but despite the differences in [C]
related to soil drainage, POM and S+A fraction were still the main stores of SOC in well drained sands
(0.33 and 0.3 of total [C], respectively).
On the other hand, HAC soils had consistently most of their [C] associated to the clay and silt
fraction (0.43) and the resistant carbon (0.28) associated to clay and silt ($R_{C+S}$). On average 0.72 of [C] was
found in these two fine earth fractions (Fig. 11c). The S+A fractions only had on average 0.13 of HAC soils
[C], while POM and DOC had 0.13 and 0.01 respectively. In general, the HAC fractions varied little in
proportion with increasing [C].
Soil C fractions in the three groups also differed in the way they relate to other edaphic properties
such as texture, the abundance of Fe and Al oxides, and bulk soil mineralogy (Table 8). In LAC, soil
carbon associated to both C+S and $R_{C+S}$ fractions did not show any significant correlation with Fe and Al
oxides, nor with clay content, but with C+S being correlated with soil silt content (Kendall $\tau$ = 0.45
$p<0.025$). On the other hand, the S+A fraction, the main pool of SOC, was significantly correlated to clay
content ($\tau$ = 0.55; $p<0.01$). S+A was also negatively correlated with our PCA axis 1 which indicates a
positive relationship with the abundance of 1:1 clay minerals (see Section 3.2) as axis 1 (Ч$_1$ Table 8)
represents to a large degree the abundance of kaolinite, Goethite and Gibbsite (Kendall $\tau$ = -0.39 $p<0.05$).
S+A was also negatively correlated to sand content (Kendall $\tau$ = -0.52 $p<0.01$), S+A was also significantly



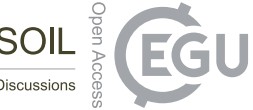

correlated to Fe oxides (Kendall $\tau$ = 0.44; $p < 0.03$ and 0.39 $p < 0.05$ for $Fe_d$ and $Fe_{d\text{-}o}$, respectively). The
DOC fraction was significantly correlated to clay (Kendall $\tau$ = 0.61 p<0.01), $I_E$ (Kendall $\tau$ = 0.48 p<0.02)
and $Al_d$ ($\tau$ = 0.39 p<0.05). DOC was also correlated to Ч$_1$ (Kendall $\tau$ = -0.39 p<0.05). The POM
fraction was significantly correlated to $Fe_{d\text{-}o}$ (Kendall $\tau$ = 0.39 p<0.05).

The small number of Arenic soils in this analysis (n=5) makes correlations unreliable and difficult

to interpret. At $n$ = 5, a Kendall $\tau$ = 0.8 does not differentiate critical values at $p$ = 0.1 and 0.05., and
significance can only be attained for Kendall $\tau$ = 1. Therefore, correlations in Table 8 should be taken just
as a guidance for the direction of the relationship and are not considered further here.

HAC fractions showed totally different correlations to edaphic properties when compared to LAC

soils. For example, he C+S fraction was significantly correlated to clay content ($\tau$ = 0.59 p<0.01), $I_E$ ($\tau$ =
0.62 p<0.01) and with the weathering index TRB ($\tau$ = 0.64 p<0.01). C+S also showed a positive correlation
with PCA axis 1, indicating a positive correlation with the abundance of 2:1 clays ($\tau$ = 0.49 p<0.02). $R_{C+S}$
in HAC soils also showed an effect of both $Fe_d$ and $Al_d$ (Kendall $\tau$ = 0.62 p<0.01 and 0.41, p<0.04,
respectively) and $I_E$ (Kendall $\tau$ = 0.44 p<0.03).

In striking difference to LAC, S+A in HAC soils was an insignificant storage for SOC and showed

no significant correlation to the concentration of any oxides, clay content or any other of the measured
parameters. DOC on the other hand behaved in a more similar manner to LAC soils, also showing
significant associations with $I_E$ ($\tau$ = 0.60 $p < 0.01$) and clay content ($\tau$ = 0.41 p<0.04) and an iron oxide
effect ($Fe_d$: $\tau$ = 0.49; $p$ <0.02). POM on the other hand was correlated to $Fe_o$ ($\tau$ = 0.51; $p < 0.02$) and $Al_o$ a ($\tau$
= 0.41; $p < 0.05$) and $I_E$ ($\tau$ = 0.49; $p< 0.02$, respectively).

### 3.9 Carbon stocks versus carbon concentrations

Although the analysis here has focused on soil carbon concentrations, for carbon inventory purposes the
actual carbon stock (i.e. carbon per unit ground area; $C_S$) is usually of more interest, and with the two being
related according to
$$C_s = \int_d^0 [C]_z \cdot \rho_z \; dz$$

where $[C]_z$ and $\rho_z$ represents the carbon concentrations and bulk density of the soil at depth $z$ below the
soil surface respectively and $d$ is the maximum sampling depth. Thus with the actual calculations done
layer by layer (*viz.* 0 to 0.05 m, 0.05 to 0.10 m, 0.10 to 0.20 m and 0.20 to 0.30 m) Figure 12 shows (top
panels) the relationship between $[C]$ and $\rho$ for the three soil groups with regressions shown were
significant at $p < 0.05$ or better. This shows a reasonably strong relationship for the LAC soils across the 0
to 0.3 m depth (Fig 12a, $\log(\rho)$ = 0.881 - 0.298× $\log[C]$: $r^2$ = 0.43; $p < 0.001$) and with a similar though



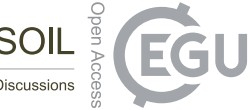

somewhat less convincing relationship being observed for the HAC soils (Fig 12b, $\log(\rho)$ = 0.678 - 0.
219× log[C]: $r^2$ = 0.25; $p <$ 0.001) but no readily discernable relationship evident for the Arenic soils
(Fig. 12c, $\log(\rho)$ = 0.697 - 0.233× log[C]: $r^2$ = 0.20; $p <$ 0.08).
These negative [C] *vs.* $\rho$ associations across all three soil groupings necessitate that $C_s$ is a
saturating function of [C] as is shown in the lower panels of Fig 12 with the slopes of the log-log scaling
relationships being 0.62 ± 0.05 for LAC soils (Fig. 12d) , 0.71 ± 0.05 for the HAC soils (Fig. 12e), 0.23±
0.15 for the Arenic soils (Fig 12f) and 0.59 ± 0.04 for the dataset as a whole. This means, for example, that
– on average – an increase in [C] of 50% will result in only an increase in $C_S$ of $(1.5^{0.59}$ - 1) or just 27%.
This negative covariance between [C] *vs.* $\rho$ also means that within a given soil group variation in
$C_S$ is typically much less than for [C]. For example, as is shown in Table 9, the 12 RSG examined show a
lower coefficient of variation for $C_S$ than is the case for [C] and with this difference being especially
marked for Cambisols (0.63 for [C] *vs.* 0.39 for $C_S$). Also shown in Table 9 are the mean $C_S$ for the 12 RGS
we have examined as compared to the values given by Batjes, 1996) for which we note that in the majority
of cases our estimates are surprisingly close: with one exception being the Alisols for which our estimate of
around 46 t C ha$^{-1}$ is only 53% that of the Batjes (1996) estimate of *ca.* 86 t C ha$^{-1}$ to 0.3 m depth. Our
Leptosols and Podzol $Cs$ estimates are also much higher than those of Batjes (1996).

## 4 Discussion

According to our analysis, the three soil groups studied here are characterised by different soil C
stabilisation mechanisms. Specifically, highly weathered soils, dominated by low activity clays such as
Ferralsols and Acrisols (our LAC group) have SOC densities that are strongly dependent on their clay and
silt contents. However, such simple relationships with soil fine earth fraction could not explain SOC
variations in for less weathered soils with SOC stabilization was predominantly related to interactions with
Al, and the formation of Al/organic matter coprecipitates for HAC grouping. For our Arenic soils group, it
appears that most of the SOC present is in loose particulate organic matter form, and therefore not
stabilized by mineral interactions, though with a surprisingly strong effect of their small clay and silt
content variations.
Such differences in the stabilization mechanisms can be considered to arise from the different soils
examined being at contrasting pedogenetic development stages and/or differences in parent material.
Highly weathered soils such our LAC group have been under constant tropical weathering rates for
timescales that range from 100 million to 2 billion years (Hoorn et al., 2010; Quesada et al., 2011), with
some of the central and eastern Amazon Basin soils having suffered several cycles of weathering (Herrera
et al., 1978; Irion, 1978; Quesada and Lloyd, 2016). This extreme weathering of LAC soils has resulted in
a deep uniformisation of their mineralogy, which is dominated by kaolinite (Sombroek, 1984), and in the
depletion of rock derived elements. It has also resulted in the development of favorable soil physical




properties such as free drainage, low bulk densities and the formation of very deep soil horizons (Quesada
et al. 2010).
Nevertheless, it also needs to remembered that the Amazon Basin has a complex mosaic of soils,
with *ca.* 40% having young and intermediate pedogenetic development levels (Quesada et al., 2011;
Richter and Babbar, 1991; Sanchez, 1976). Most of these less weathered soils occur in the west of the
Basin and were influenced by the uprising of the Andean Cordillera (Hoorn et al., 2010) and thus having
much younger geological ages. Much of the soil formation process in this region only came into effect after
the Pliocene, with most of the soils in that region having less than 2 million years (Hoorn and Wesselingh,
2011; Quesada et al., 2011; Quesada and Lloyd, 2016). Soils in that region have a diverse mineralogy, with
high abundance of 2:1 clays and sometimes also some easily weatherable minerals and relatively high
levels of rock derived (Irion, 1978; Quesada et al., 2010, 2011; Sombroek, 1966, this study). One important
characteristic of many  HAC soils is the very high amount of Al that is released through the weathering of
2:1 clays (Marques et al., 2002). High active clays are unstable in environments depleted of silica, alkaline
and alkaline earth cations, thus releasing soluble aluminium from the octahedral internal layers of the 2:1
clay minerals, and with such Al release also increasing with depth (Quesada et al. 2011).
The Arenic soil group on the other hand is strongly influenced by its parent material. It comprises
the Arenosol and Podzol reference groups, with the latter also being predominantly sandy in Amazonia (Do
Nascimento et al., 2004). Both soil types are thought to have evolved from the weathering of aeolian and
riverine sediments of siliceous rocks, or in some cases, being locally weathered and deposited in colluvial
zones through selective erosion (Buol et al., 2011; Driessen et al., 2000). As quartz usually makes up more
than 90% of their mineral fraction, their surface exchange capacity is very small, resulting in very low
nutrient levels as a consequence of a high degree of leaching (Buol et al., 2011; Quesada et al., 2010;
2011). The very low nutrient content of these soils, often associated with high groundwater levels, results in
the formation of thick root mats in the soil surface (Herrera et al. 1978) which then strongly influences the
amount and vertical distribution of their SOC stocks.
Therefore, our HAC, LAC and Arenic soils groups consist in very different soils, with contrasting
geological formation and chemical and physical properties. Not surprisingly, such wide variations also
resulted in different mechanisms of SOC stabilization.

**4.1 Mechanisms of SOC stabilization**
**4.1.1 SOC stabilization in low activity clays**
Since soil C content might reasonably be expected to depend, at least in part,  on specific surface area
(SSA) because a higher density of exchange sites per unit volume should result in more soil carbon
stabilization through mineral-organic matter associations (Saidy et al. 2012), the uniform mineralogy of 1:1





soils means that, as is shown in Figure 5 and elsewhere (Burke et al., 1989; Dick et al., 2005; Feller and
Beare, 1997; Telles et al., 2003), that for LAC soil organic C scales linearly with clay content since, at the
variation in clay content is the main source of variation in SSA.
The observed variation in clay content across LAC soils studied here was large, from 0.05 to 0.89.
This reflects differences in parent material, with Acrisols tending to have sandier top soils (West et al.,
1997). Central and East Amazonia are known for having very clay rich soils, often having clay content well
above 60% (Chauvel et al., 1987; Sombroek, 1966) with such clays originating from ancient fluvio-
lacustrine sediments deposited on the Barreiras and Alter do Chão geological formations locally known as
Belterra clays (Sioli, 1984; Sombroek, 1966, 2000). Other regions where Ferralsols dominate, such as the
southern fringe of the Basin (Quesada et al. 2010), often have much sandier soils.
The uniformity in the clay;C relationships shown by our best OLS models indicate an overruling
effect of clay content and with some effect from silt (Table 4). The superior predictive power of sand
content (–[clay+silt]), compared to clay as a main determinant of SOC in highly weathered tropical soils
has already been shown by Saiz et al. (2012), with these authors concluding that sand content shows less
confounding effects than that of clay in these systems. The association of clay with aluminum and iron
oxides in highly weathered tropical soils may promote the formation of sesquioxides. Saiz et al. (2012)
have shown that these particles confer the soil a coarse-like texture, which exerts a strong influence on soil
bulk density and water retention properties. Furthermore, results from Figure 3a,c also suggest a wide
variation of Fe oxides to occur on LAC soils and with Figure 6 and Tables 3 and 5 indicating that the
abundance of $Al_d$ is also correlated with SOC. This could be related to increments in SSA resulting from
the greater abundance of such minerals (Eusterhues et al. 2005, Kleber et al. 2005, Wiseman and Püttmann
2006, Saidy et al. 2012) in which an increment in the number of exchange sites may provide additional
stabilization of carbon via direct complexation (Parfitt et al., 1997; Schwertmann et al., 2005) and with
direct interactions between SOC, Fe and Al oxides, and clay particles (Wiseman and Püttmann 2006) also
being important. However, Fe and Al hydroxides may also indirectly protect carbon from decomposition
through their role in the formation of stable aggregates which make carbon physically inaccessible to
decomposers (Kitagawa 1983, Six et al., 2004; Wagai and Mayer 2007). This may be of importance for
LAC soils since stable clay aggregates were found to store most of SOC (Section 3.5).
Using soil carbon fractionations to gain further insights on the stabilization mechanisms that
underlie soil organic matter dynamics (Denef et al., 2010), Fig. 11a shows that the sand and aggregate (S +
A) fraction is responsible for holding most of SOC in LAC soils. This fraction is essentially formed by a
mixture of clay, silt, oxides and organic matter, and within this fraction aggregation may promote increased
SOC protection as it influences the accessibility of substrate to microorganisms, thus limiting the extent
that the diffusion of reactants and products from extracellular synthesis (i.e. soil enzymes) can reach the
organic matter (Sollins et al., 1996). For example, pore spaces inside aggregates can be too small to allow
access of bacteria (Van Veen and Kuikman, 1990) and efficient enzyme diffusion (Sollins et al. 1996). This



then retains SOC in inaccessible micropores inside aggregates (Baldock and Skjemstad, 2000) which
ultimately protects SOC from decay, explaining the positive correlation often found between the level of
soil aggregation and SOC concentration (Six et al., 2004; Tisdall and Oades, 1982).

Soil aggregation level is also affected by other chemical, microbial, plant, animal and physical
processes, many of which seem to be favoured by the tropical climate and thriving biological activity of the
tropical moist forest environment. For instance, microbial activity releases polysaccharides that act as
binding agents in soil aggregates (Lynch and Bragg, 1985; Oades, 1993) and fungal hyphae are known to
bind solid particles together (Sollins et al. 1996). Plant roots also influence soil aggregation by releasing
exudates that can directly flocculate colloids and bind or stabilize aggregates (Glinski, 2018). Root
exudates may also foster microbial activity which can lead to aggregate formation and stabilization. Plant
roots and associated hyphae can also enmesh soil particles by acting as a "sticky string bag" (Oades, 1993)
which binds soil particles. Also, the pressure exerted by roots and soil fauna on soil also promotes
aggregation (Oades 1993; Sollins et al. 1996). Soil fauna (including earthworms, termites, collembola,
beetles, isopods and milipeds) form fecal pellets and excrete binding agents that form aggregates (Oades
1993; Sollins et al. 1996). Nevertheless, the presence of Fe and Al oxides in these soils may also favour the
formation of soil aggregates (Kitagawa 1983, Wagai and Mayer 2007) since they act as binding agents with
clays in a process thought to be associated to the large abundance of aggregates in Ferralsols and Acrisols
(Paul et al., 2008; Sanchez, 1976; Sollins et al., 1996).

Soil C stabilization in the surface of Amazonian Ferralsols and Acrisols (1:1 clays) is thus
interpreted here as the summation of the effect of variations in kaolinite clay content (varying SSA) and the
additional physical protection given by the extensive level of aggregation common to these soils.

### 4.1.2 Processes of C retention in sandy soils

Since quartz is devoid of significant surface area and exchange sites, the retention of SOC in sand rich soils
is difficult to predict on the basis of soil physiochemical properties as there is no, or very little, mineral-
organic matter interaction. Thus, the bulk SOC variation in our Arenic soil group most likely reflects
varying edapho-environmental conditions such as groundwater levels and/or moisture regimes, vertical root
distribution and/or litter quality. However, small changes in clay and silt content were still found to have
large effects on soil [C] (Table 4), with this OLS regression giving a slope ten-fold greater than that of
LAC soils. This is similar to what Hartemink and Huting (2008) found for 150 Arenosols in Southern
Africa, where soil carbon content varied from about 0.5 to 12 g kg$^{-1}$ along a change in clay fraction ranging
from  effectively zero to just 0.12. Similar findings (i.e. 0.8 to 14.5 g kg$^{-1}$) were also obtained on heavily
coarse-textured soils sampled along a 1000 km moisture gradient spanning from Southern Botswana, into
southern Zambia (Bird et al., 2004).



In addition, groundwater fluctuations and the often extremely low nutrient availability of these
soils often result in the formation of root mats, covering the top 10 to 50 cm of the soil surface with an
impressive mixture of roots and organic matter in different stages of decomposition (Herrera et al. 1978).
Such soil mats may reasonably be expected to exert a strong influence on soil SOC concentrations, since
they concentrate the inputs of organic matter into a single layer close to the surface. Moreover, because
many of these soils are seasonally waterlogged (Quesada et al. 2011) the associated anaerobic conditions
should also inhibit decomposition. It is therefore not a surprise then that we observed some of the highest
[C] in these soils.
Our fractionation results again provided additional information for the understanding of SOC
retention with the bulk of the SOC in Arenic soils found as free particulate organic matter, and with this
proportion increasing as [C] increases (Fig. 11b). This was particularly the case for seasonally wet sands
(up to 60% of SOC), but with POM also being a significant fraction of the total SOC even in the drier sands
(~ 30%). The implication here is that chemical recalcitrance of organic matter may also have a role in these
soils: favouring the maintenance of residual, hard to decay organic particles.
The latter are thought to be common due to the extreme dystrophic status of these soils, with total
P levels often as low as 10 mg kg$^{-1}$: and with these being *ca.* 10 fold greater than in LAC soils and
generally 20-50 times greater in HAC soils (see Quesada et al. 2010 for further details). Such a low level of
nutrient content often results in high levels of plant investment in secondary defense compounds against
herbivory (Coley et al., 1985; Fine et al., 2004) and such chemical recalcitrance may affect the
decomposition process and thus slight increase residence time of uncomplexed C in the soil. This may
affect POM levels particularly, considering that the most recalcitrant part will be left undecomposed
following microbial attack. This is given support by the observations made by Luizão and Schubart (1987),
who found that leaf litter decomposition in Amazonian white sands takes twice as long than for  Ferralsols
and Acrisols during the dry season nearly seven times longer in the wet season when decomposition is
more dynamic in the non-white sand soils. Organic acids from residual decomposition from these soils are
known to colour the rivers of the region, with the Rio Negro  with its head waters within a vast white sand
forest region (Quesada et al. 2011) getting its name by virtue of its high humic and fulvic acid content
(Fittkau, 1971).

### 4.1.3 SOC stabilization in less weathered soils

Our results suggest that Al/organic matter (Al/OM) interaction, or coprecipitation is a fundamental
mechanism of SOC stabilization for the less weathered HAC forest soils of the Amazon Basin with the
OLS models presented here involving complex interactions between Al species (Al$_d$), soil pH and the
abundance of aromatic, carboxyl-rich organic matter. The complexity of the models and their high ability to





explain SOC densities suggest that this mechanism is fundamental to an understanding of HAC soil C
storage.

To our knowledge this is the first time that Al/OM interactions have been suggested as  a key
factor explaining SOC densities in the Amazon forest soils. Nevertheless, with DOC being ubiquitously
present in such a highly dynamic system, and with exchangeable Al often abundant  as has already been
shown to be the case in western Amazon soils (Quesada et al. 2010; 2011, Marques et al. 2002; this study),
it is intuitive that Al/OM interactions should encompass a continuum from low-polymeric metal-organic
complexes to well crystalline phases with surface attached organic matter (Kleber et al., 2015). Thus
Al/OM interactions forming coprecipitates is likely to be a widespread mechanism that has previously been
overlooked  because most of the studies in the Amazon Basin have to date only focused on highly
weathered soils such as Ferralsols and Acrisols (i.e. Telles et al., 2003). Nevertheless, with less weathered
soils occupying circa 40% of the Amazon Basin (Quesada et al. 2011), it is important to further investigate
the role of Al/OM interactions, in particular with regard to their influence over SOC mean residence times
(MRT), since they are likely to be different from what is known for Ferralsols. For example MRT of SOC
in Amazon Ferralsols is about 10 years (Trumbore and Camargo 2009) as determined by [14]C studies, but to
our knowledge, no [14]C information is available for western Amazon soils, nor is such information is
available for MRT of Al/OM co-precipitates. As organic polyelectrolytes reorganize on mineral surfaces
over time they form additional polar covalent bonds and this aging process can then lead to a decreased
desorbability of OM (Kleber et al. 2015) MRT of Al/OM co-precipitates could well extend to decades or
even centuries.

In that respect, it is clear that organic matter becoming  co-precipitated with Al results in it
becoming more resistant to microbial decay (Kalbitz and Kaiser, 2008; Nierop et al., 2002). At Al/OM
concentrations typical of forest soils, up to 80% of DOC can coprecipitate (Nierop et al. 2002; Scheel et al.
2007) and with mineralisation rates of Al/OM coprecipitates formed from DOM much lower than the
compounds from which it originates (Boudot et al., 1989; Scheel et al., 2007). For instance, using
incubations, Scheel et al. (2007) found that the mineralisation extent of Al/OM precipitates ranged from 0.5
to 7.7% while the DOM that originated the precipitates had much higher rates (5 to 49%). Kalbitz and
Kaiser (2008) found that up to 50% of total SOC in their study site was stabilized from DOM following
Al/OM interaction, with the authors suggesting that Al coprecipitation has a stronger capacity to reduce
mineralization than sorption in phyllosilicates.

The formation of Al/OM coprecipitates is influenced by several factors and interacting processes
with, according to the extensive review from Kleber et al. (2015), the most important factors being the
prevalent metal to carbon ratios in the soil solution (M/C), the presence of aromatic organic compounds,
the pH value of soil solution and the metal species present (in which Fe also may have a role). Increasing
M/C ratios increase the probability of reaction with OM while the solution pH controls the solubility and
speciation of metals (Al, Fe). With an increasing pH, the efficiency of the process increases, causing larger



amounts of precipitates (Scheel et al. 2007). Also, co-precipitation occurs preferentially with aromatic,
carboxyl-rich organic structures such as derived from lignin and tannin decomposition due to their higher
affinity for Al complexation sites (Scheel et al. 2007; 2008; Kleber et al. 2015), interactions which were
also made clear through the importance of litter lignin content and soil C/N ratio in our OLS results. With
regard to metal speciation, our OLS models selected for dithionite extractable Al ($Al_d$) which, having a
broad capacity to extract Al bearing minerals, we interpret as a continuum of likely different forms such as
free Al ($Al^{+3}$), Al from Al-interlayer minerals, Al-OM complexes and both crystalline and amorphous Al
hydroxides (particularly at higher pH values).
Further insights may again be found from the fractionations study, with Fig 11c suggesting that for
HAC the Al/OM precipitates are held together within C+S fractions, this being despite there being no
simple correlations with clay fraction in the extended dataset. Although this could perhaps be attributed to
the use of only a subset of sites used in the fractionation analysis, where the reduced dataset shows stronger
associations between [C] and clay content, we suggest that such colloidal sized Al/OM precipitates should
be stored alongside the fine earth fraction. Remarkably 75% of SOC occurs associated to C+S (and its
resistant fraction) in these soils, with this fraction being reasonably consistent across a range of soil [C].

### 4.2 Possible influences of confounding factors

As noted in the Introduction, our approach to modelling the [C] storage potential has here been primarily
hypothesis based, but also as noted in Section 3.6, there were some models that – on the basis of their AIC -
did appear superior to those presented as best models here. For example in modelling the [C] storage of
HAC soils solely on the basis of soil mineralogical properties, then a model including both $Fe_o$ and $Al_o$
seemed to be the best (equation of Table 6q). Nevertheless, following our rationalisation that plant organic
matter quality inputs should also be important, once soil CN ratio data was added to the model, then the
hard to explain apparent negative $Fe_o$ effect became redundant (equations of Table 6r and Table 6s).
Likewise in Section 3.6 we also noted that Total Reserve Basses seemed to be a better predictor than pH in
a model of soil C stocks with $[Al]_d$ and CN ratio as covariates, we chose pH for our final model on the
basis of its known effect of the SOC precipitation process and with the apparent TRB effect rationalized as
a simple consequence of its high correlation with pH in HAC soils ($\tau = 0.52$; $p < 0.0001$: Table 3).
Also, not included in our final models were the effects of either mean annual temperature or
precipitation, for which, as well as showing poor associations with SOM storage for all three of our soil
groups when considered individually as well as when all soils were pooled together as a whole, also
showed no significant association with model residuals (Appendix Figure A1). Nor – as is also shown in
Appendix Figure A1 – was there any suggestions of variations in carbon inputs having any influence on
Amazon forest C stocks. This suggests that, across the temperature and precipitation range of our dataset
that litter input quality and soil mineral stabilization mechanisms are the primary determinants of the SOM

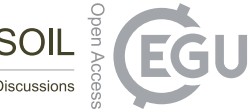

storage variations:  a result which is consistent with microbial decomposition rates acclimating to both
temperature (Bradford et al., 2008) and precipitation (Deng et al., 2012).

That is not to say of course, that our results also mean that any future changes in temperature or

precipitation should inevitably have no effect on the amount of carbon stored in the forests of the Amazon
Basin. For example, Cotrufo et al., (2013) have postulated that although interactions of organic materials
within the soil mineral matrix are the ultimate controllers of SOM stabilization over long timescales, it is
the microbially mediated delivery of organic products to this matrix that provides the critical link between
plant litter inputs and what products are available for stabilization. In this respect a consideration of depths
substantially greater than the upper 0.3 m examined here must also be critical for the accurate
determination of any future changes in climate stocks as below 0.3 m Amazon Basin forest soil C are
generally quite low and with there likely existing reactive mineral surfaces yet to be saturated with SOM
(Quesada, 2008; Quesada et al., 2010). Moreover, any future inputs into these lower layers, including those
mediated though increased litter inputs due to likely ongoing [$CO_2$] induced increases in stand-level
productivities: (Lloyd and Farquhar, 2008), are likely to be microbially derived (Schrumpf et al., 2013).
Quite likely the extent of any such additional stabilization of SOM at these lower depths will differ between
HAC, LAC and Arenic soils in accordance with the different stabilization mechanisms as suggested
throughout this paper. But in the absence of more detailed information and indeed, precise confirmations as
to the apparent different mechanisms involved in SOM storage as suggested here; then whether or not it is
really the case that Amazon forest soil C stocks are currently increasing in response to higher litter inputs
with soil developmental stage also influencing that response must remain a matter of simple conjecture.
**5 Acknowledgements**

This manuscript is a product of the RAINFOR network. It integrates the effort of several

researchers, technicians and field assistants across Amazonia. We thank the following individuals in
particular: Michael Schwarz, Gabriel Batista de Oliveira Borges, Claudia Czimczk, Jens Schmerler,
Alexandre J.B. Santos, Garreth lloyd, Jonas O. Moraes Filho, Orlando F. C. Junior, José Edivaldo Chaves
and Raimundo Nonato de Araújo Filho.  Support for RAINFOR has come from the Natural Environment
Research Council (NERC) Urgency Grants and "TROBIT" (NE/D005590/1) and the Gordon and Betty
Moore Foundation.

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






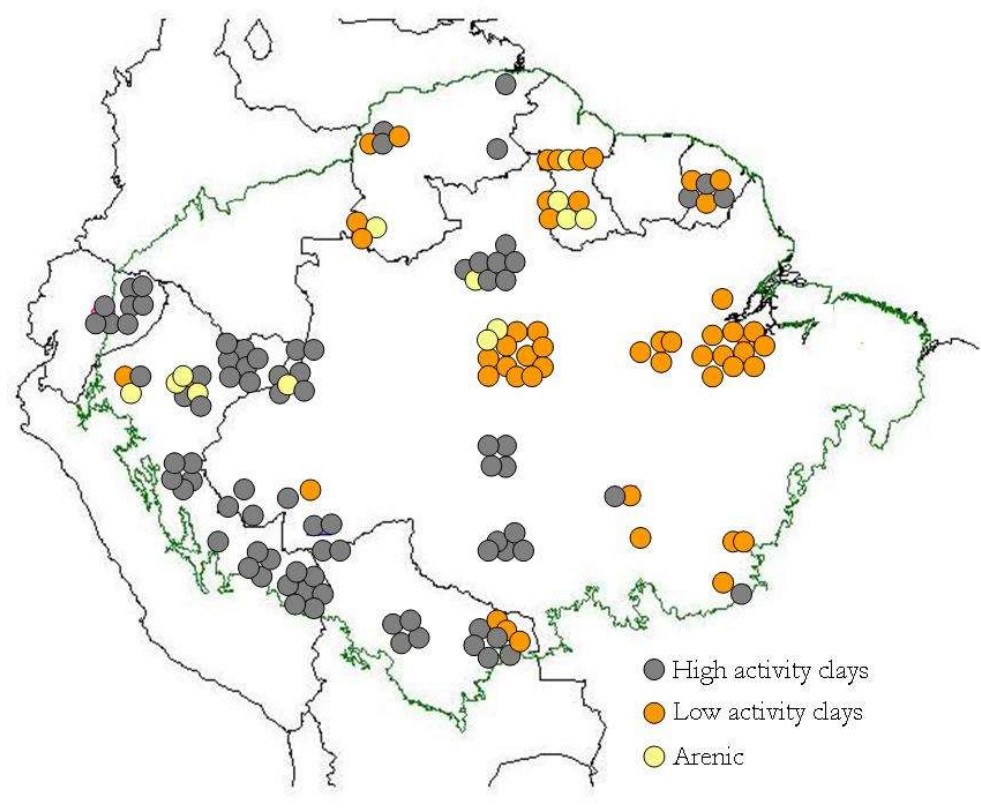



**Fig. 1. Geographic distribution of 147 study sites across the Amazon Basin, according to the
different soil groups. Each point is a 1 ha forest inventory permanent plot. Geographical locations
have been manipulated in the map to allow visualization of site clusters at this scale.**


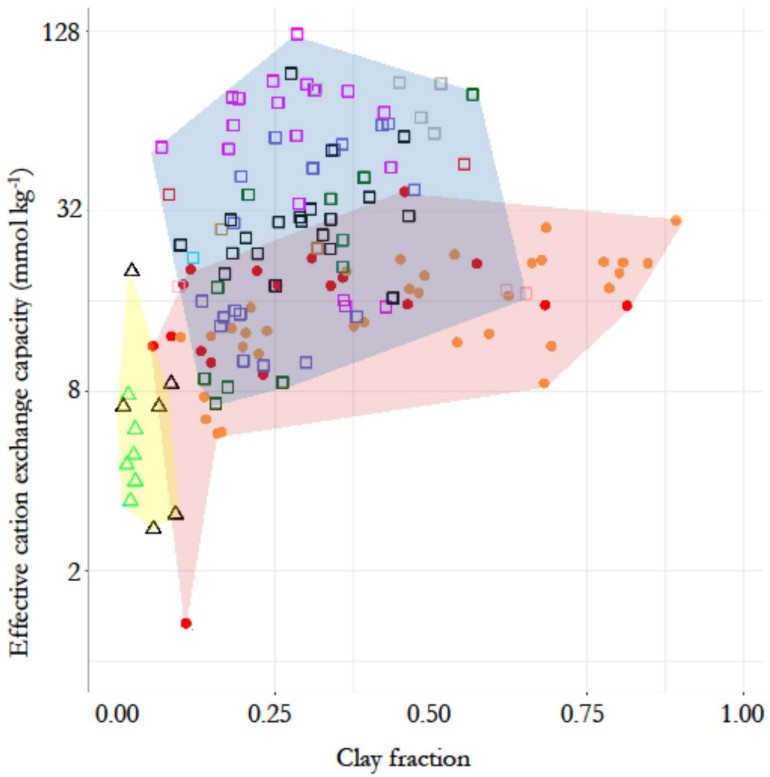


**Fig. 2. Contrasting chemical characteristics of the three soil groups, evidenced by the relationship between top soil clay fraction and effective cation exchange capacity (0-30 cm). Triangles with yellow background represent the Arenic soil group, consisting of Arenosols (green) and Podzols (black). Filled circles with pink background represent the low activity clay soils (LAC) which consists of Ferralsols (yellow) and Acrisols (red). Soils having high activity clay (HAC) are show as open squares with light blue background. They are the Alisol (black), Cambisol (pink), Fluvisol (grey), Gleysol (green), Leptosol (brown), Lixisol (red), Luvisol (purple), Plinthosol (blue), Regosol (cyan) and Umbrisol (light green) soil groups.**

1149



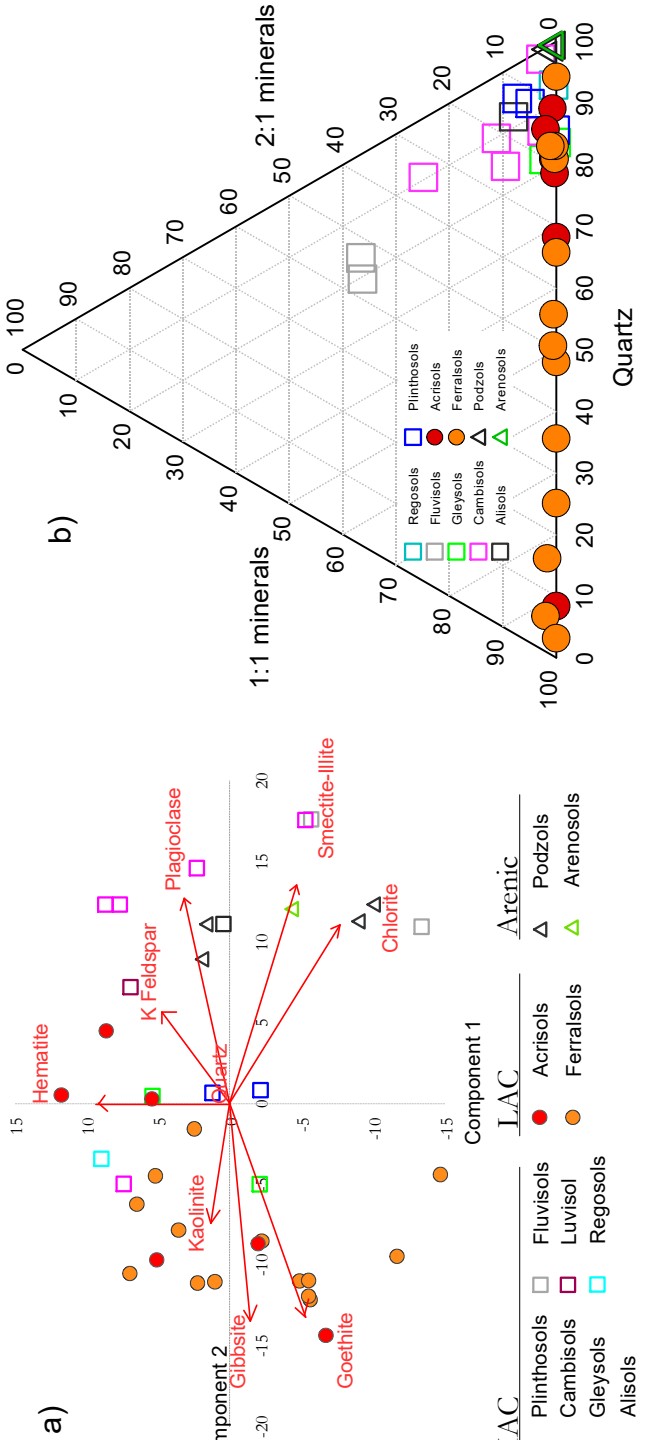

**Fig. 3 Contrasting mineralogical characteristics of the different soils in this study. a) Principal Components Analysis (PCA) ordination on semi-quantitative X-ray Diffraction Spectroscopy (XRD) data. b) Compositional plot showing contrasting relationships between the 1:1 and 2:1 minerals considered along with variations in quartz content.**



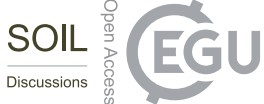

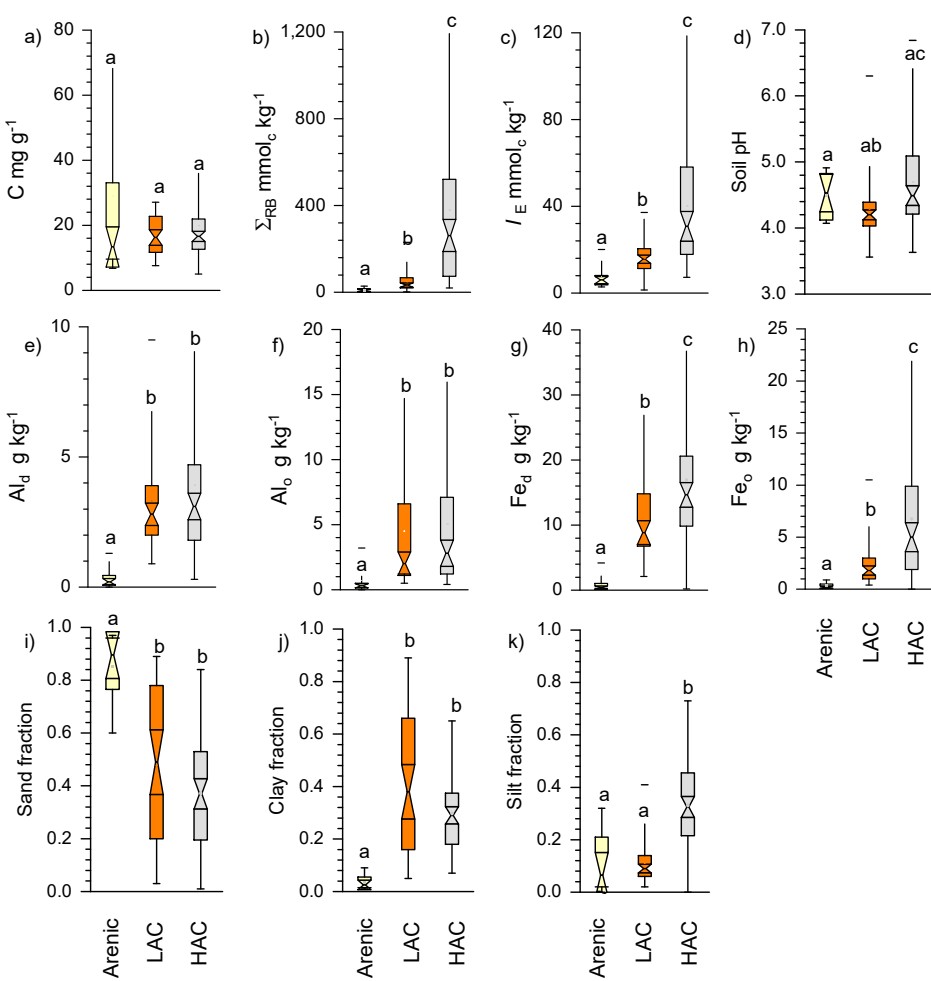

Fig. 4. Contrasts between the three soil clusters for selected variables. Statistical differences are given through the
5  non-parametric Kruskal-Wallis test.



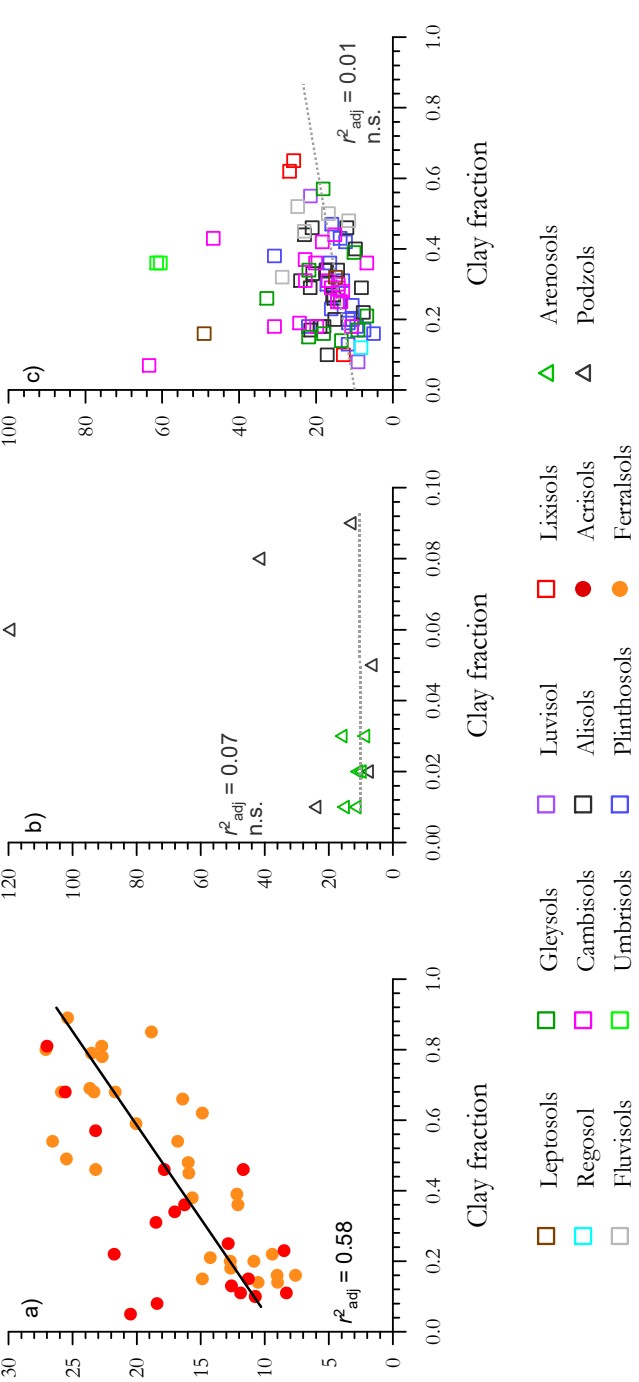

Fig. 5. Associations between soil organic C and clay fraction for the three soil groups. a) low activity clay (LAC), b) arenic and c) soils containing high activity clays (HAC).
5  Only LAC shows a significant regression. Non-significant regressions in arenic and HAC soils are shown as dotted lines.



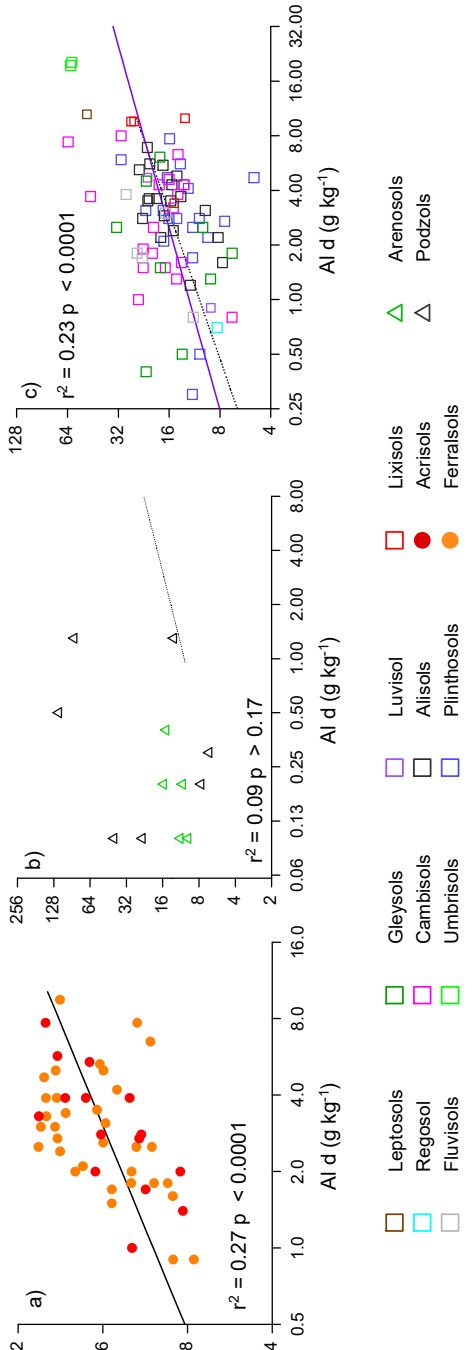

Figure 6. The association between soil organic C and dithionite extractable Al (Al d) for the studied soils. The regression line for LAC soils (Fig. 6a) is repeated as a dotted line in Fig.6b (Arenic) and 6c (HAC) for comparison.



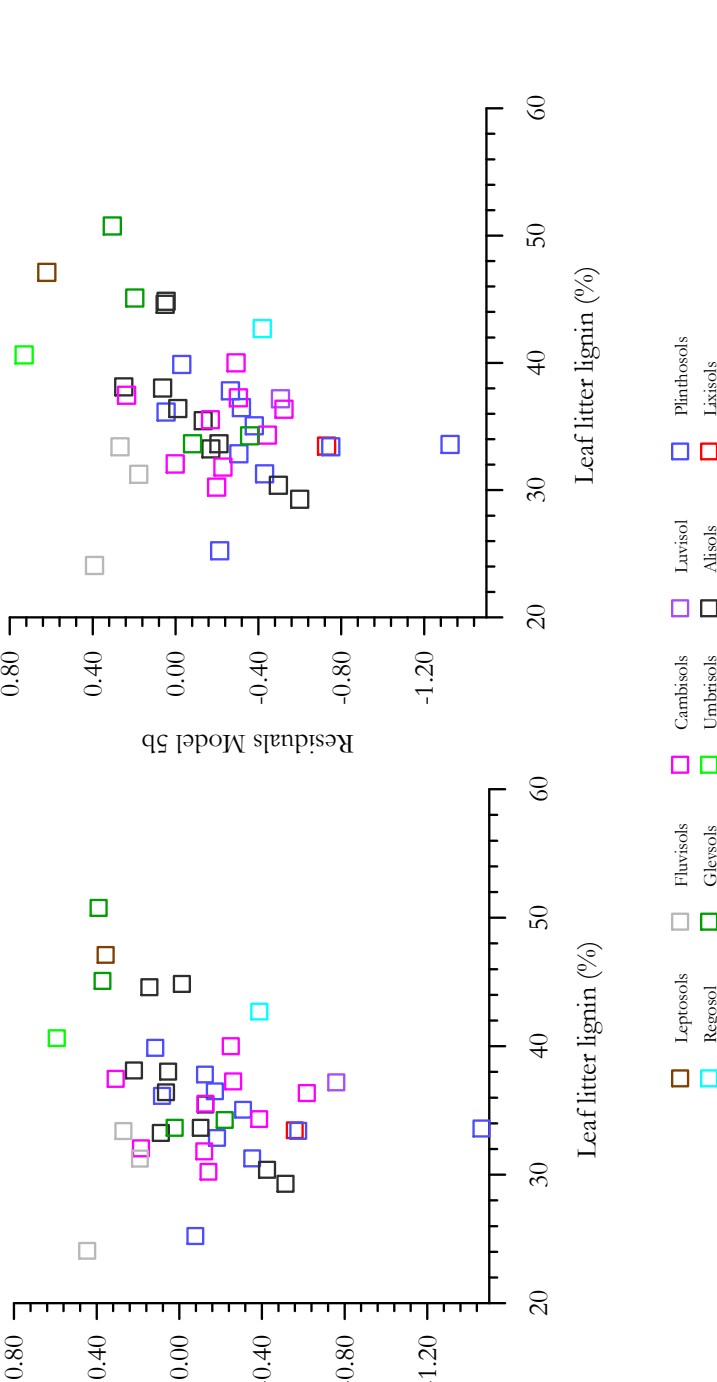

**Fig. 7. The effect of litter lignin content, a surrogate for the abundance of aromatic C compounds, on the residuals of model regressions 6q (Table 6; Fig. 7a) and a simplified additional model with only pH and Al d included (Fig. 7b).**



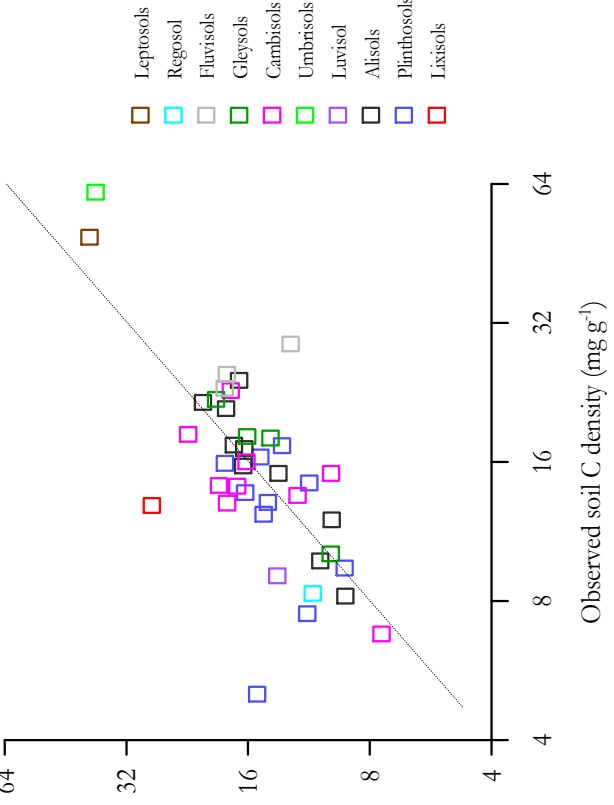

**Fig. 8. Fitted vs observed SOC densities for regression model 7v (Table 7).**



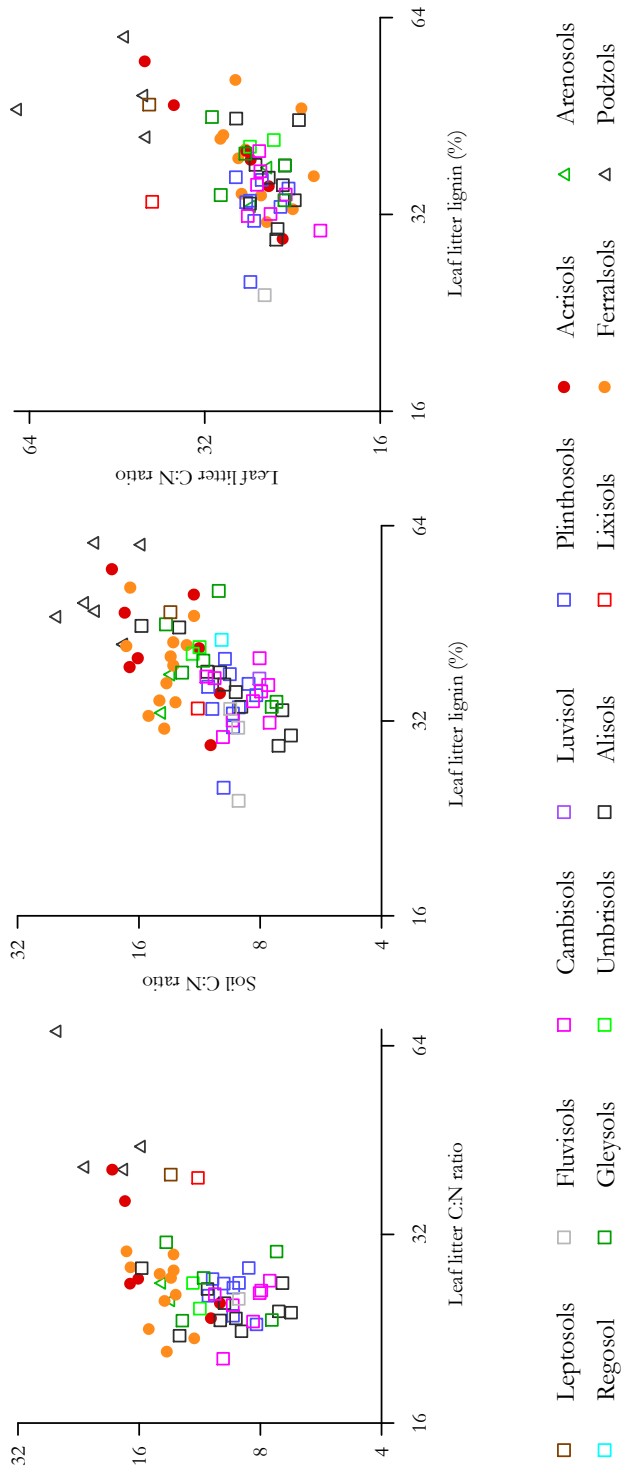

**Fig. 9. The relationship of leaf litter lignin content with both leaf litter and soil C:N ratios.**



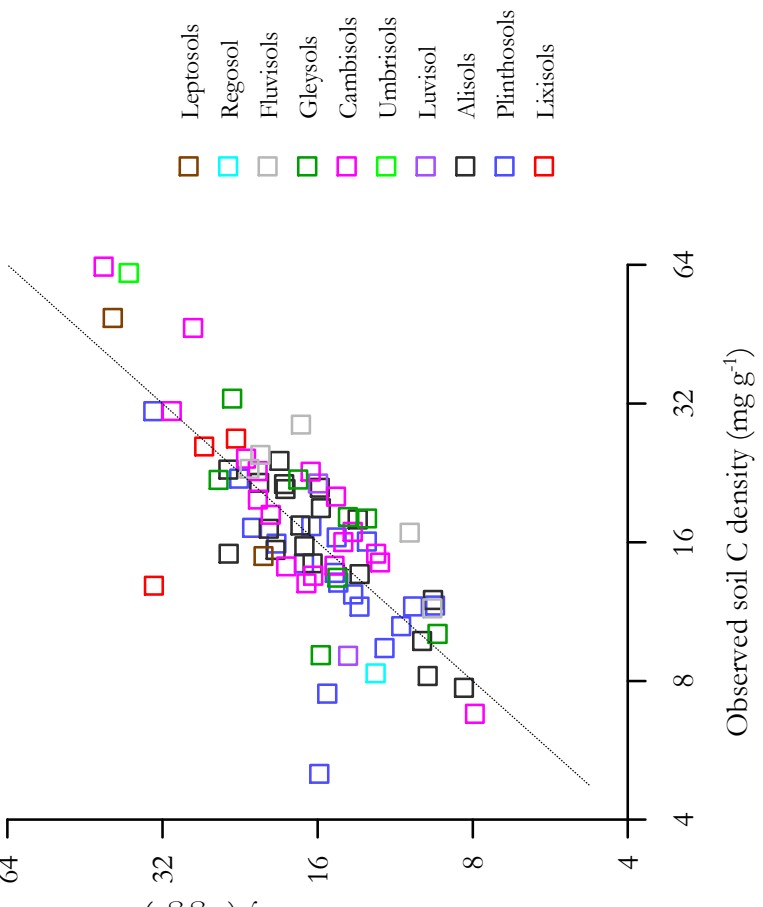

**Fig 10. Fitted vs observed SOC densities for regression model 6s (Table 6).**



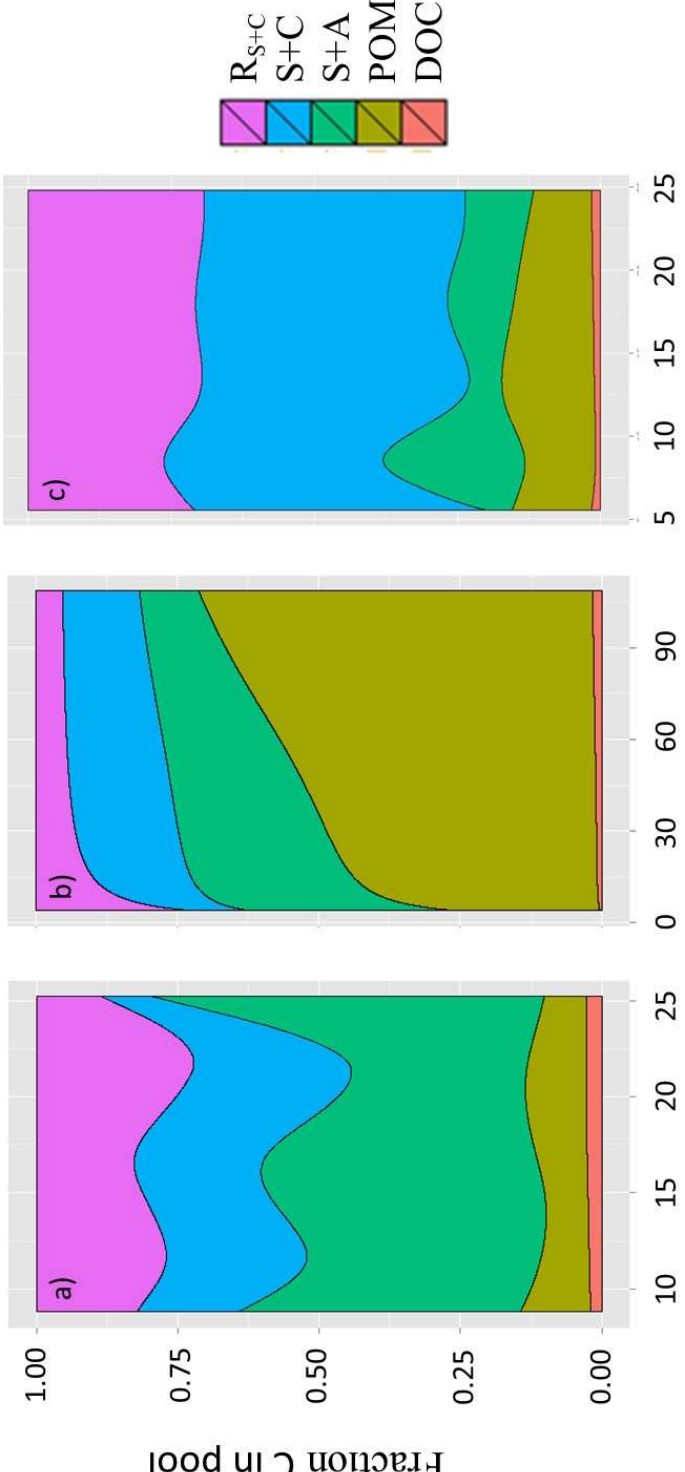

**Figure 11. Fraction of soil carbon in the different pools for the three soil groups. a) LAC soils, b) arenic and c) HAC. Dissolved organic carbon (DOC), particulate organic matter (POM), sand and aggregates (S+A), silt and clay (S+C) and resistant SOC associated to silt and clay fractions.**



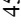

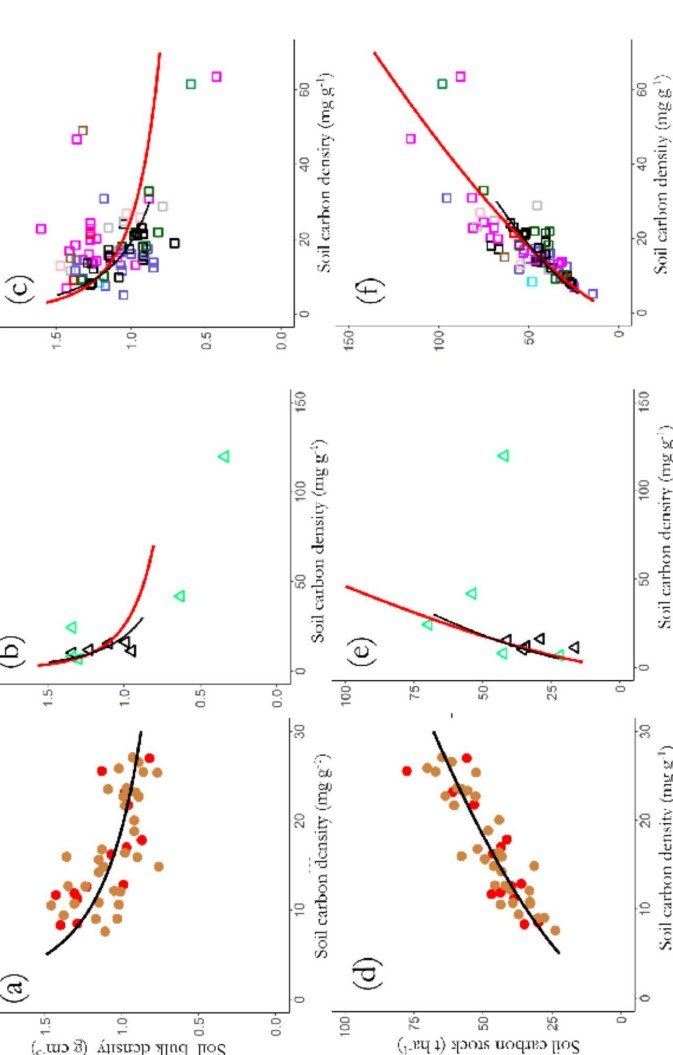

**Figure 12. Variations in bulk density (a) LAC; (b) HAC and (c) arenic; and top soil SOC stocks (d) LAC; (e) HAC and (f) arenic as a function of SOC content. Significant regression lines (see text for details) for each soil group are plotted together for comparison.**



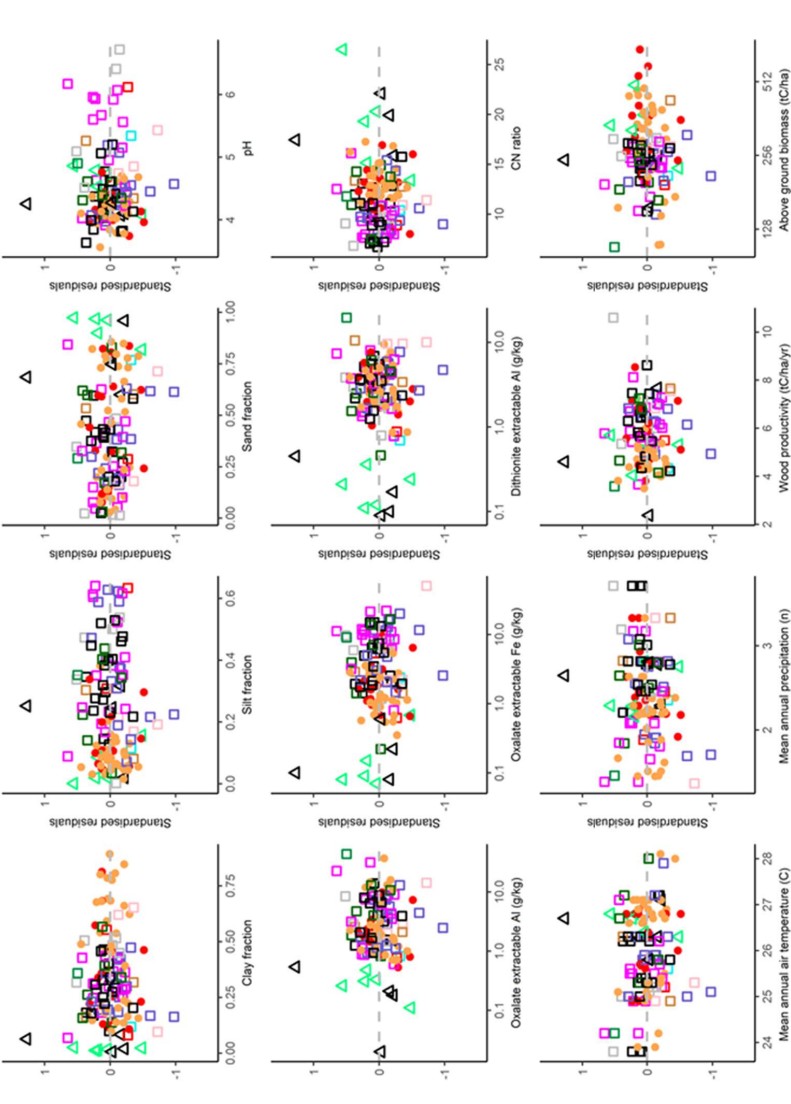

**Figure A1. Standardized regression model residuals plotted against selected climatic, edaphic and vegetation variables.**



**Table 1. Climate/site details and summary of soil physical and chemical characteristics (0.0-0.3m).** Abbreviations used: $T_A$ – mean annual temperature; $P_A$ – mean annual precipitation; $E_V$ –elevation; $I_E$ – effective cation exchange capacity; $\Sigma_B$– sum of bases; $\Sigma_{B(R)}$– total reserve bases; Ch – Chlorite; Gi – gibbsite; Go-goethite; He – Haematite; Il – Illite; Ka – kaolinite; Mi – Mica; Mu – Muscovite; Or/K – orthoclase/K-feldspar; Pl – Plagioglase; Sm – Smectite, Albite – Al, Microcline – Mc. ND – not determined. Soils from the Arenic group (Arenosols/Podzols) followed by F indicate seasonally flooded white sands. For the mineralogies, blank columns indicate that measurements were not made; * = identification uncertain; 0 – none identified. Sites have been numbered and ordered according to their upper layer (0.0-0.3m) soil C content as given in Table A1 (Appendix).

| Soil | Classification | Location | $T_A$ (°C) | $P_A$ (mm) | $E_V$ (m) | pH | Particle fraction | | | $\Sigma_B$ | $I_E$ | $\Sigma_{B(R)}$ | Mineralogy | |
| | | | | | | | Sand | Clay | Silt | (mmol$_c$ kg$^{-1}$) | | | 1° | 2° |
|---|---|---|---|---|---|---|---|---|---|---|---|---|---|---|
| 1 | Plinthosols | Brazil, Acre | 25,1 | 1705,1 | 260 | 4,57 | 0,61 | 0,16 | 0,22 | 7,1 | 13,2 | 189,3 | Ka | Mu, Go, He |
| 2 | Gleysols | Peru, North | 26,3 | 2751,5 | 126 | 4,26 | 0,53 | 0,21 | 0,27 | 4,0 | 36,2 | 40,6 | Mi | Ka |
| 3 | Cambisols | Peru, South | 25,2 | 2457,0 | 358 | 4,53 | 0,23 | 0,36 | 0,41 | 11,0 | 15,4 | 206,5 | Ka | Or/K, Mu, Ch |
| 4 | Podzols$^F$ | Brazil, Roraima | 27,9 | 1836,0 | 46 | 4,91 | 0,78 | 0,05 | 0,17 | 1,1 | 2,8 | 20,1 | | |
| 5 | Plinthosols | Brazil, Acre | 25,0 | 1689,5 | 259 | 4,45 | 0,62 | 0,17 | 0,22 | 7,4 | 14,1 | 215,0 | Mu | Ka, Gi, He |
| 6 | Ferralsols | Venezuela | 28,0 | 2382,0 | 70 | 4,68 | 0,79 | 0,16 | 0,06 | 1,1 | 5,8 | 20,6 | | |
| 7 | Alisols | Peru, South | 25,4 | 2457,6 | 216 | 4,21 | 0,40 | 0,22 | 0,38 | 7,5 | 23,0 | 463,6 | | |
| 8 | Podzols | Brazil, Amazonas | 27,1 | 2289,2 | 92 | 4,10 | 0,96 | 0,02 | 0,02 | 3,1 | 20,1 | 3,1 | Pl | He, Ch |
| 9 | Alisols | Peru, South | 25,3 | 2536,5 | 216 | 4,41 | 0,18 | 0,29 | 0,53 | 5,7 | 29,6 | 362,1 | Il-Sm | Mi, Ka, Al, Go, Gi |
| 10 | Regosol | Brazil, Mato Grosso | 25,6 | 2353,1 | 280 | 5,34 | 0,77 | 0,12 | 0,11 | 20,2 | 22,3 | 109,0 | Ka | Gi, He, Or/K |
| 11 | Acrisols | Brazil, Pará | 26,8 | 2191,6 | 55 | 3,74 | 0,84 | 0,11 | 0,06 | 0,2 | 1,3 | 44,7 | | |
| 12 | Acrisols | Brazil, Acre | 26,0 | 1919,8 | 194 | 4,13 | 0,62 | 0,23 | 0,15 | 6,2 | 9,1 | 85,1 | | |
| 13 | Ferralsols | Venezuela | 28,1 | 2337,0 | 58 | 4,16 | 0,85 | 0,14 | 0,02 | 1,3 | 7,6 | 21,7 | | |
| 14 | Ferralsols | Brazil, Mato Grosso | 25,5 | 1613,1 | 352 | 4,20 | 0,78 | 0,16 | 0,06 | 1,5 | 5,8 | 38,2 | Ka | Gi, Go, He |
| 15 | Luvisols | Peru, South | 25,2 | 2457,0 | 358 | 6,12 | 0,29 | 0,08 | 0,63 | 32,9 | 36,3 | 326,3 | Mu | Ka, Pl, Or/K, He, Gi |
| 16 | Gleysols | Brazil, Roraima | 27,2 | 1839,0 | 60 | 4,40 | 0,73 | 0,17 | 0,10 | 4,2 | 8,3 | 41,1 | | |
| 17 | Arenosols$^F$ | Peru, North | 26,3 | 2751,5 | 127 | 4,14 | 0,94 | 0,03 | 0,04 | 1,7 | 4,0 | 13,0 | Il-Sm | Ka |
| 18 | Ferralsols | Brazil, Pará | 26,7 | 2211,9 | 35 | 4,09 | 0,73 | 0,22 | 0,04 | 2,4 | 10,6 | 63,7 | | |
| 19 | Plinthosols | Brazil, Acre | 25,9 | 1907,0 | 203 | 4,23 | 0,19 | 0,18 | 0,62 | 10,2 | 29,2 | 145,9 | Il-Sm | Ka |
| 20 | Alisols | Peru, South | 25,4 | 2457,6 | 216 | 4,32 | 0,20 | 0,40 | 0,40 | 7,0 | 35,6 | 578,0 | Il-Sm | Mi, Ka, Al |
| 21 | Gleysols | Peru, South | 25,4 | 2457,6 | 217 | 4,05 | 0,17 | 0,39 | 0,44 | 3,4 | 41,4 | 486,0 | Mi | Ka, Il-Sm, Al |
| 22 | Arenosols | Guyana | 26,4 | 2813,3 | 125 | 4,73 | 0,96 | 0,02 | 0,02 | 2,5 | 3,4 | 8,0 | | |
| 23 | Plinthosols | Brazil, Amazonas | 26,4 | 2593,7 | 71 | 3,98 | 0,26 | 0,20 | 0,54 | 1,2 | 10,1 | 44,5 | | |
| 24 | Ferralsols | Brazil, Pará | 26,7 | 2211,9 | 44 | 4,02 | 0,80 | 0,14 | 0,06 | 2,0 | 6,4 | 52,2 | | |
| 25 | Plinthosols | Brazil, Mato Grosso | 25,3 | 1509,7 | 281 | 4,65 | 0,66 | 0,24 | 0,10 | 7,4 | 12,7 | 51,3 | Ka | Gi, He |
| 26 | Ferralsols | Brazil, Mato Grosso | 25,0 | 1854,4 | 326 | 4,19 | 0,86 | 0,10 | 0,04 | 1,2 | 12,1 | 9,5 | Ka | Gi, Mi |





| Soil | Classification | Location | $T_A$ (°C) | $P_A$ (mm) | $E_V$ (m) | pH | Particle fraction | | | $\Sigma_B$ | $I_E$ | $\Sigma_{B(R)}$ | Mineralogy | |
| | | | | | | | Sand | Clay | Silt | (mmol$_c$ kg⁻¹) | | | 1° | 2° |
|---|---|---|---|---|---|---|---|---|---|---|---|---|---|---|
| 27 | Acrisols | Bolivia | 23,3 | 1142,6 | 447 | 5,88 | 0,75 | 0,10 | 0,14 | 17,8 | 18,2 | 230,1 | Ka | Gi,He, Or/K, Pl |
| 28 | Cambisols | Bolivia | 24,8 | 813,4 | 310 | 6,06 | 0,48 | 0,18 | 0,35 | 51,3 | 51,6 | 679,7 | Ka | Gi, Go, He, Mu |
| 29 | Ferralsols | Bolivia | 23,9 | 1451,2 | 299 | 4,63 | 0,74 | 0,20 | 0,06 | 1,6 | 12,5 | 48,8 | Ka | Gi, Go, He, Pl |
| 30 | Arenosols | Peru, North | 26,3 | 2751,5 | 126 | 4,07 | 0,82 | 0,02 | 0,16 | 4,2 | 4,9 | 4,1 | Mu* | Ch |
| 31 | Acrisols | Guyana | 26,4 | 2813,3 | 124 | 4,24 | 0,81 | 0,15 | 0,05 | 3,4 | 10,0 | 17,6 | | |
| 32 | Fluvisols | Peru, South | 25,1 | 2399,4 | 381 | 5,08 | 0,02 | 0,48 | 0,50 | 64,9 | 65,7 | 435,1 | | |
| 33 | Plinthosols | Brazil, Acre | 25,9 | 1946,3 | 205 | 5,19 | 0,18 | 0,20 | 0,63 | 31,0 | 41,8 | 546,4 | | |
| 34 | Plinthosols | Brazil, Amazonas | 26,3 | 2553,3 | 70 | 4,01 | 0,22 | 0,19 | 0,59 | 4,2 | 14,4 | 62,7 | | |
| 35 | Plinthosols | Brazil, Amazonas | 26,3 | 2553,3 | 70 | 3,94 | 0,14 | 0,13 | 0,73 | 4,6 | 16,0 | 44,0 | | |
| 36 | Acrisols | Brazil, Pará | 26,8 | 2178,1 | 38 | 3,96 | 0,24 | 0,46 | 0,30 | 2,6 | 15,6 | 18,7 | Ka | Sm |
| 37 | Arenosols | Guyana | 26,7 | 2282,1 | 97 | 4,79 | 0,97 | 0,01 | 0,02 | 4,0 | 4,5 | 6,9 | | |
| 38 | Acrisols | Brazil, Mato Grosso | 25,6 | 2353,1 | 274 | 4,65 | 0,79 | 0,11 | 0,10 | 15,7 | 20,4 | 66,8 | Ka | Gi |
| 39 | Alisols | Peru, South | 25,3 | 2536,5 | 216 | 5,06 | 0,02 | 0,46 | 0,52 | 49,9 | 56,7 | 978,3 | Ka | Mu, Or/K, Ch, He |
| 40 | Ferralsols | Bolivia | 24,2 | 1456,7 | 198 | 4,70 | 0,58 | 0,36 | 0,06 | 13,2 | 20,1 | 36,5 | Ka | Gi, Sm |
| 41 | Ferralsols | Brazil, Pará | 26,8 | 2191,6 | 43 | 4,23 | 0,52 | 0,39 | 0,09 | 2,7 | 13,6 | 77,7 | | |
| 42 | Plinthosols | Colombia | 25,8 | 2804,1 | 106 | 4,50 | 0,21 | 0,42 | 0,37 | 10,0 | 62,1 | 327,0 | Il-Sm | Ka, Mi |
| 43 | Acrisols | Guyana | 25,7 | 2932,2 | 124 | 4,44 | 0,82 | 0,13 | 0,05 | 2,8 | 10,9 | 31,0 | | |
| 44 | Ferralsols | Guyana | 26,6 | 2633,8 | 108 | 4,25 | 0,79 | 0,18 | 0,03 | 2,7 | 13,0 | 21,8 | | |
| 45 | Ferralsols | Guyana | 26,6 | 2633,8 | 106 | 4,03 | 0,76 | 0,20 | 0,04 | 2,9 | 11,2 | 22,0 | | |
| 46 | Acrisols | Brazil, Pará | 26,8 | 2178,1 | 40 | 4,00 | 0,64 | 0,25 | 0,11 | 3,0 | 18,1 | 10,8 | Ka | Gi |
| 47 | Lixisols | Venezuela | 25,3 | 1364,4 | 291 | 5,43 | 0,71 | 0,10 | 0,19 | 17,8 | 17,9 | 45,2 | Ka | Sm, Mi |
| 48 | Cambisols | Peru, North | 26,3 | 2805,5 | 97 | 5,15 | 0,10 | 0,28 | 0,62 | 50,7 | 57,2 | 496,4 | Il-Sm | Ka, Mi, Al, Mc |
| 49 | Plinthosols | Venezuela | 25,8 | 2810,2 | 98 | 4,13 | 0,38 | 0,31 | 0,31 | 2,8 | 44,4 | 233,4 | Il-Sm | Ka, Mi, Gi |
| 50 | Podzols^F | Brazil, Amazonas | 27,1 | 2289,2 | 100 | 4,73 | 0,89 | 0,09 | 0,02 | 1,3 | 3,1 | 1,6 | | |
| 51 | Gleysols | Venezuela | 28,0 | 2499,0 | 89 | 4,61 | 0,83 | 0,14 | 0,03 | 1,9 | 8,8 | 20,4 | | |
| 52 | Cambisols | Brazil, Acre | 25,7 | 1803,7 | 278 | 5,56 | 0,39 | 0,25 | 0,35 | 73,7 | 73,7 | 564,9 | Ka | Pl, Or/K, Mu, He |
| 53 | Alisols | Bolivia | 25,0 | 3076,8 | 229 | 4,24 | 0,43 | 0,25 | 0,32 | 4,8 | 18,0 | 304,4 | | |
| 54 | Plinthosols | Colombia | 25,8 | 2804,1 | 107 | 4,29 | 0,19 | 0,43 | 0,38 | 10,2 | 62,60 | 385,1 | Il-Sm | |
| 55 | Cambisols | Peru, South | 25,4 | 2457,6 | 219 | 4,22 | 0,47 | 0,29 | 0,24 | 2,2 | 33,80 | 185,1 | Il-Sm | Ka, Mi, Gi |
| 56 | Cambisols | Ecuador | 24,9 | 3172,3 | 261 | 4,95 | 0,47 | 0,30 | 0,23 | 77,9 | 84,80 | 928,5 | Il-Sm | Ka |
| 57 | Ferralsols | Bolivia | 23,9 | 1451,2 | 300 | 4,39 | 0,73 | 0,21 | 0,06 | 1,7 | 15,2 | 50,1 | Ka | Gi, He, Mu |
| 58 | Alisols | Brazil, Rondônia | 27,2 | 2208,0 | 78 | 3,81 | 0,20 | 0,34 | 0,46 | 2,0 | 30,0 | 78,6 | | |
| 59 | Plinthosols | Brazil, Acre | 25,9 | 1907,0 | 205 | 5,07 | 0,16 | 0,25 | 0,59 | 50,0 | 56,2 | 345,3 | Il-Sm | Mi, Ka |
| 60 | Cambisols | Peru, South | 25,6 | 2095,9 | 203 | 5,60 | 0,15 | 0,25 | 0,60 | 85,5 | 86,9 | 1047,9 | | |
| 61 | Ferralsols | Brazil, Amazonas | 26,9 | 2409,0 | 114 | 4,29 | 0,82 | 0,62 | 0,13 | 2,6 | 16,6 | 45,0 | | |
| 62 | Ferralsols | Guyana | 26,6 | 2633,8 | 101 | 4,37 | 0,82 | 0,15 | 0,03 | 3,6 | 12,2 | 19,4 | | |

| Soil | Classification | Location | $T_A$ (°C) | $P_A$ (mm) | $E_V$ (m) | pH | Sand | Clay | Silt | $\Sigma_B$ | $I_E$ | $\Sigma_{B(R)}$ | 1° | 2° |
|---|---|---|---|---|---|---|---|---|---|---|---|---|---|---|
| | | | | | | | Particle fraction | | | | (mmol$_c$ kg$^{-1}$) | | Mineralogy | |
| 63 | Leptosols | French Guyana | 25,0 | 3329,2 | 140 | 4,34 | 0,60 | 0,32 | 0,08 | 4,5 | 24,0 | 74,0 | | |
| 64 | Cambisols | Peru, South | 25,4 | 2457,6 | 218 | 3,91 | 0,40 | 0,44 | 0,17 | 2,2 | 44,7 | 272,8 | Il-Sm | Mi, Ka, Al, Mc |
| 65 | Alisols | Colombia | 25,8 | 2777,6 | 120 | 4,13 | 0,58 | 0,20 | 0,22 | 2,4 | 26,0 | 80,1 | Ka | Il-Sm, Mi, Gi |
| 66 | Alisols | Brazil, Rondônia | 27,2 | 2208,0 | 83 | 3,82 | 0,27 | 0,26 | 0,48 | 1,2 | 29,3 | 75,0 | | |
| 67 | Arenosols | Guyana | 26,8 | 2158,5 | 102 | 4,53 | 0,90 | 0,01 | 0,09 | 3,0 | 7,8 | 28,3 | | |
| 68 | Ferralsols | French Guyana | 24,9 | 3329,2 | 140 | 4,40 | 0,52 | 0,38 | 0,10 | 4,6 | 13,2 | 72,6 | Ka | Gi, Go |
| 69 | Alisols | Peru, North | 26,3 | 2805,5 | 97 | 5,20 | 0,32 | 0,27 | 0,40 | 68,8 | 92,1 | 464,1 | Il-Sm | Ka, Sm, Gi |
| 70 | Plinthosols | Peru, North | 26,3 | 2814,8 | 113 | 4,55 | 0,38 | 0,47 | 0,14 | 7,9 | 37,7 | 275,4 | Ka | Il-Sm |
| 71 | Ferralsols | Brazil, Mato Grosso | 25,3 | 1509,7 | 281 | 4,20 | 0,47 | 0,45 | 0,08 | 4,8 | 22,0 | 103,0 | Ka | Gi, He, Go, Or/K |
| 72 | Ferralsols | Brazil, Pará | 26,9 | 2197,2 | 42 | 4,03 | 0,46 | 0,48 | 0,06 | 2,7 | 17,0 | 71,1 | | |
| 73 | Cambisols | Ecuador | 24,9 | 3172,3 | 266 | 4,63 | 0,36 | 0,29 | 0,35 | 89,8 | 124,7 | 835,0 | Il-Sm | Ka, Mi |
| 74 | Plinthosols | Bolivia | 25,0 | 3076,8 | 229 | 4,07 | 0,30 | 0,23 | 0,47 | 4,2 | 9,7 | 261,9 | | |
| 75 | Arenosols | Guyana | 26,8 | 2289,6 | 98 | 4,86 | 0,97 | 0,03 | 0,00 | 4,6 | 6,0 | 6,3 | | |
| 76 | Acrisols | Guyana | 26,8 | 2289,6 | 98 | 4,20 | 0,59 | 0,36 | 0,05 | 4,1 | 19,1 | 27,7 | | |
| 77 | Ferralsols | Brazil, Pará | 25,4 | 1883,1 | 145 | 3,78 | 0,23 | 0,66 | 0,10 | 4,0 | 21,4 | 10,7 | Ka | Sm |
| 78 | Plinthosols | Venezuela | 25,8 | 2810,2 | 98 | 3,97 | 0,24 | 0,36 | 0,40 | 2,9 | 53,4 | 296,3 | Il-Sm | Ka, Mi, Gi |
| 79 | Fluvisols | Peru, South | 25,0 | 3192,2 | 274 | 4,51 | 0,02 | 0,50 | 0,47 | 47,5 | 58,1 | 952,4 | | |
| 80 | Ferralsols | Brazil, Pará | 26,7 | 2211,9 | 42 | 3,79 | 0,33 | 0,54 | 0,14 | 4,1 | 22,8 | 61,6 | Ka | Go, Gi |
| 81 | Cambisols | Peru, South | 25,5 | 2079,3 | 203 | 5,93 | 0,05 | 0,31 | 0,64 | 78,9 | 80,7 | 1253,3 | | |
| 82 | Acrisols | Guyana | 26,8 | 2387,0 | 90 | 4,07 | 0,60 | 0,34 | 0,06 | 3,3 | 18,0 | 28,5 | | |
| 83 | Alisols | Peru, North | 26,3 | 2777,8 | 126 | 4,47 | 0,78 | 0,10 | 0,13 | 1,3 | 24,6 | 114,7 | Il-Sm | Mi, Ka |
| 84 | Plinthosols | Venezuela | 27,9 | 2510,0 | 114 | 4,44 | 0,63 | 0,30 | 0,07 | 2,6 | 10,0 | 21,1 | | |
| 85 | Alisols | Brazil, Rondônia | 27,7 | 923,5 | 83 | 3,64 | 0,48 | 0,34 | 0,18 | 1,6 | 23,8 | 40,4 | | |
| 86 | Plinthosols | Peru, North | 26,3 | 2751,5 | 127 | 4,46 | 0,33 | 0,34 | 0,33 | 10,8 | 51,3 | 94,0 | Il-Sm | Ka, Gi |
| 87 | Alisols | Ecuador | 25,3 | 3008,9 | 237 | 4,61 | 0,43 | 0,34 | 0,23 | 33,8 | 51,0 | 441,8 | Ka | Il-Sm, Mi |
| 88 | Acrisols | Peru, North | 26,3 | 2814,8 | 113 | 4,42 | 0,32 | 0,46 | 0,22 | 5,4 | 37,1 | 224,9 | Ka | Sm, Gi |
| 89 | Alisols | Brazil, Rondônia | 26,2 | 2205,4 | 78 | 3,63 | 0,47 | 0,18 | 0,35 | 1,5 | 23,1 | 32,6 | | |
| 90 | Gleysols | Peru, North | 26,7 | 2645,1 | 140 | 4,31 | 0,62 | 0,16 | 0,22 | 4,1 | 17,8 | 172,0 | Ka | He |
| 91 | Gleysols | Ecuador | 25,3 | 3008,9 | 235 | 4,39 | 0,03 | 0,57 | 0,40 | 71,0 | 78,3 | 832,3 | Il-Sm | Ka, Sm |
| 92 | Cambisols | Peru, South | 25,5 | 2079,3 | 203 | 6,07 | 0,05 | 0,42 | 0,52 | 68,2 | 68,2 | 1225,3 | Mica | Il, Ka |
| 93 | Acrisols | Venezuela | 26,2 | 3425,0 | 109 | 4,79 | 0,88 | 0,08 | 0,03 | 3,0 | 12,2 | 6,5 | Ka | Gi, Mi |
| 94 | Acrisols | Bolivia | 24,1 | 1270,3 | 268 | 6,30 | 0,49 | 0,31 | 0,20 | 21,3 | 22,2 | 209,0 | Ka | He, Pl |
| 95 | Ferralsols | Brazil, Amazonas | 27,1 | 2289,2 | 100 | 4,34 | 0,08 | 0,85 | 0,05 | 1,9 | 21,4 | | | |
| 96 | Alisols | Peru, North | 26,3 | 2814,8 | 114 | 3,99 | 0,38 | 0,18 | 0,45 | 3,8 | 29,9 | 185,2 | | |
| 97 | Cambisols | Bolivia | 24,3 | 1066,0 | 373 | 5,23 | 0,55 | 0,18 | 0,26 | 60,7 | 61,9 | 283,4 | | |
| 98 | Ferralsols | Brazil, Amazonas | 27,0 | 2444,4 | 111 | 4,17 | 0,30 | 0,59 | 0,11 | 2,9 | 12,4 | 34,9 | | |





| Soil | Classification | Location | $T_A$ (°C) | $P_A$ (mm) | $E_V$ (m) | pH | Particle fraction | | | $\Sigma_B$ | $I_E$ | $\Sigma_{B(R)}$ | Mineralogy | |
| | | | | | | | Sand | Clay | Silt | (mmol. kg⁻¹) | | | 1° | 2° |
|---|---|---|---|---|---|---|---|---|---|---|---|---|---|---|
| 99 | Cambisols | Brazil, Roraima | 27,0 | 1855,0 | 153 | 4,25 | 0,43 | 0,36 | 0,22 | 9,5 | 16,1 | 120,0 | | |
| 100 | Acrisols | Venezuela | 26,2 | 3425,0 | 99 | 5,03 | 0,89 | 0,05 | 0,06 | 1,8 | 11,3 | 1,9 | Ka | Sm, Mi |
| 101 | Alisols | Ecuador | 23,8 | 3710,7 | 431 | 4,49 | 0,40 | 0,33 | 0,27 | 11,1 | 26,6 | 333,9 | Ka | Mi, Gi |
| 102 | Alisols | Peru, North | 26,3 | 2814,8 | 113 | 4,03 | 0,39 | 0,46 | 0,15 | 3,0 | 30,7 | 85,4 | | |
| 103 | Alisols | Brazil, Rondônia | 26,2 | 2205,4 | 87 | 3,84 | 0,60 | 0,17 | 0,24 | 1,8 | 19,7 | 32,1 | | |
| 104 | Luvisols | Brazil, Acre | 25,7 | 1883,8 | 228 | 4,26 | 0,14 | 0,55 | 0,31 | 25,8 | 45,9 | 461,1 | | |
| 105 | Alisols | Ecuador | 23,8 | 3710,7 | 432 | 4,77 | 0,41 | 0,29 | 0,30 | 20,4 | 30,5 | 330,2 | Ka | Mi |
| 106 | Ferralsols | Brazil, Amazonas | 27,1 | 2245,7 | 95 | 4,24 | 0,16 | 0,68 | 0,16 | 2,4 | 28,1 | 4,7 | Ka | Sm, Gi |
| 107 | Acrisols | Peru, North | 26,8 | 2630,0 | 122 | 3,98 | 0,44 | 0,22 | 0,34 | 2,6 | 20,2 | 68,8 | Ka | Go, Gi |
| 108 | Gleysols | Colombia | 25,8 | 2799,9 | 120 | 4,34 | 0,32 | 0,34 | 0,34 | 5,5 | 35,1 | 150,0 | Ka | Mu, Gi, He, Go |
| 109 | Gleysols | Brazil, Roraima | 27,3 | 1840,0 | 62 | 4,51 | 0,78 | 0,15 | 0,06 | 2,6 | 7,3 | 22,3 | | |
| 110 | Plinthosols | Brazil, Amazonas | 26,4 | 2593,7 | 71 | 4,00 | 0,36 | 0,18 | 0,45 | 1,9 | 14,9 | 47,3 | | |
| 111 | Ferralsols | Brazil, Amazonas | 27,1 | 2245,7 | 93 | 3,98 | 0,09 | 0,78 | 0,13 | 3,5 | 21,6 | 42,1 | | |
| 112 | Ferralsols | Brazil, Amapá | 26,8 | 2377,1 | 80 | 4,05 | 0,04 | 0,81 | 0,15 | 5,3 | 21,5 | 25,5 | Ka | Gi |
| 113 | Cambisols | Peru, South | 25,5 | 2079,3 | 203 | 5,96 | 0,08 | 0,31 | 0,61 | 80,1 | 81,2 | 1304,2 | | |
| 114 | Cambisols | Brazil, Acre | 25,8 | 1652,5 | 236 | 5,92 | 0,25 | 0,37 | 0,38 | 80,3 | 80,4 | 845,6 | Ka | Mu, Pl, Or/K, He |
| 115 | Alisols | Brazil, Roraima | 27,3 | 1841,0 | 126 | 4,08 | 0,33 | 0,44 | 0,23 | 3,8 | 16,4 | 73,9 | | |
| 116 | Fluvisols | Peru, South | 25,2 | 2477,1 | 356 | 6,72 | 0,01 | 0,45 | 0,54 | 85,3 | 85,9 | 1688,1 | Ka | Mu, Ch, Pl, Go |
| 117 | Ferralsols | Brazil, Amazonas | 27,1 | 2193,2 | 110 | 4,27 | 0,13 | 0,46 | 0,41 | 1,9 | 17,5 | 24,3 | | |
| 118 | Acrisols | French Guyana | 24,9 | 3329,2 | 140 | 4,16 | 0,33 | 0,57 | 0,10 | 5,3 | 21,3 | 31,2 | | |
| 119 | Ferralsols | Brazil, Pará | 26,9 | 2175,8 | 43 | 4,13 | 0,23 | 0,68 | 0,09 | 2,7 | 8,5 | 70,5 | | |
| 120 | Ferralsols | Brazil, Pará | 26,7 | 2211,6 | 45 | 4,27 | 0,14 | 0,79 | 0,08 | 3,1 | 17,7 | 68,3 | | |
| 121 | Ferralsols | Brazil, Amazonas | 27,1 | 2193,2 | 112 | 4,14 | 0,10 | 0,69 | 0,20 | 1,0 | 11,3 | 30,2 | | |
| 122 | Alisols | Ecuador | 23,8 | 3710,7 | 431 | 4,37 | 0,42 | 0,31 | 0,28 | 9,0 | 32,4 | 288,2 | Ka | Mi, Il-Sm |
| 123 | Podzols[F] | Colombia | 25,8 | 2799,9 | 120 | 4,27 | 0,75 | 0,01 | 0,25 | 6,4 | 7,1 | 3,3 | Mu | Ch |
| 124 | Cambisols | Bolivia | 24,3 | 1066,0 | 373 | 6,84 | 0,58 | 0,19 | 0,23 | 75,6 | 75,9 | 566,7 | | |
| 125 | Fluvisols | Peru, South | 25,2 | 2457,0 | 356 | 6,41 | 0,48 | 0,52 | 0,00 | 84,5 | 85,2 | 1688,7 | Mu | Ka, Ch, Or/K, Pl |
| 126 | Ferralsols | Brazil, Pará | 25,1 | 2015,9 | 197 | 3,84 | 0,03 | 0,89 | 0,08 | 6,4 | 29,7 | 16,6 | Ka | 0 |
| 127 | Ferralsols | Brazil, Mato Grosso | 25,1 | 1665,8 | 373 | 4,10 | 0,46 | 0,49 | 0,05 | 2,3 | 19,5 | 28,3 | Ka | Gi, Go, He, Mu |
| 128 | Acrisols | French Guyana | 24,9 | 3329,2 | 140 | 4,76 | 0,12 | 0,68 | 0,20 | 10,9 | 15,5 | 87,9 | Ka | Gi, Go |
| 129 | Lixisols | French Guyana | 24,9 | 3329,2 | 140 | 4,85 | 0,18 | 0,65 | 0,17 | 13,6 | 16,9 | 65,5 | | |
| 130 | Ferralsols | Brazil, Amazonas | 27,1 | 2289,2 | 106 | 3,94 | 0,20 | 0,68 | 0,12 | 3,7 | 21,9 | 5,0 | Ka | 0 |
| 131 | Ferralsols | Brazil, Amazonas | 27,1 | 2193,2 | 105 | 3,56 | 0,08 | 0,54 | 0,38 | 2,4 | 11,6 | 30,0 | | |
| 132 | Lixisols | French Guyana | 24,9 | 3329,2 | 140 | 4,74 | 0,17 | 0,62 | 0,21 | 16,2 | 17,4 | 64,2 | | |
| 133 | Acrisols | Brazil, Amazonas | 26,9 | 2457,9 | 119 | 4,29 | 0,08 | 0,81 | 0,11 | 2,7 | 15,4 | 43,3 | | |
| 134 | Ferralsols | Brazil, Amazonas | 27,1 | 2245,7 | 93 | 4,08 | 0,10 | 0,80 | 0,10 | 4,9 | 19,8 | 8,6 | | |



| Soil | Classification | Location | $T_A$ (°C) | $P_A$ (mm) | $E_V$ (m) | pH | Particle fraction | | | $\Sigma_B$ | $I_E$ | $\Sigma_{B(R)}$ | Mineralogy | |
| | | | | | | | Sand | Clay | Silt | (mmol$_c$ kg$^{-1}$) | | | 1° | 2° |
|---|---|---|---|---|---|---|---|---|---|---|---|---|---|---|
| 135 | Fluvisols | Ecuador | 23,8 | 3710,7 | 394 | 5,09 | 0,35 | 0,32 | 0,34 | 81,7 | 81,9 | 1181,6 | Il-Sm | Ka, Sm |
| 136 | Plinthosols | Brazil, Roraima | 27,2 | 1841,0 | 59 | 4,43 | 0,43 | 0,38 | 0,19 | 4,0 | 14,2 | 72,1 | | |
| 137 | Cambisols | Bolivia | 24,2 | 1383,6 | 248 | 5,67 | 0,63 | 0,18 | 0,19 | 76,5 | 76,6 | 755,3 | Il-Sm | Ka, Mi |
| 138 | Gleysols | Brazil, Roraima | 27,2 | 1840,0 | 64 | 4,61 | 0,60 | 0,26 | 0,14 | 2,9 | 8,6 | 24,7 | | |
| 139 | Podzols[F] | Peru, North | 26,3 | 2777,8 | 124 | 4,07 | 0,60 | 0,08 | 0,32 | 6,8 | 8,5 | 10,5 | Pl | Ch, Ka, He |
| 140 | Cambisols | Brazil, Roraima | 27,1 | 1846,0 | 85 | 4,02 | 0,33 | 0,43 | 0,25 | 3,1 | 15,3 | 64,0 | | |
| 141 | Leptosols | Venezuela | 26,3 | 2820,7 | 366 | 5,26 | 0,53 | 0,16 | 0,30 | 23,5 | 27,7 | | | |
| 142 | Umbrisols | Bolivia | 24,2 | 1456,7 | 195 | 4,74 | 0,29 | 0,36 | 0,35 | 6,6 | 25,5 | 259,7 | | |
| 143 | Umbrisols | Bolivia | 24,2 | 1456,7 | 195 | 4,90 | 0,29 | 0,36 | 0,35 | 8,7 | 20,8 | 179,5 | Ka | Il, Mi |
| 144 | Cambisols | Bolivia | 24,2 | 1383,6 | 248 | 6,17 | 0,84 | 0,07 | 0,09 | 50,7 | 52,2 | 715,1 | Mi | Ka |
| 145 | Podzols[F] | Venezuela | 26,2 | 3425,0 | 99 | 4,88 | | | | 18,2 | 18,6 | | | |
| 146 | Gleysols | Peru, North | 26,3 | 2801,3 | 114 | 4,03 | | | | 6,20 | 62,3 | | | |
| 147 | Podzols[F] | Peru, North | 26,7 | 2646,5 | 127 | 4,25 | 0,69 | 0,06 | 0,25 | 3,9 | 7,1 | 20,0 | Mu* | Ch |



**Table A1. Soil carbon and associated measures of the study soils (0.0-0.3m). [C] - C concentration; CN – carbon/nitrogen ratio; $\rho$ - bulk density; $\int_C$ – total soil C; Fe$_d$ – dithionite extractable iron, Fe$_o$ – oxalate extractable iron, Al$_d$ – dithionite extractable aluminium, Al$_o$ – oxalate extractable aluminium, Al$_o$ – pyrophosphate extractable aluminium**

| Soil | [C] (mg g$^{-1}$) | CN | $\rho$ (kg dm$^{-3)}$ | $\int_C$ (Mg ha$^{-1}$) | Fe$_d$ | Fe$_o$ | Fe$_d$ –Fe$_o$ | Al$_d$ | Al$_o$ | Al$_p$ |
|---|---|---|---|---|---|---|---|---|---|---|
| | | | | | | | g kg$^{-1}$ | | | |
| 1 | 5,03 | 9,04 | 1,05 | 14,26 | 19,61 | 2,54 | 17,08 | 4,71 | 2,46 | 0,60 |
| 2 | 6,78 | 11,07 | 1,15 | 119,23 | 2,01 | 0,00 | 2,01 | 1,79 | | 0,77 |
| 3 | 6,79 | 7,64 | 1,43 | 24,50 | 3,15 | 0,80 | 2,35 | 0,78 | 0,60 | 0,54 |
| 4 | 6,80 | 22,67 | 1,30 | 21,90 | 0,24 | 0,19 | 0,05 | 0,28 | 0,03 | 0,27 |
| 5 | 7,51 | 10,52 | 1,17 | 25,11 | 13,90 | 11,57 | 2,33 | 2,69 | 4,42 | 0,57 |
| 6 | 7,60 | 12,67 | 1,11 | 23,80 | 4,65 | 1,13 | 3,52 | 0,87 | 1,00 | 1,68 |
| 7 | 7,73 | 7,00 | 1,27 | 25,96 | 11,25 | 3,44 | 7,81 | 1,55 | 0,93 | 1,09 |
| 8 | 7,93 | 15,94 | 1,34 | 42,57 | 0,25 | 0,22 | 0,04 | 0,17 | 0,18 | 0,12 |
| 9 | 8,20 | 7,20 | 1,26 | 27,02 | 11,24 | 5,43 | 5,81 | 2,20 | 3,16 | 0,98 |
| 10 | 8,31 | 9,97 | 1,21 | 48,02 | 4,50 | 2,38 | 2,12 | 0,69 | 0,82 | 1,36 |
| 11 | 8,31 | 13,07 | 1,40 | 35,01 | 8,60 | 1,61 | 6,99 | 1,40 | 0,53 | 1,54 |
| 12 | 8,49 | 7,80 | 1,29 | 29,95 | 17,04 | 0,66 | 16,38 | 2,01 | 0,80 | 0,83 |
| 13 | 9,00 | 12,86 | 1,17 | 30,05 | 5,60 | 1,11 | 4,49 | 0,86 | 0,71 | 1,12 |
| 14 | 9,03 | 14,53 | 1,03 | 27,81 | 20,46 | 1,02 | 19,44 | 1,65 | 0,69 | 2,27 |
| 15 | 9,07 | 8,04 | 1,38 | 29,66 | 6,24 | 0,62 | 5,62 | 0,88 | 0,77 | 0,14 |
| 16 | 9,10 | 13,00 | 1,33 | 35,07 | 14,60 | 1,43 | 13,17 | 1,33 | 0,44 | 0,82 |
| 17 | 9,12 | 14,19 | 0,89 | 21,69 | 0,37 | 0,34 | 0,03 | 0,03 | | |
| 18 | 9,41 | 11,88 | 1,38 | 37,06 | 7,65 | 0,35 | 7,30 | 1,76 | 0,72 | 2,14 |
| 19 | 9,43 | 9,34 | 1,32 | 32,36 | 14,40 | 4,73 | 9,67 | 2,23 | 2,83 | 1,72 |
| 20 | 9,77 | 6,71 | 1,08 | 28,50 | 15,43 | 6,16 | 9,27 | 3,05 | 3,97 | 1,35 |
| 21 | 10,12 | 7,29 | 1,18 | 30,78 | 11,04 | 7,04 | 4,00 | 2,52 | 4,01 | 1,46 |
| 22 | 10,14 | 22,79 | 1,34 | 35,31 | 0,30 | 0,07 | 0,23 | 0,12 | 0,33 | 0,06 |
| 23 | 10,52 | 12,57 | | 21,66 | 1,60 | 0,79 | 0,81 | 0,49 | 0,75 | 0,80 |
| 24 | 10,52 | 12,35 | 1,46 | 43,57 | 7,35 | 0,54 | 6,81 | 1,77 | 0,86 | 2,17 |
| 25 | 10,61 | 13,65 | 1,02 | 32,88 | 0,57 | 0,55 | 0,02 | 2,78 | 1,68 | 2,94 |
| 26 | 10,71 | 14,23 | 1,31 | 39,13 | 3,18 | 1,37 | 1,81 | 2,49 | 6,60 | 1,16 |
| 27 | 10,75 | 9,56 | 1,34 | 45,40 | | | | | | |
| 28 | 10,76 | 9,56 | 1,27 | 48,91 | | | | | | |
| 29 | 10,85 | 12,99 | 1,15 | 33,01 | 9,59 | 6,28 | 3,31 | 6,47 | 13,08 | 2,11 |
| 30 | 11,26 | 13,47 | 0,95 | 16,36 | 0,68 | 0,56 | 0,12 | 0,24 | 0,11 | 0,03 |
| 31 | 11,28 | 13,22 | 1,29 | 38,63 | 6,03 | 0,68 | 5,35 | 1,69 | 0,87 | 1,69 |
| 32 | 11,50 | 7,52 | 1,40 | 38,91 | 4,77 | 1,17 | 3,60 | 0,79 | 0,89 | 0,21 |
| 33 | 11,60 | 7,16 | 1,37 | 33,97 | 28,62 | 3,00 | 25,62 | 2,54 | 1,14 | 0,86 |
| 34 | 11,61 | 9,58 | 1,06 | 55,28 | 10,14 | 5,03 | 5,11 | 1,65 | 0,94 | 1,16 |
| 35 | 11,66 | 11,32 | | 25,81 | 0,23 | 0,15 | 0,08 | 0,31 | 0,59 | 0,62 |
| 36 | 11,68 | 16,10 | 1,43 | 46,93 | 7,40 | 6,43 | 0,97 | 2,81 | 7,30 | 1,19 |
| 37 | 11,77 | 22,47 | 1,23 | 34,37 | 1,26 | 0,09 | 1,17 | 0,11 | 0,31 | 0,03 |
| 38 | 11,88 | 10,08 | 1,31 | 43,66 | 6,12 | 1,83 | 4,29 | 2,75 | 7,64 | 2,03 |
| 39 | 11,99 | 7,05 | 1,25 | 37,36 | 14,87 | 5,20 | 9,68 | 1,24 | 6,37 | 0,86 |
| 40 | 12,09 | 11,68 | 1,01 | 33,12 | 11,54 | 6,37 | 5,17 | 7,71 | 15,97 | 0,02 |
| 41 | 12,17 | 11,46 | 1,05 | 39,92 | 10,62 | 0,68 | 9,94 | 2,51 | 1,53 | 1,73 |
| 42 | 12,33 | 8,18 | 0,85 | 28,95 | 20,53 | 8,55 | 11,98 | 4,13 | 5,67 | 1,92 |
| 43 | 12,60 | 12,90 | 1,23 | 39,71 | 4,26 | 0,96 | 3,30 | 1,04 | 0,92 | 1,84 |
| 44 | 12,65 | 14,28 | 1,24 | 41,86 | 6,66 | 0,69 | 5,97 | 2,04 | 0,95 | 1,96 |
| 45 | 12,69 | 11,69 | 1,35 | 45,64 | 6,24 | 0,96 | 5,28 | 1,83 | 1,17 | 2,20 |
| 46 | 12,85 | 16,87 | 0,99 | 36,07 | 6,76 | 3,14 | 3,62 | 3,89 | 9,89 | 1,14 |
| 47 | 12,88 | 11,43 | 1,47 | 54,28 | 76,11 | 50,27 | 25,84 | 10,00 | 14,20 | 1,03 |
| 48 | 13,03 | 8,34 | 0,97 | 33,32 | 35,32 | 10,88 | 24,44 | 4,30 | 4,31 | 1,52 |
| 49 | 13,08 | 9,52 | 1,07 | 38,04 | 11,44 | 10,08 | 1,36 | 4,31 | 7,37 | 1,47 |
| 50 | 13,35 | 17,63 | | 34,87 | 1,20 | 0,88 | 0,32 | 1,30 | 3,16 | 3,32 |
| 51 | 13,40 | 14,89 | 1,26 | 42,55 | 0,22 | 0,20 | 0,02 | 0,46 | 0,82 | 0,53 |
| 52 | 13,54 | 9,90 | 1,25 | 31,89 | 7,72 | 6,12 | 1,60 | 1,63 | 3,48 | 0,58 |
| 53 | 13,65 | 8,58 | 1,29 | 45,24 | 20,01 | 1,84 | 18,17 | 3,69 | 1,66 | 2,60 |
| 54 | 13,73 | 8,55 | 0,85 | 31,36 | 20,71 | 15,97 | 4,74 | 5,62 | 8,85 | 1,84 |



| Soil | [C] (mg g⁻¹) | CN | ρ (kg dm⁻³) | ∫C (Mg ha⁻¹) | Fe_d | Fe_o | Fe_d −Fe_o | Al_d | Al_o | Al_p |
|------|------|------|------|------|------|------|------|------|------|------|
| | | | | | | | g kg⁻¹ | | | |
| 55 | 14,18 | 10,38 | 1,23 | 46,31 | 13,81 | 8,50 | 5,31 | 6,26 | 8,99 | 2,02 |
| 56 | 14,23 | 8,03 | 1,14 | 41,99 | 15,87 | 8,30 | 7,57 | 3,83 | 6,90 | 1,22 |
| 57 | 14,25 | 13,84 | 1,15 | 43,24 | 7,47 | 3,02 | 4,45 | 4,18 | 14,53 | 3,00 |
| 58 | 14,40 | 11,08 | 0,92 | 44,70 | 32,60 | 2,53 | 30,07 | 4,76 | 1,54 | 2,63 |
| 59 | 14,41 | 9,86 | 1,36 | 50,54 | 26,80 | 12,77 | 14,03 | 2,82 | 3,72 | 3,62 |
| 60 | 14,46 | 6,80 | 1,32 | 40,82 | 14,49 | 9,93 | 4,56 | 1,34 | 1,12 | 0,41 |
| 61 | 14,87 | 11,62 | 0,76 | 31,39 | 5,19 | 2,74 | 2,45 | 1,48 | 1,20 | 0,92 |
| 62 | 14,87 | 14,66 | 1,12 | 46,06 | 5,25 | 0,57 | 4,68 | 1,72 | 1,04 | 2,99 |
| 63 | 14,93 | 12,63 | 1,40 | 63,47 | 11,82 | 1,41 | 10,41 | 3,38 | 2,10 | 2,57 |
| 64 | 15,11 | 9,37 | 1,09 | 43,71 | 13,93 | 10,64 | 3,29 | 3,23 | 8,43 | 1,87 |
| 65 | 15,11 | 15,77 | 1,14 | 43,09 | 3,08 | 1,94 | 1,14 | 2,35 | 3,91 | 1,71 |
| 66 | 15,40 | 12,83 | 0,92 | 40,55 | 28,85 | 2,06 | 26,79 | 4,32 | 1,48 | 3,21 |
| 67 | 15,44 | 16,08 | 1,10 | 41,04 | 4,20 | 0,15 | 4,05 | 0,36 | 0,48 | 0,76 |
| 68 | 15,65 | 12,18 | 1,15 | 49,26 | 10,23 | 2,88 | 7,35 | 3,13 | 3,64 | 2,95 |
| 69 | 15,68 | 8,91 | 1,15 | 40,69 | 17,57 | 13,13 | 4,44 | 3,50 | 6,65 | 1,45 |
| 70 | 15,89 | 9,35 | 0,91 | 37,79 | 32,32 | 19,93 | 12,38 | 7,68 | 12,92 | 0,67 |
| 71 | 15,92 | 14,96 | 0,90 | 43,15 | 44,70 | 2,36 | 42,34 | 4,96 | 3,16 | 4,09 |
| 72 | 15,97 | 11,81 | 1,36 | 57,74 | 12,00 | 0,90 | 11,10 | 2,62 | 1,91 | 2,43 |
| 73 | 16,01 | 7,96 | 1,28 | 52,90 | 17,77 | 8,16 | 9,61 | 4,55 | 7,43 | 2,35 |
| 74 | 16,06 | 9,16 | 1,00 | 38,89 | 14,73 | 2,17 | 12,56 | 2,85 | 1,69 | 1,92 |
| 75 | 16,16 | 31,81 | 0,99 | 28,95 | 0,72 | 0,08 | 0,64 | 0,21 | 0,26 | 0,10 |
| 76 | 16,25 | 13,15 | 1,07 | 46,40 | 10,50 | 1,17 | 9,33 | 2,75 | 1,44 | 2,37 |
| 77 | 16,40 | 13,67 | 0,98 | 44,21 | 18,34 | 5,36 | 12,98 | 5,33 | 11,12 | 0,96 |
| 78 | 16,40 | 9,79 | 1,07 | 45,01 | 16,24 | 11,59 | 4,65 | 4,68 | 7,32 | 1,80 |
| 79 | 16,79 | 6,98 | 1,08 | 41,36 | 22,14 | 5,90 | 16,24 | 2,95 | 2,86 | 1,55 |
| 80 | 16,79 | 13,15 | 1,13 | 51,93 | 15,72 | 1,20 | 14,52 | 3,47 | 1,70 | 2,42 |
| 81 | 16,85 | 6,78 | 1,41 | 52,47 | 16,55 | 11,13 | 5,42 | 1,50 | 0,86 | 0,50 |
| 82 | 17,02 | 15,00 | 0,97 | 43,39 | 3,50 | 1,10 | 2,40 | 1,98 | 2,33 | 2,01 |
| 83 | 17,11 | 12,70 | 1,15 | 66,72 | 7,73 | 7,42 | 0,31 | 2,90 | 5,58 | 1,74 |
| 84 | 17,20 | 14,33 | 1,07 | 46,18 | 21,45 | 1,45 | 20,00 | 2,12 | 1,11 | 2,51 |
| 85 | 17,32 | 11,65 | 1,02 | 41,95 | | | | | | |
| 86 | 17,35 | 10,77 | 0,89 | 43,74 | 7,23 | 5,37 | 1,85 | 3,11 | 4,57 | 1,25 |
| 87 | 17,40 | 9,20 | 1,01 | 44,51 | 22,17 | 7,42 | 14,74 | 5,49 | 8,64 | 2,01 |
| 88 | 17,84 | 10,62 | 0,87 | 41,30 | 22,57 | 10,48 | 12,08 | 5,37 | 9,32 | 2,28 |
| 89 | 17,93 | 11,96 | 0,92 | 70,74 | 7,07 | 1,92 | 5,15 | 2,18 | 1,24 | 2,04 |
| 90 | 18,02 | 10,14 | 1,06 | 54,78 | 9,63 | 3,94 | 5,69 | 1,54 | 1,85 | 1,23 |
| 91 | 18,16 | 7,49 | 0,90 | 38,83 | 18,45 | 13,74 | 4,71 | 6,13 | 12,86 | 1,81 |
| 92 | 18,35 | 7,58 | 1,37 | 55,53 | 23,89 | 21,99 | 1,89 | 4,25 | 8,34 | 0,61 |
| 93 | 18,40 | 17,36 | 1,22 | 64,33 | 2,11 | | 2,11 | 3,88 | | 2,25 |
| 94 | 18,48 | 10,80 | 1,29 | 69,52 | | | | | | |
| 95 | 18,84 | 16,82 | 0,92 | 48,09 | 9,15 | 2,24 | 6,91 | 2,13 | 1,60 | 1,87 |
| 96 | 18,97 | 10,83 | 0,71 | 40,47 | 15,87 | 1,73 | 14,14 | 3,62 | 2,06 | 12,22 |
| 97 | 19,80 | 11,65 | 1,27 | 67,10 | 15,55 | 1,58 | 13,97 | 2,54 | 1,88 | 1,26 |
| 98 | 20,05 | 12,23 | 0,92 | 44,10 | 6,89 | 2,83 | 4,06 | 2,04 | 1,52 | 1,22 |
| 99 | 20,10 | 11,82 | 1,23 | 71,13 | 22,00 | 2,41 | 19,59 | 1,83 | 0,84 | 1,23 |
| 100 | 20,49 | 18,68 | 1,14 | 63,59 | | | | | | |
| 101 | 20,87 | 10,06 | 0,98 | 51,94 | 12,72 | 6,46 | 6,26 | 5,62 | 9,49 | 2,56 |
| 102 | 21,01 | 10,72 | 0,96 | 52,89 | 14,70 | 2,13 | 12,57 | 3,60 | 2,07 | 3,68 |
| 103 | 21,40 | 12,49 | 0,92 | 40,49 | 12,63 | 1,41 | 11,22 | 3,50 | 1,24 | 2,65 |
| 104 | 21,46 | 8,82 | 1,27 | 57,95 | 37,53 | 5,34 | 32,19 | 4,70 | 3,25 | 3,72 |
| 105 | 21,53 | 9,82 | 0,96 | 51,38 | 16,61 | 14,91 | 1,70 | 6,88 | 13,92 | 1,65 |
| 106 | 21,68 | 13,35 | 0,98 | 60,36 | 6,95 | 2,65 | 4,30 | 3,39 | 7,61 | 1,61 |
| 107 | 21,76 | 11,69 | 0,96 | 53,12 | 14,82 | 1,65 | 13,17 | 3,89 | 2,05 | 4,19 |
| 108 | 21,85 | 13,71 | 0,82 | 38,35 | 16,61 | 15,48 | 1,13 | 4,51 | 10,66 | 1,79 |
| 109 | 21,90 | 16,85 | 0,97 | 46,77 | 1,20 | 0,90 | 0,30 | 0,44 | 0,80 | 0,76 |
| 110 | 21,99 | 13,83 | | 48,94 | 16,75 | 3,54 | 13,21 | 3,07 | 1,36 | 2,30 |
| 111 | 22,70 | 11,65 | 0,89 | 52,62 | 7,70 | 2,98 | 4,72 | 2,45 | 1,98 | 1,49 |
| 112 | 22,73 | 13,15 | 0,99 | 63,55 | 19,64 | 10,34 | 9,30 | 9,47 | 37,03 | 1,85 |
| 113 | 22,77 | 6,82 | 1,60 | 80,81 | 17,42 | 11,91 | 5,51 | 1,48 | 0,88 | 0,40 |
| 114 | 22,83 | 10,88 | 1,27 | 69,23 | 10,57 | 8,53 | 2,04 | 1,86 | 4,45 | 0,68 |
| 115 | 23,00 | 15,33 | 0,93 | 58,49 | 11,41 | 2,31 | 9,10 | 2,83 | 1,77 | 1,22 |
| 116 | 23,09 | 9,07 | 1,15 | 78,66 | 23,52 | 7,08 | 16,44 | 1,66 | 1,45 | 0,24 |



| Soil | [C] (mg g$^{-1}$) | CN | $\rho$ (kg dm$^{-3}$) | $\int$C (Mg ha$^{-1}$) | Fe$_d$ | Fe$_o$ | Fe$_d$ –Fe$_o$ | Al$_d$ | Al$_o$ | Al$_p$ |
|---|---|---|---|---|---|---|---|---|---|---|
|  |  |  |  |  |  |  | g kg$^{-1}$ |  |  |  |
| 117 | 23,20 | 13,47 | 0,91 | 56,77 | 9,02 | 2,59 | 6,43 | 2,68 | 1,83 | 1,52 |
| 118 | 23,21 | 12,93 | 0,98 | 60,79 | 26,40 | 2,12 | 24,28 | 5,73 | 1,75 | 4,32 |
| 119 | 23,34 | 12,53 | 0,94 | 55,58 | 13,50 | 1,04 | 12,46 | 3,90 | 2,77 | 3,26 |
| 120 | 23,53 | 11,93 | 1,09 | 57,84 | 19,62 | 0,87 | 18,75 | 4,97 | 2,55 | 3,10 |
| 121 | 23,65 | 12,24 | 0,97 | 59,31 | 9,75 | 2,89 | 6,86 | 3,02 | 1,89 | 2,17 |
| 122 | 24,03 | 10,83 | 1,05 | 60,18 | 12,33 | 4,92 | 7,41 | 5,15 | 7,97 | 2,84 |
| 123 | 24,30 | 22,03 | 1,34 | 3,12 | 0,60 | 0,41 | 0,19 | 0,09 | 0,02 | 0,03 |
| 124 | 24,30 | 11,05 | 1,27 | 74,80 | 16,70 | 1,36 | 15,34 | 0,95 | 1,01 | 0,59 |
| 125 | 24,76 | 9,49 | 1,05 | 68,86 | 21,66 | 6,28 | 15,38 | 1,77 | 1,44 | 0,66 |
| 126 | 25,39 | 15,15 | 0,77 | 52,49 | 14,82 | 1,09 | 13,73 | 3,28 | 2,26 | 1,60 |
| 127 | 25,48 | 16,20 | 0,86 | 67,13 | 21,55 | 2,85 | 18,70 | 3,88 | 2,25 | 4,09 |
| 128 | 25,57 | 11,35 | 1,13 | 77,51 | 36,21 | 1,60 | 34,61 | 7,66 | 3,27 | 2,61 |
| 129 | 25,82 | 10,79 | 0,94 | 64,92 | 58,14 | 2,19 | 55,95 | 9,61 | 2,77 | 2,50 |
| 130 | 25,87 | 17,21 | 1,02 | 70,11 | 8,44 | 3,55 | 4,89 | 4,71 | 11,92 | 1,26 |
| 131 | 26,57 | 12,57 | 0,89 | 61,29 | 9,71 | 3,12 | 6,59 | 3,02 | 1,97 | 2,37 |
| 132 | 26,86 | 9,86 | 1,03 | 76,89 | 53,64 | 2,19 | 51,45 | 9,60 | 1,99 | 1,69 |
| 133 | 27,00 | 11,82 | 0,82 | 55,83 | 8,72 | 3,58 | 5,14 | 3,32 | 2,49 | 2,57 |
| 134 | 27,09 | 11,56 | 0,93 | 64,68 | 7,71 | 2,64 | 5,07 | 2,51 | 1,82 | 1,85 |
| 135 | 28,80 | 9,05 | 0,79 | 45,16 | 10,39 | 9,63 | 0,76 | 3,83 | 8,40 | 2,47 |
| 136 | 30,80 | 15,40 | 1,18 | 95,49 | 67,20 | 2,03 | 65,17 | 5,88 | 1,24 | 1,72 |
| 137 | 30,82 | 10,75 | 0,88 | 81,36 | 21,34 | 12,14 | 9,20 | 7,99 | 31,37 | 2,36 |
| 138 | 32,80 | 14,26 | 0,88 | 74,97 | 3,70 | 1,41 | 2,29 | 2,49 | 4,70 | 2,79 |
| 139 | 41,81 | 20,72 | 0,63 | 53,70 | 0,24 | 0,08 | 0,16 | 0,10 | 0,21 | 0,01 |
| 140 | 46,70 | 16,10 | 1,36 | 115,48 | 21,40 | 3,17 | 18,23 | 3,74 | 3,14 | 2,53 |
| 141 | 49,08 | 13,36 | 1,32 | 166,86 | 20,10 | 2,87 | 17,23 | 10,49 | 2,55 | 1,81 |
| 142 | 60,47 | 11,31 |  |  | 14,50 |  |  | 20,27 |  | 11,69 |
| 143 | 61,44 | 11,77 | 0,60 | 97,92 | 9,02 | 8,34 | 0,67 | 19,53 | 43,52 | 10,26 |
| 144 | 63,43 | 12,51 | 0,43 | 87,72 | 11,14 | 5,25 | 5,89 | 7,36 | 22,54 | 8,61 |
| 145 | 89,26 | 25,82 | 1,58 | 363,55 | 0,36 | 0,34 | 0,02 | 1,34 | 2,75 | 1,18 |
| 146 | 93,06 | 12,50 | 0,89 | 219,25 |  |  |  |  |  |  |
| 147 | 119,82 | 20,79 | 0,34 | 42,19 | 0,90 | 0,10 | 0,80 | 0,45 | 0,54 | 0,27 |



**Table 2. A guide for interpretation of selective dissolution data following Parfait and Childs (1988).**

| Form | description |
|---|---|
| $Fe_d$ | Dissolves almost all iron oxides not differentiating between crystalline and short-range oxides. Provides estimates of total amount of iron oxides in the soil |
| $Fe_o$ | Estimates short range minerals such as ferrihydrite and possibly other amorphous minerals. Do not extract crystalline oxides |
| $Fe_p$ | Extracts a variety of Fe forms, thus it does not specifically relate to any particular form of Fe in soil. Should not be used to estimate Fe-humus complexes |
| $Al_d$ | Probably arises from Al substitution in both crystalline and amorphous oxides, free Al and interlayer Al. Similar to $Fe_d$ it provides wide estimates of Al oxides in the soil. |
| $Al_o$ | Estimates Al in short-range minerals, such as allophane and imogolite. May also represent Al substitution in ferrihydrite and the presence of Al hydroxy interlayer minerals. Do not extract crystalline Al hydroxides. |
| $Al_p$ | Correspond to Al-humus complexes in most soils such as occurring in Podzols and Andosols |
| $Fe_d-Fe_o$ | Provides estimation of crystalline oxides only. Excludes the content of ferrihydrite and other short-range oxides which are extracted by $Fe_o$. |





**Low activity clay soils** (above diagonal) / **All soils combined** (below diagonal)

| | [C] | CN | $T_a$ | $P_a$ | $E_v$ | $D_b$ | sand | clay | silt | pH | $I_E$ | [P]t | TRB | [Fe]d | [Fe]o | [Fe]do | [Al]d | [Al]o | [Al]do |
|---|---|---|---|---|---|---|---|---|---|---|---|---|---|---|---|---|---|---|---|
| [C] | | 0.08 | 0.19 | 0.13 | -0.05 | -0.47 | -0.56 | 0.54 | 0.28 | -0.12 | 0.37 | 0.08 | -0.09 | 0.27 | 0.27 | 0.23 | 0.37 | 0.30 | 0.14 |
| CN | 0.14 | | 0.02 | -0.07 | -0.01 | -0.07 | 0.04 | -0.02 | -0.16 | -0.16 | 0.01 | -0.09 | -0.44 | -0.05 | 0.10 | -0.03 | 0.10 | 0.11 | -0.05 |
| $T_a$ | 0.01 | 0.33 | | 0.16 | -0.52 | -0.21 | -0.27 | 0.27 | 0.10 | -0.28 | -0.06 | -0.29 | -0.21 | -0.14 | 0.03 | -0.18 | -0.24 | -0.16 | -0.04 |
| $P_a$ | 0.06 | -0.08 | -0.10 | | -0.10 | -0.01 | 0.05 | -0.01 | -0.08 | 0.12 | -0.07 | -0.17 | -0.16 | -0.19 | -0.08 | -0.06 | -0.15 | -0.18 | 0.01 |
| $E_v$ | 0.00 | -0.32 | -0.61 | -0.10 | | -0.07 | 0.09 | -0.11 | 0.00 | 0.30 | 0.08 | 0.12 | 0.12 | 0.09 | 0.13 | 0.04 | 0.16 | 0.14 | -0.02 |
| $D_b$ | -0.33 | -0.10 | -0.07 | -0.08 | 0.08 | | 0.48 | -0.48 | -0.26 | 0.09 | -0.32 | 0.00 | 0.05 | -0.20 | -0.31 | -0.11 | -0.20 | -0.18 | -0.06 |
| sand | -0.21 | -0.34 | 0.07 | -0.05 | -0.04 | 0.14 | | -0.87 | -0.46 | 0.22 | -0.41 | -0.16 | -0.07 | -0.37 | -0.32 | -0.21 | -0.33 | -0.28 | -0.10 |
| clay | 0.31 | 0.34 | -0.03 | 0.05 | -0.02 | -0.23 | -0.59 | | 0.33 | -0.22 | 0.38 | 0.16 | 0.04 | 0.36 | 0.27 | 0.22 | 0.31 | 0.27 | 0.10 |
| silt | 0.05 | -0.16 | -0.17 | -0.02 | 0.16 | 0.02 | -0.51 | 0.10 | | -0.10 | 0.30 | 0.16 | 0.21 | 0.25 | 0.35 | 0.08 | 0.28 | 0.26 | 0.03 |
| pH | 0.01 | -0.43 | -0.31 | 0.06 | 0.36 | 0.20 | 0.05 | -0.13 | 0.06 | | -0.06 | 0.07 | 0.17 | -0.16 | -0.04 | -0.14 | 0.03 | 0.04 | -0.10 |
| $I_E$ | 0.13 | -0.21 | -0.31 | -0.03 | 0.30 | -0.03 | -0.40 | 0.25 | 0.42 | 0.19 | | 0.49 | 0.60 | 0.34 | 0.52 | 0.13 | 0.43 | 0.45 | -0.09 |
| [P]t | 0.17 | -0.47 | -0.35 | 0.02 | 0.28 | 0.05 | -0.36 | 0.26 | 0.38 | 0.24 | 0.49 | | 0.60 | 0.41 | -0.09 | 0.37 | 0.36 | 0.17 | 0.20 |
| TRB | 0.02 | -0.48 | -0.39 | -0.02 | 0.36 | 0.09 | -0.34 | 0.17 | 0.47 | 0.31 | 0.60 | 0.60 | | 0.31 | -0.11 | 0.31 | 0.09 | -0.01 | 0.28 |
| [Fe]d | 0.18 | -0.64 | -0.19 | 0.02 | 0.16 | -0.04 | -0.33 | 0.34 | 0.24 | 0.05 | 0.34 | 0.51 | 0.38 | | 0.16 | 0.75 | 0.44 | 0.19 | 0.32 |
| [Fe]o | 0.13 | -0.32 | -0.26 | 0.02 | 0.24 | -0.10 | -0.36 | 0.27 | 0.35 | 0.17 | 0.52 | 0.37 | 0.46 | 0.39 | | 0.05 | 0.40 | 0.53 | -0.36 |
| [Fe]do | 0.12 | -0.41 | -0.04 | -0.03 | 0.05 | 0.02 | -0.20 | 0.27 | 0.06 | -0.08 | 0.10 | 0.31 | 0.15 | 0.66 | 0.05 | | 0.25 | -0.03 | 0.49 |
| [Al]d | 0.28 | -0.15 | -0.25 | 0.03 | 0.12 | -0.24 | -0.22 | 0.35 | 0.10 | 0.00 | 0.28 | 0.32 | 0.21 | 0.49 | 0.43 | 0.32 | | 0.66 | 0.02 |
| [Al]o | 0.19 | -0.19 | -0.29 | 0.04 | 0.20 | -0.20 | -0.19 | 0.30 | 0.10 | 0.08 | 0.37 | 0.23 | 0.26 | 0.25 | 0.56 | 0.01 | 0.62 | | -0.41 |
| [Al]do | 0.05 | 0.08 | 0.18 | -0.04 | -0.14 | 0.03 | -0.04 | 0.05 | -0.03 | -0.19 | -0.17 | 0.03 | -0.09 | 0.18 | -0.31 | 0.43 | -0.03 | -0.41 | |

**High activity clay soils** (above diagonal) / **Arenic soils** (below diagonal)

| | [C] | CN | $T_a$ | $P_a$ | $E_v$ | $D_b$ | sand | clay | silt | pH | $I_E$ | [P]t | TRB | [Fe]d | [Fe]o | [Fe]do | [Al]d | [Al]o | [Al]do |
|---|---|---|---|---|---|---|---|---|---|---|---|---|---|---|---|---|---|---|---|
| [C] | | 0.30 | -0.03 | 0.02 | -0.01 | -0.29 | 0.06 | 0.19 | -0.20 | 0.08 | 0.08 | 0.21 | 0.02 | 0.14 | 0.05 | 0.08 | 0.29 | 0.17 | -0.01 |
| CN | 0.15 | | 0.33 | -0.15 | -0.33 | -0.17 | 0.39 | -0.21 | -0.37 | -0.20 | -0.40 | -0.35 | -0.56 | -0.13 | -0.31 | 0.00 | 0.03 | -0.12 | 0.15 |
| $T_a$ | -0.32 | 0.16 | | -0.13 | -0.70 | -0.05 | 0.08 | -0.11 | -0.04 | -0.33 | -0.28 | -0.32 | -0.41 | -0.02 | -0.21 | 0.06 | -0.20 | -0.27 | 0.23 |
| $P_a$ | 0.26 | -0.33 | -0.40 | | 0.02 | 0.19 | -0.11 | 0.18 | 0.03 | 0.40 | 0.08 | -0.03 | 0.06 | 0.05 | 0.06 | 0.03 | 0.18 | 0.18 | -0.08 |
| $E_v$ | -0.17 | 0.33 | -0.61 | -0.40 | | 0.22 | -0.01 | 0.04 | 0.01 | 0.21 | 0.29 | 0.22 | 0.44 | 0.03 | 0.15 | -0.01 | 0.05 | 0.18 | -0.18 |
| $D_b$ | -0.24 | 0.09 | -0.05 | 0.00 | 0.19 | | -0.07 | -0.10 | 0.16 | -0.01 | 0.04 | 0.07 | 0.14 | -0.18 | -0.08 | 0.10 | -0.33 | -0.30 | 0.15 |
| sand | 0.00 | -0.03 | 0.08 | -0.11 | -0.01 | 0.37 | | -0.43 | -0.63 | -0.04 | -0.28 | -0.30 | -0.31 | 0.26 | -0.19 | -0.06 | 0.02 | 0.03 | -0.03 |
| clay | 0.23 | -0.11 | -0.11 | 0.18 | 0.04 | -0.56 | -0.87 | | 0.05 | 0.01 | 0.21 | 0.32 | 0.19 | 0.03 | 0.18 | 0.13 | 0.20 | 0.18 | -0.06 |
| silt | -0.03 | 0.55 | 0.03 | 0.03 | 0.01 | 0.34 | 0.37 | 0.05 | | -0.12 | 0.25 | 0.13 | 0.30 | 0.08 | 0.11 | -0.02 | -0.15 | -0.12 | 0.05 |
| pH | 0.43 | -0.04 | -0.33 | 0.40 | 0.21 | 0.06 | -0.11 | 0.01 | -0.12 | | 0.32 | 0.32 | 0.43 | 0.21 | 0.22 | -0.05 | -0.02 | 0.09 | -0.17 |
| $I_E$ | 0.24 | -0.45 | -0.28 | 0.08 | 0.29 | -0.47 | -0.45 | 0.21 | 0.25 | -0.25 | | 0.35 | 0.65 | 0.36 | 0.41 | -0.09 | 0.06 | 0.26 | -0.24 |
| [P]t | 0.00 | -0.03 | -0.32 | -0.03 | 0.22 | -0.01 | -0.15 | 0.32 | 0.13 | -0.65 | 0.41 | | 0.51 | 0.16 | 0.25 | 0.21 | 0.19 | 0.12 | 0.00 |
| TRB | 0.26 | -0.16 | -0.41 | 0.06 | 0.44 | 0.48 | 0.20 | 0.19 | 0.30 | 0.09 | 0.51 | 0.27 | | 0.20 | 0.41 | -0.06 | 0.03 | 0.22 | -0.24 |
| [Fe]d | -0.06 | -0.32 | -0.02 | 0.05 | 0.03 | 0.18 | -0.17 | 0.03 | 0.08 | 0.24 | 0.13 | 0.36 | 0.16 | | 0.20 | 0.62 | 0.39 | 0.10 | 0.19 |
| [Fe]o | 0.13 | 0.00 | -0.21 | 0.06 | 0.15 | 0.10 | 0.24 | 0.18 | 0.11 | 0.22 | 0.41 | 0.25 | 0.41 | 0.36 | | -0.18 | 0.33 | 0.58 | -0.46 |
| [Fe]do | 0.15 | 0.00 | 0.06 | 0.03 | -0.01 | 0.01 | -0.12 | 0.13 | -0.02 | -0.05 | 0.22 | 0.19 | -0.06 | 0.21 | -0.45 | | 0.20 | -0.16 | 0.49 |
| [Al]d | 0.27 | -0.03 | -0.20 | 0.18 | 0.05 | 0.16 | 0.13 | 0.20 | -0.15 | -0.02 | 0.00 | 0.12 | 0.03 | 0.38 | 0.14 | 0.21 | | 0.58 | -0.12 |
| [Al]o | -0.18 | -0.06 | -0.27 | 0.18 | 0.18 | -0.23 | -0.16 | 0.18 | -0.12 | 0.09 | 0.04 | -0.24 | 0.16 | 0.20 | -0.02 | 0.45 | 0.48 | | -0.54 |
| [Al]do | -0.06 | -0.18 | 0.23 | -0.08 | -0.18 | 0.15 | -0.03 | -0.06 | 0.05 | -0.17 | -0.24 | 0.00 | -0.24 | 0.19 | 0.14 | -0.30 | -0.21 | -0.73 | |

**Table 3.** Kendall's $\tau$ correlations between a wide range of soil and climate properties potentially involved in differences in soil carbon storage. Four one sided correlation matrices are shown *viz.* for each of the Arenic, LAC and HAC clusters as well as for the (combined) dataset as a whole. Here, with $n > 30$ for the LAC and HAC clusters we have indicated in bold all cases where $\tau > 0.30$ for these two groupings (as well as the combined dataset) with this associating roughly with the probability of Type-II error being less than 0.05. For the Arenic soil cluster with $n = 13$ the equivalent value is $\tau > 0.52$ and in all cases where one or more of the four grouping has $p > 0.05$, we have indicated – using different colours to help cross-referencing across the four diagonal matrices.





**Table 4. Summary of OLS regression coefficients for soil organic carbon and texture associations.**

| | *b* | *s.e.* | *β* | *t* | *p* | *Lower* | *Upper* |
|---|---|---|---|---|---|---|---|
| *a.* LAC soils: $r^2 = 0.57$, $p < 0.001$, $AIC = 292.1$ | | | | | | | |
| intercept | 9.56 | 1.03 | — | 9.31 | 0.000 | 7.50 | 11.62 |
| Clay fraction | 17.91 | 2.15 | 0.762 | 8.32 | 0.000 | 13.60 | 22.24 |
| *b.* LAC soils: $r^2 = 0.61$, $p < 0.001$, $AIC = 288.6$ | | | | | | | |
| intercept | 8.50 | 1.08 | — | 7.84 | 0.000 | 6.32 | 10.68 |
| clay fraction | 16.58 | 2.13 | 0.716 | 7.75 | 0.000 | 12.24 | 20.89 |
| silt fraction | 14.39 | 6.19 | 0.212 | 2.32 | 0.024 | 1.94 | 26.83 |
| *c.* LAC soils: $r^2 = 0.61$, $p < 0.001$, $AIC = 286.7$ | | | | | | | |
| intercept | 8.44 | 1.06 | - | 7.96 | 0.000 | 6.32 | 10.57 |
| (clay + silt) fractions | 16.23 | 1.79 | 0.789 | 9.07 | 0.000 | 12.63 | 19.82 |
| *d.* HAC soils: $r^2 = 0.00$, $p < 0.335$, $AIC = 628.2$ | | | | | | | |
| intercept | 16.16 | 3.21 | — | 5.04 | 0.000 | 9.78 | 22.54 |
| clay fraction | 9.58 | 9.87 | 0.088 | 0.97 | 0.335 | -10.07 | 29.22 |
| *e.* HAC soils: $r^2 = 0.05$, $p < 0.006$, $AIC = 625.3$ | | | | | | | |
| intercept | 21.67 | 4.02 | — | 5.41 | 0.000 | 13.70 | 29.69 |
| clay fraction | 9.26 | 9.64 | 0.088 | 0.96 | 0.340 | -9.94 | 28.44 |
| silt fraction | -16.29 | 7.40 | -0.196 | -2.21 | 0.037 | -31.03 | -1.55 |
| *f.* HAC soils: $r^2 = 0.05$, $p < 0.259$, $AIC = 627.8$ | | | | | | | |
| intercept | 23.36 | 4.03 | — | 5.81 | 0.000 | 15.35 | 31.37 |
| (clay + silt) fractions | -6.87 | 6.04 | -0.103 | -1.14 | 0.259 | -18.90 | 5.16 |
| *g.* Arenic soils: $r^2 = 0.07$, $p < 0.206$, $AIC = 119.92$ | | | | | | | |
| intercept | 8.35 | 14.55 | — | 0.574 | 0.579 | -24.07 | 40.77 |
| clay fraction | 431.39 | 319.17 | 0.352 | 1.352 | 0.206 | -279.75 | 1142.5 |
| *h.* Arenic soils: $r^2 = 0.23$, $p < 0.119$ $AIC = 118.26$ | | | | | | | |
| intercept | -0.38 | 14.04 | — | -0.03 | 0.979 | -32.13 | 31.38 |
| clay fraction | 143.77 | 80.24 | 0.254 | 1.79 | 0.107 | -37.75 | 325.30 |
| silt fraction | 228.66 | 310.22 | 0.254 | 0.74 | 0.480 | -473.18 | 930.39 |
| *i.* Arenic soils: $r^2 = 0.31$, $p < 0.035$ $AIC = 116.34$ | | | | | | | |
| intercept | 1.09 | 12.08 | — | 0.09 | 0.930 | -25.84 | 28.01 |
| (clay + silt) fractions | 154.67 | 63.43 | 0.225 | 2.44 | 0.035 | 13.26 | 296.07 |
| *j.* All soils: $r^2 = 0.01$, $p < 0.13$, $AIC = 1154.3$ | | | | | | | |
| intercept | 16.14 | 1.96 | — | 8.220 | 0.000 | 12.25 | 20.15 |
| clay fraction | 7.98 | 5.23 | 0.106 | 1.524 | 0.130 | -2.37 | 18.32 |
| *k.* All soils: $r^2 = 0.00$, $p < 0.32$, $AIC = 1156.3$ | | | | | | | |
| intercept | 15.96 | 2.43 | — | 6.58 | 0.000 | 11.18 | 20.79 |
| clay fraction | 7.98 | 5.25 | 0.106 | 1.52 | 0.131 | -2.41 | 18.36 |
| silt fraction | 0.68 | 6.01 | 0.007 | 0.10 | 0.917 | -11.25 | 12.51 |
| *l.* All soils: $r^2 = 0.01$, $p < 0.23$, $AIC = 1155.2$ | | | | | | | |
| intercept | 16.01 | 2.43 | - | 6.59 | 0.000 | 11.20 | 20.80 |
| (clay + silt) fractions | 4.80 | 3.96 | 0.084 | 1.21 | 0.228 | -3.03 | 12.63 |





**Table 5. Summary of OLS regression coefficients for soil organic carbon and dithionite extractable Al.**

|  | *b* | *s.e.* | *β* | *t* | *p* | *Lower* | *Upper* |
|---|---|---|---|---|---|---|---|
| *m.* LAC soils: $r^2 = 0.27$, $p < 0.0001$, $AIC = 30.26$ | | | | | | | |
| intercept | 2.36 | 0.100 | — | 23.69 | 0.000 | 2.16 | 2.57 |
| $[Al]_d$ | 0.372 | 0.084 | | 4.39 | 0.000 | 0.201 | 0.542 |
| *n.* HAC soils: $r^2 = 0.23$, $p < 0.0001$, $AIC = 95.83$ | | | | | | | |
| intercept | 2.50 | 0.08 | — | 31.25 | 0.000 | 2.34 | 2.66 |
| $\log [Al]_d$ | 0.300 | 0.060 | | 5.00 | 0.000 | 0.180 | 0.419 |
| *o.* Arenic soils: $r^2 = 0.09$, $p < 0.17$, $AIC = 37.05$ | | | | | | | |
| intercept | 3.42 | 0.433 | - | 7.96 | 0.000 | 2.47 | 4.38 |
| $[Al]_d$ | 0.343 | 0.236 | | 0.17 | 0.174 | -0.176 | 0.863 |
| *p.* All soils: $r^2 = 0.08$, $p < 0.0004$, $AIC = 200.18$ | | | | | | | |
| intercept | 2.69 | 0.052 | | 52.13 | 0.000 | 2.59 | 2.79 |
| $[Al]_d$ | 0.141 | 0.039 | | 3.65 | 0.000 | 0.06 | 0.217 |





**Table 6. Summary of OLS regression coefficients for soil organic carbon in HAC soils.**

|  | $b$ | s.e. | $\beta$ | $t$ | $p$ | Lower | Upper | VIF |
|---|---|---|---|---|---|---|---|---|
| *q.* HAC soils: log[C] (mg g$^{-1}$), $r^2 = 0.32$, $p < 0.001$, $AIC = 78.09$ | | | | | | | | |
| intercept | 1.490 | 0.313 | — | 4.77 | 0.000 | 0.867 | 2.113 | |
| pH | 0.241 | 0.066 | 0.359 | 3.66 | 0.000 | 0.109 | 0.372 | 1.18 |
| log [Al]$_d$ (mg g$^{-1}$) | 0.403 | 0.071 | 0.673 | 5.66 | 0.000 | 0.261 | 0.544 | 1.62 |
| log [Fe]$_o$ (mg g$^{-1}$) | -0.156 | 0.055 | -0.347 | -2.84 | 0.006 | -0.266 | -0.047 | 1.72 |
| *r.* HAC soils: log[C] (mg g$^{-1}$), $r^2 = 0.55$, $p < 0.001$, $AIC = 46.42$ | | | | | | | | |
| intercept | -1.387 | 0.522 | — | -2.56 | 0.010 | -2.429 | -0.344 | |
| pH | 0.262 | 0.054 | 0.399 | 4.91 | 0.000 | 0.155 | 0.368 | 1.18 |
| log [Al]$_d$ (mg g$^{-1}$) | 0.314 | 0.059 | 0.524 | 5.30 | 0.000 | 0.195 | 0.432 | 1.71 |
| log [Fe]$_o$ (mg g$^{-1}$) | -0.010 | 0.050 | -0.018 | -0.20 | 0.844 | -0.110 | 0.090 | 2.19 |
| CN ratio (g g$^{-1}$) | 1.132 | 0.181 | 0.567 | 6.29 | 0.000 | 0.777 | 1.500 | 1.36 |
| *s.* HAC soils: log[C] (mg g$^{-1}$), $r^2 = 0.56$, $p < 0.001$, $AIC = 44.46$ | | | | | | | | |
| intercept | -1.417 | 0.496 | — | -2.85 | 0.006 | -2.406 | -0.426 | |
| pH | 0.259 | 0.050 | 0.395 | 5.12 | 0.000 | 0.158 | 0.359 | 1.08 |
| log [Al]$_d$ (mg g$^{-1}$) | 0.307 | 0.045 | 0.513 | 6.78 | 0.000 | 0.216 | 0.396 | 1.01 |
| CN ratio (g g$^{-1}$) | 1.155 | 0.160 | -0.573 | -7.24 | 0.000 | 0.837 | 1.474 | 1.07 |



**Table 7. Summary of coefficients from OLS regression models for HAC soils. Interactions of soil organic carbon, soil pH, leaf litter lignin content (Л) and dithionite extractable Al.**

| | *b* | *s.e.* | *β* | *t* | *p* | *Lower* | *Upper* | *VIF* |
|---|---|---|---|---|---|---|---|---|
| *t.*  HAC soils: log[C] (mg g⁻¹),  $r^2 = 0.38$, $p < 0.001$, $AIC = 42.37$ | | | | | | | | |
| intercept | 0.887 | 0.482 | — | 1.84 | 0.073 | -0.090 | 1.864 | |
| pH | 0.286 | 0.091 | 0.395 | 3.13 | 0.003 | 0.101 | 0.471 | 1.09 |
| log [Al]$_d$ (mg g⁻¹) | 0.469 | 0.107 | 0.673 | 4.37 | 0.000 | 0.251 | 0.687 | 1.58 |
| log [Fe]$_o$ (mg g⁻¹) | -0.055 | 0.087 | -0.092 | -0.63 | 0.532 | -0.233 | 0.122 | 1.47 |
| *u.*  HAC soils: log[C] (mg g⁻¹),  $r^2 = 0.46$, $p < 0.001$, $AIC = 38.77$ | | | | | | | | |
| intercept | -0.488 | 2.556 | — | -1.91 | 0.064 | -10.07 | 0.300 | |
| pH | 0.318 | 0.087 | 0.449 | 3.62 | 0.000 | 0.140 | 0.496 | 1.12 |
| log [Al]$_d$ (mg g⁻¹) | 0.415 | 0.104 | 0.584 | 3.97 | 0.000 | 0.203 | 0.626 | 1.70 |
| log [Fe]$_o$ (mg g⁻¹) | 0.019 | 0.089 | 0.006 | 0.22 | 0.830 | -0.161 | 0.200 | 1.70 |
| log [Λ] (mg g⁻¹) | 0.942 | 0.410 | 0.341 | 2.29 | 0.027 | 0.109 | 1.774 | 1.20 |
| *v.*  HAC soils: log[C] (mg g⁻¹),  $r^2 = 0.47$, $p < 0.001$, $AIC = 36.83$ | | | | | | | | |
| intercept | -4.676 | 2..340 | — | -2.00 | 0.054 | -9.417 | 0.065 | |
| pH | 0.319 | 0.086 | 0.452 | 3.70 | 0.000 | 0.143 | 0.494 | 1.12 |
| log [Al]$_d$ (mg g⁻¹) | 0.428 | 0.083 | 0.618 | 5.18 | 0.000 | 0.261 | 0.595 | 1.07 |
| log [Λ] (mg g⁻¹) | 0.909 | 0.377 | 0.323 | 2.41 | 0.021 | -0.145 | 1.674 | 1.04 |





**Table 8.** Kendall's τ correlations for soil organic carbon fractions and a range of soil and mineralogical properties. Four one sided correlation matrices are shown *viz.* for each of the Arenic, LAC and HAC clusters as well as for the (combined) dataset as a whole. Data shown here is a subset of our entire dataset ($n > 30$). We have indicated in bold all cases where the probability of Type-II error being less than 0.05. For the entire dataset, a τ > 0.22 is associated to a probability of p < 0.05. For the HAC soil cluster with $n = 13$ the equivalent value is τ > 0.36 and for LAC, with $n = 12$ the value is τ > 0.39. For the Arenic soil cluster, with only $n = 5$, the associated probability of p < 0.05 requires a τ > 0.80. In all cases where one or more of the four grouping has $p > 0.05$, we have indicated – using different colours to help cross-referencing across the four diagonal matrices.

Low Activity Clay Soils

High activity clay soils

Arenic Soils

All soils combined



**Table 9. Mean soil organic carbon stocks (0-30 cm) for 12 RGS examined in this study. Stocks from Batjes, (1996) are also given for comparison.**

| RSG | n | Soil carbon concentration | | Soil carbon stock | | SOTER-LAC estimated soil carbon stock | |
|---|---|---|---|---|---|---|---|
| | | Mean (mg g⁻¹) | C.V. | Mean (t ha⁻¹) | C.V. | Mean (t ha⁻¹) | C.V. |
| Acrisol | 18 | 16.3 | 0.35 | 49.5 | 0.27 | 44.0 | 0.50 |
| Alisol | 20 | 16.6 | 0.28 | 45.6 | 0.27 | 85.7 | 0.42 |
| Arenosol | 6 | 12.3 | 0.23 | 29.6 | 0.31 | 20.7 | 0.50 |
| Cambisol | 19 | 21.3 | 0.63 | 58.9 | 0.39 | 55.9 | 0.61 |
| Ferralsol | 34 | 17.1 | 0.35 | 47.3 | 0.26 | 50.5 | 0.48 |
| Fluvisol | 5 | 21.0 | 0.33 | 54.6 | 0.33 | 34.2 | 0.52 |
| Gleysol | 10 | 24.5 | 1.03 | 70.1 | 0.84 | 67.4 | 0.62 |
| Leptosol | 2 | 32.0 | 0.75 | 115.2 | 0.63 | 51.5 | 0.63 |
| Lixosol | 3 | 21.9 | 0.36 | 65.4 | 0.17 | 38.5 | 0.45 |
| Luvisol | 2 | 15.3 | 0.57 | 43.8 | 0.46 | 46.7 | 0.51 |
| Plinthosol | 18 | 14.2 | 0.40 | 41.1 | 0.44 | 34.0 | 0.48 |
| Podzol | 7 | 48.3 | 0.92 | 98.9 | 1.32 | 54.9 | 0.54 |