# Peer review of "Variations in soil chemical and physical properties explain basin-wide variations in Amazon forest soil carbon densities"

_SOIL, 2019_

## Referee Comment (RC1) · Anonymous Referee #1 · 9 Aug 2019

The paper is the result of a comprehensive study on the soil organic carbon stabilization in Amazon forest soils; by examining the correlations between soil physico-chemical properties and carbon level in soils and their size fractions the authors attempted to understand the mechanisms behind the carbon stabilization under the ecosystem. The sample collection, lab analyses and data statistics employed in the investigation are appropriate, and the findings and conclusions made are well supported by the obtained data. Overall the work shows a positive contribution to our understanding of carbon dynamics in Amazon forests. However, it would increase the readability of the article if some texts can be shortened and simplified. The sentences are unusual long, repetitive and complexing reflecting the authors' personal writing habit, but not the benefit of

broad readers. There are quite few of specific (lot of them are editorial) comments I would like to list in the following and hope they are beneficial to the authors. Line 83-92: kaolinite contains no "-"charges; if any, they could be from impurities of 2:1 clays. Under the acidic pHs of the soils, variable charges are positive too. When you discuss soil minerals' role in chemical interaction, often the coatings of oxidic and hydroxidic components of metals on mineral particles should not be ignored since rarely, pure minerals or "clean" minerals are present in soils. Figure 4. add "a, b, c etc." to the title? Line 402-409: use the mean, range of the variables instead of the graphic feature to make the comparisons. Figure 5 and 6: throughout the text and figures the word "association" and "correlation" are exchangeable. "association" may imply the presence of a physical relationship, but "correlation" is a statistical likeness and may not mean a physical relationship at all. It is expected they are used with discretion. Line 401: r2 = 0.58 in Fig. 5a. Line 433: [C];Ald should be [C]:Ald or [C] vs. Ald. There are other cases too. Table 5: p<0.17 should read p>0.17; and also p<0.004? Figure 9 (line 469-475): "a, b, c" on the graphics and title? Table 6: use "soil C/N ratio" or the symbol. Line 476-480: soil C/N should not be an index of litter quality though a good correction exists with liter quality. Also, C/N ratio is not quite independent variable in this equation, so a greater correlation is no surprising. C/N ratio is numerically related to C level in a sample. Line 493: If VIPF and AIC be used as criteria for evaluation, they should be explained in the Method section. Line 500: Table 6s, not Table 6v; Line 496, Table 6c? Line 546: "on the other hand" be deleted. Line 569: it should read "the C+S . . .. Line 613-615: check your sentence. Line 661: "as shown, Line 662: "that' deleted. Line 664: 0.05-0.89, what is the unit? Line 671: "Clay;C" reads "Clay-C or Clay:C or Clay vs. C". Line 708: delete "also", there are a lot of "also" used in these sentences. Line 745: "is". Line 745-747: do you want to say "low or high"? to clarify, use "P levels" for "these". Line 781-784: re-edit the sentence, I think you want "As . . . , they . . ..; and this . . .. So that MRT of . . ..". Line 824: "hard"? Line 849: re-edit the sentence. It is unnecessarily long. Try to be simple and clear!

---

## Referee Comment (RC2) · Anonymous Referee #2 · 23 Aug 2019

Review of soil-2019-24 Title: Variations in soil chemical and physical properties explain basin-wide variations in Amazon forest soil carbon densities Corresponding Author: Carlos Alberto Quesada

General comments This paper focuses on the controls of SOC stabilization and retention in different soil orders across the Amazon Basin. The dataset is impressively large, and there is value in the findings. However, significant changes to the writing need to be addressed. The paper is too long and the sentences are often poorly constructed, burying the amazing findings as a result! A heavy revision and resubmission of the paper is recommended in order to condense the paper, make the writing more

straightforward and direct, and to address a few important questions addressed below. There are also too many figures with too many colors and shapes, which both dilutes the impact of individual graphs and muddles the findings. Lastly, this is an incredible amount of work for a poorly studied region in terms of its soils, and this paper would benefit from emphasizing this fact!

Specific comments/questions: -Introduction is far too long and spends too much time on general soil chemistry processes (i.e. lines 70-124). Much of this is background knowledge (i.e. line 105 "The extent to which DOM precipitates is largely influenced by soil pH"). This is not a process-based paper but a broad geographic survey of soil carbon concentrations, and the introduction should reflect the breadth and focus of the paper. -Be clear about what work was completed in this paper. Parts of the methods and results include references to previous work, which should belong in the discussion section (Line 295: "based on a previous analysis of a subset of sites"). -As previously stated, the biggest issue is the writing. The paper is far too long and the grammar construction is too verbose and confusing. Additionally, the Results section needs to be heavily revised. Rather than saying "Figure X shows...", state the findings and put the figure at the end. I.e. for Section 3.2 (Line 316-319), the length of these sentences could be halved if the writing was instead: "Mineralogy of LAC and HAC soils were distinctly different based on PCA analysis (Figure 3a). -The 2nd biggest issue is this work heavily relies on a previous paper's work. In the methods, Section 2.2 (Line 179), the author writes as if the soil classification was performed in this study. However, in Section 3.1 (Line 294), it turns out this clustering was performed by a previous study. -Terminology needs to be consistent and clear. Use specific p-values. Do not switch between "less weathered" and "low activity clays" to define for instance the 1st group. Those two descriptors do not mean the same thing. -Some QA/QC reporting for the soil density fractionation is needed. Why that particular subset of samples? And what was the recovery of soil mass and soil carbon? Were there reps? SDF analysis is often coupled with a reporting of sample recovery in the Appendix. -Remove the analysis of temperature and precipitation controls. The paper is stronger

without it. Temperature is not highly variable in this region. Rather precipitation is. However, the focus on the paper is on the mineralogical controls of SOC preservation and stabilization. Precipitation and temperature can be brought up in the discussion, especially for the Arenic soils. -Be upfront about the number of soils in each group. It appears the numbers are not consistent based on Fig. 1. How is this accounted for statistically to ensure against bias. -Lastly, land-use change is an equally important threat to this region!! What is the situation in terms of deforestation and how is that expected to change in the future? Consider this alongside climate change in both the intro and discussion.

Technical corrections/Line-by-line comments: L25 – are you sure all of these soils are "pristine"? "Minimally disturbed" might be more appropriate.

L31-33 – Omit beginning part of sentence "SOC fractionation studies further showed that..." – let the focus of the sentence be on the findings

L36-38 – run-on sentence and complicated phrasing "and with this mechanism enhanced by." Sentences need to be more straightforward and direct.

L24-45: Title uses the word densities, but abstract focuses on soil carbon concentrations and soil carbon stocks. Keep language consistent

L45: Rather than end on a negative result ("aboveground biomass nor precipitation...were found to exert any influence..."), end on a more powerful positive finding... i.e. soil and litter qualities are more powerful predictors!

L49-60: What is the focus of this paragraph? This is a list of facts that don't work together towards an overall point such as soil carbon stocks being equally as important as biomass carbon in Amazon forests.

L70 & throughout the paper: Try to use a more active voice than passive. I.e. "clay mineralogy controls specific surface area" in line 70.

L79-80: Mention anion exchange capacity

L73 and L76: Be consistent in your terminology – "Hydrous Fe and Al oxides" vs "Iron and Al hydrous oxides" – pick one and refer to it consistently

L122: Do not assume reader will know why Fe-associated co-precipitation is considerably less important than Al precipitation

L126: Edaphic factors involve physical (temperature, precipitation) and biological factors, but the previous paragraphs have focused on chemical factors. There is a large leap from chemical stabilization to all edaphic factors without an explanation for why.

L132: Wade et al. 2018 is a more recent citation.

L140: Be precise, not vauge. What does "more similar weathering levels" mean, and how is that different from chemical and mineralogical characteristics? Aren't those two descriptors collinear?

L142-143: Instead of "less weathered," can you say more rich in X, Y, Z?

151: Omit "may"

154: Be succinct! Instead, say "Here we explore the climatic. . ..

158: Make the main action verb ("associated") more clear

170: "Usually" is not a scientific word! Please omit

164-176: Study site needs more information. How many different soil orders were covered by these 147 plots? This is a big omission that should not be left to a table or figure but succinctly described in the paragraph. Additionally, line 142 has information that should be included in the section under study site not in the introduction. Bring the reader up to speed. How accurate are the classifications for this region as well?

165: "Primary forest plots" means something different than "Pristine forest plots" (Line 25). Be consistent

176: WHY are only the top 30 cm reported?? This needs to be explained and the entire

paper needs to better reflect this. For instance, mention this depth in the Abstract.

212: How was this subset of sites chosen to ensure a representative subsample?

232: Omit the sentence about leaf litter lignin estimates. That should be included in section 2.1 when the previous studies are mentioned.

259-271: Was it bulk soil XRD (<2mm) and the clay fraction, or just bulk soil?

275: What does SRTM stand for?

294: Begin the results section with the findings, not a description of a figure. It is confusing whether this cluster analysis was completed in this study or in previous work.

298: Need to define low activity clays and high activity clays in the Introduction – this seems to be an important part of the paper

306-313: What were the ranges in CEC and clay %? Were the average values for these 3 groups significantly different?

315: For the mineralogy, it is important to know what fraction of these soils are clay-sized

330: Be precise, what does varying proportions mean?

343: Give exact p-value

348-350: Confusing sentence, I don't understand it at all.

355: An appropriate way to demonstrate higher variability is to give the coefficient of variation for both groups

378: Instead of all the acronyms, which are hard to keep track of, can you say cation exchange capacity instead of IE?

403-406: Knowing the clay fraction of these two soil groups is essential to interpreting the data

430: 3.5 Soil carbon/mineralogical associations is not an appropriate subtitle when the association between leaf litter and C storage is explored

534: Is 0.49 a percentage, a mass??

656: The discussion subheadings need to be consistent. I.e. if you use low activity clays for one heading, use high activity clays rather than less weathered soils for the following heading. Also, why the change to "retention" for 4.1.2?

661: Rather than lumping other studies into the same sentence as your results, separate their findings into a different sentence.

690-691: I don't understand why the sand and aggregate fraction is lumped together. In soil density fractionation studies, it is often the light fraction (free organic matter), occluded light fraction (organic matter bound up in aggregates), and the dense mineral fraction (residual organic matter bound to minerals). This latter group can be subdivided into clay+silt, and then sand. But to clump sand and aggregate fraction together makes no sense.

767: I would be very careful with such a statement. I'm sure this process of Al/OM interactions has been studied elsewhere. Focus instead on your findings, which are interesting in and of themselves! 833-834: Do you mean biomass carbon inputs impacting soil carbon stocks? It feels like some key adjectives are missing here.

838-840: This is a classic example of this paper's poor writing. Be direct, be concise! For instance, "Our findings do not negate the possibility that future climatic changes will have a significant impact on soil carbon stocks in the Amazon Basin."

Fig. 1 – This is not a systemized, randomized design across the Amazon Basin. These plots are clustered by geographic area. I think a more appropriate statistical approach would be to cluster these sites by location and compare between groups. This is essential for considering the effects of precipitation and temperature. Within and between group variability needs to be addressed.

Fig 11. – I do not understand this graph at all, which is unfortunate because I can tell it has important results!

[Figure]

---

## Editor Comment (EC1) · Elizabeth Bach (Editor) · 23 Aug 2019

Dear Quesada et al., Two peers have reviewed your manuscript "Variations in the soil chemical and physical properties explain basin-wide variations in Amazon forest soil carbon densities." I agree with the reviewers that the dataset presented here is an important contribution to scientific knowledge of soil carbon across a poorly-understood and globally important region, the Amazon forest. However, both reviewers felt the writing style and figures could be improved for clarity and conciseness. The reviewers have provided constructive feedback, which I think will help readers understand the important key findings of this work. I encourage an extensive revision of the manuscript

to address these concerns.

---

## Author Comment (AC1) · 29 Sep 2019

We thank the reviewer for her/his comments, which were more related to style and structure of the manuscript. We are incorporating several editorial suggestions made and believe it will help to improve the paper.

Here we respond each comment separately:

General comments: Comments on manuscript length and readability:

Referee 1: "However, it would increase the readability of the article if some texts can be shortened and simplified. The sentences are unusual long, repetitive and complexing

reflecting the authors' personal writing habit, but not the benefit of paper broad readers"

Response: We have made efforts to make the text more direct (especially in terms of breaking up and shortening sentences in the Results Section) and have also attended the specific points raised by both reviewers. However, we do not agree with the view that short papers are necessarily the best or only way to communicate results. In our view, a detailed and comprehensive analysis, when well supported by data, becomes a definitive reference and thus reaches out to broad audiences. Short papers can be good, but not necessarily are, the opposite also sometimes being the case. As noticed by Reviewer 1, our personal style is somewhat 'expansive', which so far has led us to stablish a successful publishing record. For instance, previous work published on soils of the Amazon by the lead author in Biogeosciences (Quesada et al. 2010, 2011, 2012) has had an even greater number of final pages published, but this did not stop these papers to receive over 300 citations each. We feel that the points raised by the reviewers have indeed helped to make the paper clearer and we have made changes to break up longer sentences and phrases into smaller units , but hope that there is no need for further shortening of the manuscript just for the sake of it.

Specific comments from Referee 1:

"Line 83- 92: kaolinite contains no -charges; if any, they could be from impurities of 2:1 clays. Under the acidic pHs of the soils, variable charges are positive too. When you discuss soil minerals' role in chemical interaction, often the coatings of oxidic and hydroxidic components of metals on mineral particles should not be ignored since rarely, pure minerals or "clean" minerals are present in soils." Response: We accept the comment and have changed the text accordingly.

Figure 4. add "a, b, c etc." to the title? Response: Done

Line 402-409: use the mean, range of the variables instead of the graphic feature to make the comparisons. Response: Means and ranges are already shown in Figure 4. We would, thus prefer to leave this as is.

Figure 5 and 6: throughout the text and figures the word "association" and "correlation" are exchangeable. "association" may imply the presence of a physical relationship, but "correlation" is a statistical likeness and may not mean a physical relationship at all. It is expected they are used with discretion. Response: We accept the comment and have changed the text accordingly

Line 401: r2 = 0.58 in Fig. 5a. Response: Done

Line 433: [C];Ald should be [C]:Ald or [C] vs. Ald. There are other cases too. Response: all have been changed to the format [C] vs. [Al]d (etc.) in this and other cases throughout the text. Table 5: p > 0.17; and also p<0.0004 : Response: Just because a probability of type II error is greater than p = 0.05 does not mean the sign needs to be changed. Nevertheless, we do accept that p= 0.17 would be the more appropriate way to represent the lack of significance in this case and have changed the text accordingly. There was indeed an extra '0' in the quoted probability for the equation of Table 5p and we have now fixed this.

Figure 9 (line 469-475): "a, b, c" on the graphics and title? Response: Done

Table 6: use "soil C/N ratio" or the symbol. Response: Done

Line 476-480: soil C/N should not be an index of litter quality though a good correction exists with liter quality. Also, C/N ratio is not quite independent variable in this equation, so a greater correlation is no surprising. C/N ratio is numerically related to C level in a sample. Response: The text states clearly that CN ratio was used as a surrogate for litter quality with support from data on litter lignin concentrations. We have also added an extra paragraph at the end of Section 3.5 to fully reflect this (along with the possible errors in model predictions associated with the inevitable correlation of [C] with CN ratio).

Line 493: If VIF and AIC be used as criteria for evaluation, they should be explained in the Method section. Response: Done

Line 500: Table 6s, not Table 6v; Response: Done, thanks for spotting the mistake.

Line 496, Table 6c? Response: Done, thanks for spotting the mistake.

Line 546: "on the other hand" be deleted. Response: Done

Line 569: it should read "the C+S . . .. Response: Done

Line 613-615: check your sentence. Response: Done, text has been updated for clarity.

Line 661: "as shown, Response: Done

Line 662: "that' deleted. Response: Done

Line 664: 0.05-0.89, what is the unit? Response: This is a fraction, varying from 0 to 1. Text has been adjusted to make it clear.

Line 671: "Clay;C" reads "Clay-C or Clay:C or Clay vs. C". Response: Done, changed throughout the paper.

Line 708: delete "also", there are a lot of "also" used in these sentences. Response: Done, the sentence was adjusted for clarity.

Line 745: "is". Response: Done

Line 745-747: do you want to say "low or high"? to clarify, use "P levels" for "these". Response: Done, text was adjusted for clarity.

Line 781-784: re-edit the sentence, I think you want "As . . . , they . . ..; and this . . .. So that MRT of . . ..". Response: Done, text was adjusted for clarity.

Line 824: "hard"? Response: Done, text was adjusted and word "hard" was substituted.

Line 849: re-edit the sentence. It is unnecessarily long. Response: Done, text was adjusted for clarity.

---

## Author Comment (AC2) · 29 Sep 2019

We thank the referee for the comments. Most comments were dedicated to style and structure of the manuscript. We are working to incorporate the comments in the best way possible to improve the manuscript readability.

Here we respond to each comment separately:

General comments: Comments on manuscript length and readability:

Referee 2: "A heavy revision and resubmission of the paper is recommended in order to condense the paper, make the writing more straightforward and direct, and to address

a few important questions addressed below."

Response: We have made efforts to make the text more direct (especially in terms of breaking up and shortening sentences in the Results Section) and have also attended the specific points raised by both reviewers. However, we do not agree with the view that short papers are necessarily the best or only way to communicate results. In our view, a detailed and comprehensive analysis, when well supported by data, becomes a definitive reference and thus reaches out to broad audiences. Short papers can be good, but not necessarily are, the opposite also sometimes being the case. As noticed by Reviewer 1, our personal style is somewhat 'expansive', which so far has led us to stablish a successful publishing record. For instance, previous work published on soils of the Amazon by the lead author in Biogeosciences (Quesada et al. 2010, 2011, 2012) has had an even greater number of final pages published, but this did not stop these papers to receive over 300 citations each. We feel that the points raised by the reviewers have indeed helped to make the paper clearer and we have made changes to break up longer sentences and phrases into smaller units , but hope that there is no need for further shortening of the manuscript just for the sake of it.

Referee 2: "There are also too many figures with too many colors and shapes, which both dilutes the impact of individual graphs and muddles the findings"

Response: We respectfully disagree. Our figures follow the same style and standards of our previous papers on the soils as references above, to which the manuscript presented here is somewhat a continuation of such There is no actual gain in simplify the "too many colors and shapes". In truth, we believe just the opposite. Such colors and shapes reflect individual soil types and thus allow the reader to understand that pedogenetic level is an important factor in controlling SOC. This adds to the impact of the paper since readers can relate the results to their own soils of interest, and also help them to perceive the importance of common chemical and morphological characteristics to shape SOC.

Specific comments from Reviewer 2:

"Introduction is far too long and spends too much time on general soil chemistry processes (i.e. lines 70-124). Much of this is background knowledge (i.e. line 105 "The extent to which DOM precipitates is largely influenced by soil pH"). This is not a process-based paper but a broad geographic survey of soil carbon concentrations, and the introduction should reflect the breadth and focus of the paper".

Response: We have shortened the introduction. However, giving the general interest in the Amazon forest from researchers with a wide range of scientific backgrounds, we purposely aimed to provide a text that communicates to broader audiences and therefore it seems important to keep some background information to aid the understanding of our results. Thus we believe that everything remaining in the introduction is relevant to facilitate the understanding and interpretation of the data. Also, given that this is the first time that Al/OM interactions is reported in Amazonia, we feel that background information in the chemistry of this process may be really important for scientists working on that region.

"Be clear about what work was completed in this paper. Parts of the methods and results include references to previous work, which should belong in the discussion section (Line 295: "based on a previous analysis of a subset of sites")." Response: We thank the referee for spotting this and have clarified all issues on the text.

"Results section needs to be heavily revised. Rather than saying "Figure X shows. . .", state the findings and put the figure at the end. I.e. for Section 3.2 (Line 316-319), the length of these sentences could be halved if the writing was instead: "Mineralogy of LAC and HAC soils were distinctly different based on PCA analysis (Figure 3a). Response: This has been changed and text is now more direct

"The 2nd biggest issue is this work heavily relies on a previous paper's work. In the methods, Section 2.2 (Line 179), the author writes as if the soil classification was performed in this study. However, in Section 3.1 (Line 294), it turns out this clustering

was performed by a previous study." Response: We thank the referee for spotting this confusion. Soil classifications and clustering were performed in this study and the text has been changed to make it clearer.

"Terminology needs to be consistent and clear. Use specific p-values. Do not switch between "less weathered" and "low activity clays" to define for instance the 1st group. Those two descriptors do not mean the same thing." Response: We accept the comment regarding terminology and have changed the text accordingly. Where we consider p values critical to the arguments presented they have been specifically presented in the text.

"Some QA/QC reporting for the soil density fractionation is needed. Why that particular subset of samples? And what was the recovery of soil mass and soil carbon? Were there reps? SDF analysis is often coupled with a reporting of sample recovery in the Appendix." Response: Details on the selection of subset of study sites and the C recovery after fractionation have been added to the Materials and Methods.

"Remove the analysis of temperature and precipitation controls. The paper is stronger without it. Temperature is not highly variable in this region. Rather precipitation is. However, the focus on the paper is on the mineralogical controls of SOC preservation and stabilization. Precipitation and temperature can be brought up in the discussion, especially for the Arenic soils." Response: Apart from the Tables of Kendall's $\tau$ do not undertake any analysis of effects of temperature and precipitation. Rather we simply test for any model biases as affected by climate by looking at regression residuals in the Appendix, which we believe is appropriate.

"Be upfront about the number of soils in each group. It appears the numbers are not consistent based on Fig. 1. How is this accounted for statistically to ensure against bias." Response: Table 1 shows clearly the soil types for each group. Our statistical approach does not require a balanced design, but when this had to be taken in consideration, i.e. with Kendall taus, the number of observations and its influence on

significance tests has been very clearly stated.

"Lastly, land-use change is an equally important threat to this region!! What is the situation in terms of deforestation and how is that expected to change in the future? Consider this alongside climate change in both the intro and discussion." Response: We consider that this is a topic beyond the scope of this paper and, with the paper already being longer than is typically the case, we have purposely avoided discussing what we consider to essentially be diversions from the main theme of the manuscript.

L25 – are you sure all of these soils are "pristine"? "Minimally disturbed" might be more appropriate. Response: this has been corrected accordingly.

L31-33 – Omit beginning part of sentence "SOC fractionation studies further showed that. . ." – let the focus of the sentence be on the findings Response: Done

L36-38 – run-on sentence and complicated phrasing "and with this mechanism enhanced by." Sentences need to be more straightforward and direct. Response: We accept the comment and have changed the text accordingly

L24-45: Title uses the word densities, but abstract focuses on soil carbon concentrations and soil carbon stocks. Keep language consistent Response: We accept the comment and have changed the text/title accordingly

L45: Rather than end on a negative result ("aboveground biomass nor precipitation. . .were found to exert any influence. . ."), end on a more powerful positive finding. . . i.e. soil and litter qualities are more powerful predictors! Response: Again, this a matter of style and we are happy to leave this as is.

L49-60: What is the focus of this paragraph? This is a list of facts that don't work together towards an overall point such as soil carbon stocks being equally as important as biomass carbon in Amazon forests. Response: We accept the comment and have deleted this paragraph for shortening the text

L70 & throughout the paper: Try to use a more active voice than passive. I.e. "clay

mineralogy controls specific surface area" in line 70. Response: Again, is nothing more than a matter of personal style and we see no reason to change anything in this respect.

L79-80: Mention anion exchange capacity. Response: Done

L73 and L76: Be consistent in your terminology – "Hydrous Fe and Al oxides" vs "Iron and Al hydrous oxides" – pick one and refer to it consistently Response: Done

L122: Do not assume reader will know why Fe-associated co-precipitation is considerably less important than Al precipitation Response: We have adjusted the text accordingly

L126: Edaphic factors involve physical (temperature, precipitation) and biological factors, but the previous paragraphs have focused on chemical factors. There is a large leap from chemical stabilization to all edaphic factors without an explanation for why. Response: We typically use the word 'edaphic' in accordance with its dictionary definition viz. "of, produced by, or influenced by the soil" and by our reckoning this does not include the current day climate. Although, of course, we accept the critical role of climate in pedogenesis sensu Jenny. The text has been changed to avoid misunderstandings.

L132: Wade et al. 2018 is a more recent citation. Response: We have looked for this reference and two articles could be found for Wade et al. 2018, but none seemed relevant. As the full reference was not given by the referee it is possible that we did not find the right one.

L140: Be precise, not vague. What does "more similar weathering levels" mean, and how is that different from chemical and mineralogical characteristics? Aren't those two descriptors collinear? Response: We have removed he 'weathering levels and/or'

L142-143: Instead of "less weathered," can you say more rich in X, Y, Z? Response: Bearing our sometimes non-specialist audience in mind we have simply changed this

to 'generally younger soils'

151: Omit "may" Response: Done

154: Be succinct! Instead, say "Here we explore the climatic. . ... Response: Done

158: Make the main action verb ("associated") more clear Response: Done

170: "Usually" is not a scientific word! Please omit Response: Done

164-176: Study site needs more information. How many different soil orders were covered by these 147 plots? This is a big omission that should not be left to a table or figure but succinctly described in the paragraph. Additionally, line 142 has information that should be included in the section under study site not in the introduction. Bring the reader up to speed. How accurate are the classifications for this region as well? Response: Indeed, we thank the referee for spotting this. There are 14 soil orders and this information was added to section 2.1. We take the opportunity to clarify that soil classification was performed as part of this study.

165: "Primary forest plots" means something different than "Pristine forest plots" (Line 25). Be consistent Response: Done, it was corrected accordingly

176: WHY are only the top 30 cm reported?? This needs to be explained and the entire paper needs to better reflect this. For instance, mention this depth in the Abstract. Response: Done, this was added to the abstract and explanation for the reason added to material and methods.

212: How was this subset of sites chosen to ensure a representative subsample? Response: This information was added to material and methods.

232: Omit the sentence about leaf litter lignin estimates. That should be included in section 2.1 when the previous studies are mentioned. Response: Indeed it was out of place, but rather than 2.1 we decided to move it to the section 2.3.1, which deals with chemical analysis.

259-271: Was it bulk soil XRD (2 mm) and the clay fraction, or just bulk soil? Response: Bulk soil. This was made clear in the methods.

275: What does SRTM stand for? Response: This was made clear in the methods.

294: Begin the results section with the findings, not a description of a figure. It is confusing whether this cluster analysis was completed in this study or in previous work. Response: Done, things were made clear in the text

298: Need to define low activity clays and high activity clays in the Introduction – this seems to be an important part of the paper Response: Indeed. This was made clear in the introduction

306-313: What were the ranges in CEC and clay %? Were the average values for these 3 groups significantly different? Response: This comes later in the text

315: For the mineralogy, it is important to know what fraction of these soils are clay sized. Response: we are not at all sure as to what the Referee wished us to address here.

330: Be precise, what does varying proportions mean? Response: Means that the proportion of sand varies.

343: Give exact p-value(s) Response: In our view given the many comparisons involved in Fig 4 this would only serve to unnecessarily clutter the text accompanying what is essentially a 'background setting the scene descriptive diagram"

348-350: Confusing sentence, I don't understand it at all. Response: Text has been improved for clarity

355: An appropriate way to demonstrate higher variability is to give the coefficient of variation for both groups Response: We believe that in this particular case a simple 'eyeballing' of Fig. 2 is all that is required for the different variabilities to be appreciated.

378: Instead of all the acronyms, which are hard to keep track of, can you say cation

exchange capacity instead of IE? Response: Done

403-406: Knowing the clay fraction of these two soil groups is essential to interpreting the data. Response: The different clay fractions of the different groups is shown in the proceeding Fig. 4 and with clay fractions also being the independent variable in all three panels. Thus we are not at all sure as to what the Referee wished us to address here.

430: 3.5 Soil carbon/mineralogical associations is not an appropriate subtitle when the association between leaf litter and C storage is explored: Response: Correct, subtitle has been changed

534: Is 0.49 a percentage, a mass?? Response: It is a fraction (values vary between 0 and 1), text has been adjusted to make it clearer

656: The discussion subheadings need to be consistent. I.e. if you use low activity clays for one heading, use high activity clays rather than less weathered soils for the following heading. Also, why the change to "retention" for 4.1.2? Response: Correct, subtitles have been changed

661: Rather than lumping other studies into the same sentence as your results, separate their findings into a different sentence. Response: We see no issue and have chosen to leave as is.

690-691: I don't understand why the sand and aggregate fraction is lumped together. In soil density fractionation studies, it is often the light fraction (free organic matter), occluded light fraction (organic matter bound up in aggregates), and the dense mineral fraction (residual organic matter bound to minerals). This latter group can be subdivided into clay+silt, and then sand. But to clump sand and aggregate fraction together makes no sense. Response: This is a consequence of the methodology used. Zimmermann et al. 2007 initially separates DOC, clay and silt, from the remaining coarse particles (>63 $\mu$m). This coarse fraction contains POM, sand and aggregates that have

not break down in sonication. POM is later separated by density. As many tropical soils have very strong aggregates, such coarse fraction often will have clay aggregates remaining. This issue was made clear in the methods.

767: I would be very careful with such a statement. I'm sure this process of Al/OM interactions has been studied elsewhere. Focus instead on your findings, which are interesting in and of themselves! Response: We are not aware of any study that has reported Al/OM interactions in forest soils of Amazonia, with the possible exemption of particular studies on podzolization, which is a completely different matter.

833-834: Do you mean biomass carbon inputs impacting soil carbon stocks? It feels like some key adjectives are missing here. Response: Text has been adjusted for clarity

838-840: This is a classic example of this paper's poor writing. Be direct, be concise! For instance, "Our findings do not negate the possibility that future climatic changes will have a significant impact on soil carbon stocks in the Amazon Basin." Response: Text has been adjusted for clarity

Fig. 1 – This is not a systemized, randomized design across the Amazon Basin. These plots are clustered by geographic area. I think a more appropriate statistical approach would be to cluster these sites by location and compare between groups. This is essential for considering the effects of precipitation and temperature. Within and between group variability needs to be addressed. Response: Given its' geological and climatic complexity, a systemized, randomized sampling design across the Amazon Basin would be, for all practical purposes impossible. We agree there is some geographical clustering of sites, but lumping these together for any statistical analysis would not work as in many cases there are (purposely) different soil groups (and considerable variability within the groups themselves) at each geographical location. We do accept of course that there is a lack of independence between clustered sites in terms of precipitation regimes, but as precipitation is not a predictor variable in the

models presented this is not an issue in our view. The residual plots of Figure A1 do not suggest systematic biases according to precipitation regime in any case.

Fig 11. – I do not understand this graph at all, which is unfortunate because I can tell it has important results! Response: The y axis legend was adjusted and the figure caption expanded. We hope it is clear now!!

─────────────────────────────────

---

## Author Comment (AC3) · 29 Sep 2019

Dear Dr. Bach,

We have now responded to comments of both referees which were essentially directed to style and structure of the manuscript. No essencial problem has been found with the science itself. We thank the referees for their appreciation of the work. As outlined extensively in the responses for the individual referees, we have considered all comments and we are working to incorporate them in the manuscript, which we believe will have its readability much improved after corrections.

[Figure]

If you agree we will send the revised version along with the above lines later this week.

Best wishes

C.A. Quesada and co-authors

---

## Editor Comment (EC2) · Elizabeth Bach (Editor) · 1 Oct 2019

Dear Dr. Quesada and colleagues,

Thank you for addressing the revisions suggested by the reviewers. Please proceed with uploading the revised manuscript so that the reviewers and I can see the revisions in context.

Thank you, Elizabeth Bach
* * *

---

## Author Comment (AC4) · 5 Oct 2019

[revised manuscript text omitted]

-non-parametric Kruskal-Wallis test, considerable overlap existed in LAC and HAC  for all seven soil chemical properties presented in Fig. 4.

In terms of soil texture, as would reasonably be expected, $\Phi_{sand}$ was significantly higher at $p < 0.05$

for the Arenic versus LAC and/or HAC clusters (Fig. 4i). As would be expected, we also observed significantly lower $\Phi_{clay}$ for the Arenic soils ($p > 0.05$ Fig. 4j). On the other hand, there was no difference between $\Phi_{silt}$ for the Arenic *vs*. LAC soils, both of which, in turn, had a significantly lower

$\Phi_{silt}$ than the soils of the HAC cluster ($p < 0.05$; Fig. 4k). As is also evident from Fig. 2, there was much more variation in $\Phi_{clay}$ for the LAC soils compared  to the HAC soils.

Using Kendall's $\tau$ as a non-parametric measure of association, correlations between a wide range of soil and climate properties potentially involved in differences in soil carbon storage are shown in Table 3.

This  takes the form of four one-sided correlation matrices *viz*. one half-triangle for each of the Arenic,

LAC and HAC clusters as well as for the (combined) dataset as a whole. Here, with $n > 30$ for the LAC and

HAC clusters we have indicated in bold all cases where $\tau > 0.30$ for these two groupings (as well as the combined dataset) with this associating roughly with the probability of Type-II error being less than 0.05.

For the Arenic soil cluster with $n = 13$ the equivalent value is $\tau > 0.52$ and where one or more of the four groupings has $p > 0.05$, this has been indicated for all four matrices using different colours to help cross- referencing across the four diagonal matrices

Table 3 shows that, whilst there are many correlations which are significant at $p = 0.05$ or better, in only in a few case are there significant correlations found for the same bivariate combinations in two or more of the three soil clusters and/or when the three clusters are considered together. For example, although there is clear association between soil texture and soil carbon density for the

LAC soils ($\tau = -0.56$ and $\tau = 0.54$ for $\Phi_{sand}$ and $\Phi_{clay}$ respectively), this is not the case for the HAC soils ($\tau =$

0.06 and $\tau = 0.19$)~~.~-~~and with the association also being much less clear for the Arenic grouping ($\tau = -0.17$ and

$\tau = -0.24$). Consequently, when all three soil clusters are considered together we find $\tau$ of only -0.21 and 0.31

for $\Phi_{sand}$ and $\Phi_{clay}$. That is to say, when all soils are considered together there is much weaker association between soil carbon concentration and soil texture than when LAC soils are considered on their own.

This is also the case for the relationship between [C] and soil bulk density, $D_b$, for which we find $\tau = -0.47$

for LAC soils but markedly lower values for the HAC and Arenic soils ($\tau = -0.29$ and $\tau = -0.17$ respectively), as well as for the combined dataset ($\tau = -0.33$).

In a similar vein, although a high cation exchange capacity ($I_E$) is clearly associated with a high

[C] for LAC soils ($\tau = 0.37$) and perhaps the Arenic soils as well ($\tau = 0.43$), for the HAC soils we find a $\tau$ of only -0.08 for the [C]  $I_E$ . Not surprisingly then, for  the dataset as a whole τ

0.13 for the  $I_E$ vs. [C] correlation.

On the other hand (simple physically based bivariate correlations  such as $T_a$ vs. $E_v$ aside)

there are cases where the strength of the bivariate associations seems to be consistent across all three soil groups. For example, taking the relationship between total phosphorus, $[P]_t$, and mean annual air temperature,

$T_a$, shows τ = -0.29, τ = -0.32 and τ = -0.22 for the LAC, HAC and Arenic soils respectively and with the combined dataset yielding τ = -0.35.

A second example  is the relationship between dithionite extractable aluminium $[Al]_d$ and

$Φ_{clay}$ for which we find τ = 0.31 for LAC soils, τ = 0.20 for HAC soils and τ = 0.36 for Arenic soils and with

τ = 0.35 for the dataset as a whole. Although we found  many correlations between the variation oxalate/dithionite extraction metrics for Fe and Al, it was only $[Al]_d$ that, on its own, showed any marked association with [C], and with this being or the LAC soils (τ = 0.37). Although we do also note that τ = 0.29 for the HAC soils and τ = 0.28 for the dataset as whole.

Also of note are the many cases where there are reasonably high τ found for both the LAC and HAC

soils, but not for the Arenic ones: for example in the  correlations between Total Reserve Bases,

$Σ_B$, and organic matter CN ratio for which we observe τ = -0.44 for LAC soils and τ = -0.56 for HAC soils, but with a value of only τ = -0.03 for the soils in the Arenic cluster.

**3.4 Carbon/soil texture associations**

With a high τ observed for several [C] vs. soil texture  relationships (Section 3.3), the correlations between soil carbon content and $Φ_{clay}$ are shown  in Fig, 5 with a separate panel used for each of the three soil clusters; and with each panel having different ranges for both the *x*- and *y*-ordinates. For the

LAC soils (Fig 5a) strong linear relationship exists ($r^2$ = 0.58) and with there being little apparent difference between the Ferralsol and Acrisol RSGs. But when LAC OLS regression line is repeated again within the Arenic soil group [C]  $Φ_{clay}$  graph of Fig 5b (for which we also note that the variability in $Φ_{clay}$ is only one-tenth of that for Fig 5a and with

[revised manuscript text omitted]

Our findings do not negate the possibility that future climate changes will have a significant impact on
soil carbon stocks in the Amazon Basin. For example, Cotrufo et al., (2013) have postulated that although
interactions of organic materials within the soil mineral matrix are the ultimate controllers of SOM

stabilization over long timescales, it is the microbially mediated delivery of organic products to this matrix that provides the critical link between plant litter inputs and what products are available for stabilization. In this respect a consideration of depths substantially greater than the upper 0.3 m examined here must also be critical for the accurate determination of any future changes in climate stocks, as below 0.3 m Amazon Basin forest soil C are generally quite low, and with there likely existing reactive mineral surfaces yet to be saturated with SOM (Quesada, 2008; Quesada et al., 2010). Moreover, any future inputs into these lower layers, for example— as might be including those mediated though increased litter inputs due to likely ongoing as a consequence of [ $CO_2$ fertilization ] induced increases in stand-level productivities: (Lloyd and Farquhar, 2008), are likely to be microbially derived (Schrumpf et al., 2013). Quite likely the extent of any such additional stabilization of SOM at these lower depths will differ between HAC, LAC and Arenic soils in accordance with the different stabilization mechanisms as suggested throughout this paper. But in the absence of more detailed information and indeed, precise confirmations as to the apparent different mechanisms involved in SOM storage as suggested here; then whether or not it is really the case that Amazon forest soil C stocks are currently increasing in response to higher litter inputs with soil developmental stage also influencing that response must remain a matter of simple conjecture.

**5 Acknowledgements**

This manuscript is a product of the RAINFOR network. It integrates the effort of several researchers, technicians and field assistants across Amazonia. We thank the following individuals in particular: Michael Schwarz, Gabriel Batista de Oliveira Borges, Claudia Czimczk, Jens Schmerler, Alexandre J.B. Santos, Garreth Llloyd, Jonas O. Moraes Filho, Orlando F. C. Junior, José Edivaldo Chaves and Raimundo Nonato de Araújo Filho. Support for RAINFOR has come from the Natural Environment Research Council (NERC) Urgency Grants and "TROBIT" (NE/D005590/1) and the Gordon and Betty Moore Foundation. Jon Lloyd was supported by a Royal Society of London Research Merit Award and a São Paulo Excellence Chair.

[revised manuscript text omitted]

All soils combined (lower triangle) / Low Activity Clay Soils (upper triangle):

| | [C] | CN | [C]cs | [C]r | [C]sa | [C]pom | [C]doc | sand | clay | silt | pH | IE | TRB | [Fe]d | [Fe]o | [Fe]do | [Al]d | [Al]o | Ц1 | Ц2 |
|---|---|---|---|---|---|---|---|---|---|---|---|---|---|---|---|---|---|---|---|---|
| [C] | | -0.03 | 0.24 | 0.33 | 0.55 | 0.55 | 0.48 | -0.67 | 0.64 | 0.30 | -0.03 | 0.33 | 0.12 | 0.53 | -0.09 | 0.55 | 0.61 | -0.03 | -0.48 | -0.52 |
| CN | 0.20 | | 0.30 | 0.39 | -0.18 | 0.00 | 0.30 | -0.24 | 0.21 | -0.06 | -0.45 | 0.21 | -0.61 | -0.14 | 0.09 | -0.24 | -0.06 | 0.15 | 0.12 | 0.03 |
| [C]cs | 0.43 | -0.09 | | 0.79 | -0.21 | 0.21 | 0.45 | -0.27 | 0.24 | 0.45 | -0.36 | 0.36 | -0.27 | 0.11 | 0.12 | 0.15 | 0.33 | 0.00 | 0.03 | -0.24 |
| [C]r | 0.49 | 0.07 | 0.56 | | -0.12 | 0.42 | 0.55 | -0.36 | 0.33 | 0.30 | -0.33 | 0.39 | -0.36 | 0.08 | 0.03 | 0.12 | 0.30 | 0.03 | 0.00 | -0.27 |
| [C]sa | 0.42 | 0.42 | -0.10 | -0.01 | | 0.15 | 0.27 | -0.52 | 0.55 | 0.03 | 0.18 | 0.30 | 0.15 | 0.44 | -0.06 | 0.39 | 0.27 | 0.06 | -0.39 | -0.48 |
| [C]pom | 0.44 | 0.20 | 0.42 | 0.31 | 0.10 | | 0.21 | -0.33 | 0.30 | 0.27 | -0.06 | 0.18 | 0.03 | 0.29 | -0.24 | 0.39 | 0.33 | -0.24 | -0.21 | -0.18 |
| [C]doc | 0.68 | 0.33 | 0.32 | 0.46 | 0.48 | 0.17 | | -0.52 | 0.61 | 0.03 | -0.24 | 0.48 | -0.27 | 0.26 | 0.00 | 0.27 | 0.39 | 0.06 | -0.39 | -0.55 |
| sand | -0.31 | 0.18 | -0.27 | -0.32 | -0.05 | -0.03 | -0.29 | | -0.91 | -0.21 | 0.18 | -0.48 | 0.21 | -0.44 | -0.06 | -0.33 | -0.52 | -0.12 | 0.27 | 0.55 |
| clay | 0.34 | -0.06 | 0.11 | 0.32 | 0.23 | -0.14 | 0.48 | -0.66 | | 0.12 | -0.21 | 0.52 | -0.18 | 0.41 | 0.09 | 0.30 | 0.48 | 0.15 | -0.30 | -0.64 |
| silt | 0.01 | -0.27 | 0.43 | 0.15 | -0.31 | 0.15 | -0.12 | -0.32 | -0.02 | | 0.06 | 0.06 | 0.15 | 0.29 | 0.24 | 0.15 | 0.33 | -0.06 | 0.15 | -0.24 |
| pH | -0.07 | -0.53 | 0.17 | -0.01 | -0.32 | 0.01 | -0.21 | -0.06 | -0.06 | 0.33 | | -0.33 | 0.30 | -0.23 | -0.03 | -0.18 | -0.18 | 0.03 | 0.00 | -0.09 |
| IE | 0.20 | -0.34 | 0.37 | 0.31 | -0.09 | 0.13 | 0.11 | -0.41 | 0.35 | 0.16 | 0.32 | | -0.36 | 0.11 | -0.03 | 0.18 | 0.12 | 0.09 | -0.18 | -0.33 |
| TRB | 0.02 | -0.66 | 0.35 | 0.17 | -0.35 | -0.02 | -0.15 | -0.25 | 0.15 | 0.40 | 0.58 | 0.42 | | 0.29 | -0.06 | 0.33 | 0.09 | -0.30 | -0.21 | 0.12 |
| [Fe]d | 0.29 | -0.25 | 0.16 | 0.38 | 0.01 | -0.04 | 0.27 | -0.38 | 0.49 | 0.03 | 0.13 | 0.34 | 0.34 | | 0.08 | 0.69 | 0.72 | -0.14 | -0.32 | -0.35 |
| [Fe]o | 0.03 | -0.32 | 0.24 | 0.32 | -0.23 | -0.04 | 0.02 | -0.32 | 0.34 | 0.16 | 0.22 | 0.40 | 0.42 | 0.47 | | -0.24 | 0.24 | 0.58 | 0.24 | -0.15 |
| [Fe]do | 0.21 | -0.19 | -0.01 | 0.20 | 0.15 | -0.20 | 0.26 | -0.24 | 0.37 | -0.15 | -0.01 | 0.17 | 0.15 | 0.58 | 0.04 | | 0.45 | -0.30 | -0.52 | -0.24 |
| [Al]d | 0.21 | -0.01 | -0.01 | 0.24 | 0.19 | -0.18 | 0.32 | -0.36 | 0.55 | -0.09 | -0.15 | 0.15 | 0.05 | 0.64 | 0.34 | 0.43 | | 0.12 | -0.33 | -0.61 |
| [Al]o | 0.09 | -0.02 | -0.01 | 0.22 | 0.10 | -0.20 | 0.22 | -0.32 | 0.47 | -0.03 | -0.06 | 0.22 | 0.07 | 0.32 | 0.53 | 0.11 | 0.63 | | 0.12 | -0.27 |
| Ц1 | -0.16 | -0.22 | 0.28 | -0.03 | -0.44 | 0.20 | -0.35 | 0.10 | -0.31 | 0.34 | 0.39 | 0.04 | 0.29 | -0.20 | 0.04 | -0.32 | -0.46 | -0.35 | | 0.30 |
| Ц2 | -0.35 | -0.16 | -0.09 | -0.30 | -0.22 | -0.13 | -0.35 | 0.26 | -0.30 | 0.09 | 0.12 | 0.03 | 0.15 | -0.26 | -0.03 | -0.19 | -0.28 | -0.08 | 0.10 | |

Arenic Soils (lower triangle) / High activity clay soils (upper triangle):

| [C] | CN | [C]cs | [C]r | [C]sa | [C]pom | [C]doc | sand | clay | silt | pH | IE | TRB | [Fe]d | [Fe]o | [Fe]do | [Al]d | [Al]o | Ц1 | Ц2 | |
|---|---|---|---|---|---|---|---|---|---|---|---|---|---|---|---|---|---|---|---|---|
| | 0.10 | 0.74 | 0.62 | 0.43 | 0.72 | 0.73 | -0.26 | 0.49 | 0.10 | 0.36 | 0.72 | 0.49 | 0.38 | 0.28 | -0.05 | 0.18 | 0.18 | 0.23 | -0.26 | [C] |
| 0.40 | | -0.05 | 0.28 | 0.37 | 0.18 | 0.25 | 0.44 | -0.21 | -0.33 | -0.28 | -0.08 | -0.31 | 0.05 | 0.10 | -0.03 | 0.15 | 0.21 | -0.41 | 0.18 | CN |
| 1.00 | 0.40 | | 0.56 | 0.22 | 0.67 | 0.60 | -0.36 | 0.59 | 0.21 | 0.31 | 0.62 | 0.64 | 0.38 | 0.38 | -0.05 | 0.23 | 0.23 | 0.49 | -0.31 | [C]cs |
| 0.80 | 0.60 | 0.80 | | 0.19 | 0.44 | 0.76 | -0.13 | 0.41 | -0.03 | 0.18 | 0.44 | 0.36 | 0.62 | 0.41 | 0.13 | 0.41 | 0.31 | 0.10 | -0.49 | [C]r |
| 0.60 | 0.40 | 0.60 | 0.40 | | 0.45 | 0.32 | 0.17 | -0.04 | -0.12 | 0.19 | 0.25 | -0.09 | 0.04 | -0.04 | 0.01 | -0.06 | -0.09 | -0.09 | 0.14 | [C]sa |
| 1.00 | 0.40 | 1.00 | 0.80 | 0.60 | | 0.54 | -0.23 | 0.36 | 0.08 | 0.18 | 0.49 | 0.31 | 0.36 | 0.51 | -0.28 | 0.31 | 0.41 | 0.15 | -0.13 | [C]po |
| 1.00 | 0.40 | 1.00 | 0.80 | 0.60 | 1.00 | | -0.09 | 0.41 | -0.07 | 0.30 | 0.60 | 0.38 | 0.49 | 0.36 | 0.09 | 0.17 | 0.20 | 0.12 | -0.30 | [C]doc |
| -0.60 | -0.40 | -0.60 | -0.40 | -0.60 | -0.60 | -0.60 | | -0.56 | -0.69 | -0.33 | -0.33 | -0.51 | -0.10 | -0.26 | 0.18 | 0.00 | -0.15 | -0.41 | 0.33 | sand |
| 0.20 | 0.00 | 0.20 | 0.00 | 0.60 | 0.20 | 0.20 | -0.60 | | 0.26 | 0.31 | 0.41 | 0.64 | 0.28 | 0.44 | -0.15 | 0.13 | 0.28 | 0.49 | -0.46 | clay |
| 0.60 | 0.40 | 0.60 | 0.40 | 0.60 | 0.60 | 0.60 | -1.00 | 0.60 | | 0.33 | 0.23 | 0.36 | -0.05 | 0.05 | -0.08 | -0.10 | 0.10 | 0.21 | -0.13 | silt |
| 0.32 | 0.53 | 0.32 | 0.53 | -0.11 | 0.32 | 0.32 | 0.11 | -0.53 | -0.11 | | 0.59 | 0.56 | 0.26 | 0.00 | 0.18 | -0.05 | -0.10 | 0.46 | -0.23 | pH |
| 0.11 | 0.11 | 0.11 | -0.11 | 0.53 | 0.11 | 0.11 | -0.11 | 0.11 | 0.11 | 0.00 | | 0.56 | 0.26 | 0.21 | -0.08 | 0.00 | 0.05 | 0.36 | -0.13 | IE |
| 0.60 | 0.00 | 0.60 | 0.40 | 0.20 | 0.60 | 0.60 | -0.60 | 0.60 | 0.60 | -0.11 | -0.32 | | 0.28 | 0.23 | 0.00 | 0.13 | 0.13 | 0.64 | -0.36 | TRB |
| 0.00 | -0.20 | 0.00 | 0.20 | -0.40 | 0.00 | 0.00 | 0.00 | 0.00 | 0.00 | 0.32 | -0.53 | 0.40 | | 0.44 | 0.21 | 0.64 | 0.28 | 0.13 | -0.62 | [Fe]d |
| -0.60 | -0.40 | -0.60 | -0.40 | -1.00 | -0.60 | -0.60 | 0.60 | -0.60 | -0.60 | 0.00 | 0.11 | -0.53 | -0.20 | | -0.36 | 0.49 | 0.74 | 0.08 | -0.36 | [Fe]o |
| 0.60 | 0.00 | 0.60 | 0.40 | 0.60 | 0.60 | 0.60 | -0.20 | 0.60 | 0.60 | -0.11 | 0.11 | 0.60 | 0.00 | -0.60 | | 0.05 | -0.41 | 0.05 | -0.23 | [Fe]do |
| 0.00 | -0.60 | 0.00 | -0.20 | 0.00 | 0.00 | 0.00 | 0.00 | 0.40 | 0.00 | -0.11 | -0.11 | 0.40 | 0.60 | 0.00 | 0.40 | | 0.49 | 0.03 | -0.36 | [Al]d |
| 0.60 | 0.00 | 0.60 | 0.40 | 0.60 | 0.60 | 0.60 | -0.20 | 0.60 | 0.20 | -0.11 | 0.11 | 0.60 | 0.00 | -0.60 | 1.00 | 0.40 | | -0.03 | -0.15 | [Al]o |
| -0.20 | 0.00 | -0.20 | 0.00 | -0.60 | -0.20 | -0.20 | 0.20 | -0.60 | -0.20 | 0.53 | -0.53 | -0.20 | 0.40 | 0.60 | -0.60 | 0.00 | -0.60 | | -0.21 | Ц1 |
| 0.00 | -0.20 | 0.00 | -0.20 | 0.40 | 0.00 | 0.00 | 0.00 | 0.40 | 0.00 | -0.74 | 0.32 | 0.00 | -0.60 | -0.40 | 0.40 | -0.20 | 0.40 | -0.80 | | Ц2 |
| [C] | CN | [C]cs | [C]r | [C]sa | [C]pom | [C]doc | sand | clay | silt | pH | IE | TRB | [Fe]d | [Fe]o | [Fe]do | [Al]d | [Al]o | Ц1 | Ц2 | |

High activity clay soils (right-side label for the lower block)

**Table 8. Kendall's τ correlations for soil organic carbon fractions and a range of soil and mineralogical properties.– Four one sided correlation matrices are shown *viz.* for each of the Arenic, LAC and HAC clusters as well as for the (combined) dataset as a whole. Data shown here is a subset of our entire dataset ($n > 30$). We have indicated in bold all cases where the probability of Type-II error being less than 0.05. For the entire dataset, a $\tau > 0.22$ is associated to a probability of $p < 0.05$. For the HAC soil cluster with $n = 13$ the equivalent value is $\tau > 0.36$ and for LAC, with n = 12 the value is $\tau > 0.39$. For the Arenic soil cluster, with only n = 5, the associated probability of $p < 0.05$ requires a $\tau > 0.80$.– In all cases where one or more of the four grouping has $p > 0.05$, we have indicated – using different colours to help cross-referencing across the four diagonal matrices.**

**Table 9. Mean soil organic carbon stocks (0-30 cm) for 12 RGS examined in this study. Stocks from (Batjes, 1996)Batjes, (1996) are also given for comparison.**

| RSG | *n* | Soil carbon concentration | | Soil carbon stock | | SOTER-LAC estimated soil carbon stock | |
|---|---|---|---|---|---|---|---|
| | | Mean (mg g$^{-1}$) | C.V. | Mean (t ha$^{-1}$) | C.V. | Mean (t ha$^{-1}$) | C.V. |
| **Acrisol** | 18 | 16.3 | 0.35 | 49.5 | 0.27 | 44.0 | 0.50 |
| **Alisol** | 20 | 16.6 | 0.28 | 45.6 | 0.27 | 85.7 | 0.42 |
| **Arenosol** | 6 | 12.3 | 0.23 | 29.6 | 0.31 | 20.7 | 0.50 |
| **Cambisol** | 19 | 21.3 | 0.63 | 58.9 | 0.39 | 55.9 | 0.61 |
| **Ferralsol** | 34 | 17.1 | 0.35 | 47.3 | 0.26 | 50.5 | 0.48 |
| **Fluvisol** | 5 | 21.0 | 0.33 | 54.6 | 0.33 | 34.2 | 0.52 |
| **Gleysol** | 10 | 24.5 | 1.03 | 70.1 | 0.84 | 67.4 | 0.62 |
| **Leptosol** | 2 | 32.0 | 0.75 | 115.2 | 0.63 | 51.5 | 0.63 |
| **Lixosol** | 3 | 21.9 | 0.36 | 65.4 | 0.17 | 38.5 | 0.45 |
| **Luvisol** | 2 | 15.3 | 0.57 | 43.8 | 0.46 | 46.7 | 0.51 |
| **Plinthosol** | 18 | 14.2 | 0.40 | 41.1 | 0.44 | 34.0 | 0.48 |
| **Podzol** | 7 | 48.3 | 0.92 | 98.9 | 1.32 | 54.9 | 0.54 |

---

## Editor Comment (EC3) · Elizabeth Bach (Editor) · 18 Oct 2019

Dear Reviewers, I apologize for a tardy response here. Quesada et al. have posted a track-changes version of their revised manuscript. It is available as a download associated with "AC4" above. Please take a look and respond with any remaining concerns.

Thank you, Elizabeth
* * *